# MMICL: Empowering Vision-language Model with Multi-Modal In-Context Learning

**Haozhe Zhao**[*§1,2], **Zefan Cai**[*1], **Shuzheng Si**[*1], **Xiaojian Ma**[3], **Kaikai An**[1],
**Liang Chen**[1], **Zixuan Liu**[4], **Sheng Wang**[4], **Wenjuan Han**[†5], **Baobao Chang**[†1]

[1]National Key Laboratory for Multimedia Information Processing, Peking University
[2]School of Software and Microelectronics, Peking University, China
[3]National Key Laboratory of General Artificial Intelligence, BIGAI
[4]Paul G. Allen School of Computer Science and Engineering, University of Washington
[5]Beijing Jiaotong University
mimazhe55360@gmail.com, zefncai@gmail.com, sishuzheng@stu.pku.edu.cn
**https://github.com/PKUnlp-icler/MIC**

## Abstract

Since the resurgence of deep learning, vision-language models (VLMs) enhanced by large language models (LLMs) have grown exponentially in popularity. However, while LLMs can utilize extensive background knowledge and task information with in-context learning, most VLMs still struggle with understanding complex multi-modal prompts with multiple images, making VLMs less effective in downstream vision-language tasks. In this paper, we address the limitation above by 1) introducing vision-language Model with **M**ulti-**M**odal **I**n-**C**ontext **L**earning(MMICL), a new approach to allow the VLM to deal with multi-modal inputs efficiently; 2) proposing a novel context scheme to augment the in-context learning ability of the VLM; 3) constructing the Multi-modal In-Context Learning (MIC) dataset, designed to enhance the VLM's ability to understand complex multi-modal prompts. Our experiments confirm that MMICL achieves new state-of-the-art zero-shot performance on a wide range of general vision-language tasks, especially for complex benchmarks, including MME and MMBench. Our analysis demonstrates that MMICL effectively tackles the challenge of complex multi-modal prompt understanding and emerges the impressive ICL ability. Furthermore, we observe that MMICL successfully alleviates language bias in VLMs, a common issue for VLMs that often leads to hallucination when faced with extensive textual context. Our code, dataset, dataset tool, and model are available at https://github.com/PKUnlp-icler/MIC.

## 1 Introduction

General-purpose vision-language pre-trained models (VLMs) have made significant advancements (Li et al., 2022; 2023d;g; Zhu et al., 2023; Li et al., 2023b). Recent VLMs mostly augment a large language model (LLM) with a visual encoder and exhibit impressive zero-shot capacities in various visual tasks. However, unlike LLMs that can extract rich background knowledge and task information from the prompt with *in-context learning* (ICL), most VLMs still struggle to understand complex multi-modal prompts that include multiple images. Previous studies (Li et al., 2023d;b) primarily focus on handling the user queries with a single image rather than multi-modal prompts with interleaved multiple images and text. Although some VLMs like Flamingo (Alayrac et al., 2022) and Kosmos-1 (Huang et al., 2023b) can handle user queries with multiple images, their pre-training data can not provide more sophisticated multi-modal prompts than interleaved image and text crawled from the web (Awadalla et al., 2023). Hence, there is a gap between the prompts used in pre-training these VLMs and the user queries in real-world scenarios, which always contain multiple images and

---

[*]Equal contribution.
[§]Project lead.
[†]Corresponding author.

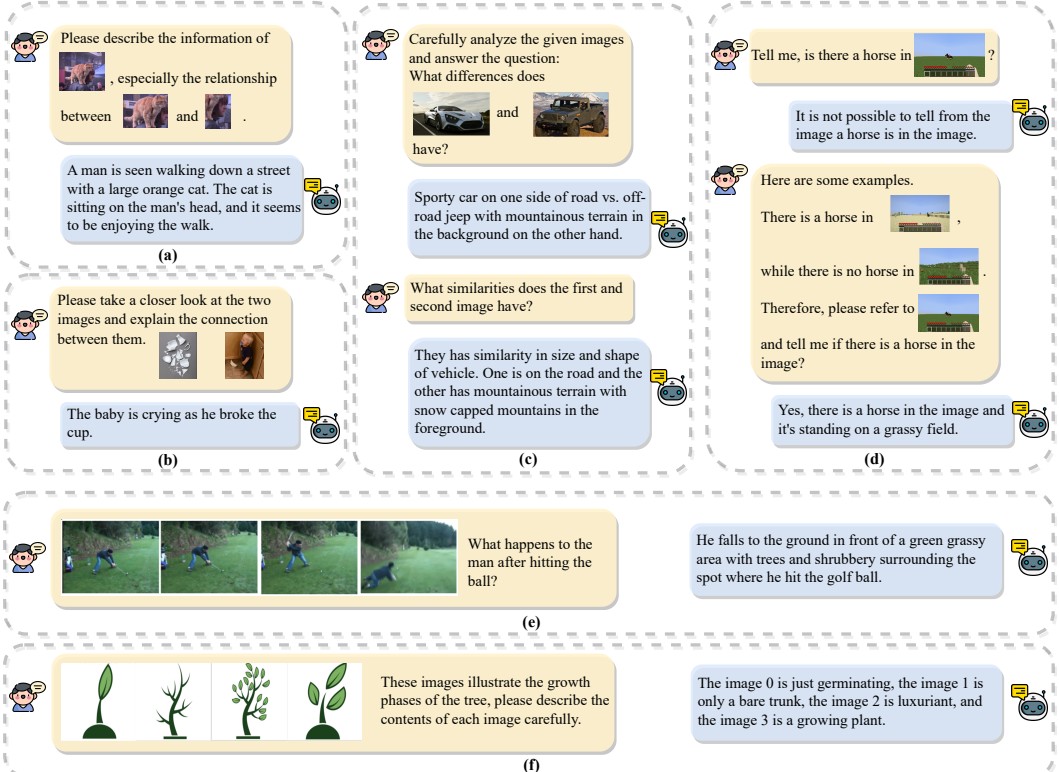

Figure 1: Examples of vision-language dialogue generated by MMICL typically contain prompts with interleaved images and text. MMICL understands spatial (**a**), logical (**b**), and temporal (**e**) relationships among images. MMICL can also grasp text-to-image references as (**c**),(**d**) and (**f**).

more sophisticated text. Specifically, these VLMs may suffer from the following three limitations, which makes VLMs less effective in downstream vision-language tasks.

**Hard to Understand Text-to-Image Reference**: Previous studies rarely attempt to address the issue of text-to-image reference in the multi-modal prompts. However, there are often intricate referential relationships between the text and images in user queries, with different words mentioning different images. For example, the user may ask a specific question about multiple images(Fig. 1.c and Fig. 1.f) or use multiple images as exemplars to ask the question only about a specific image(Fig. 1.d). However, the training data used in previous studies (Li et al., 2023d; Alayrac et al., 2022; Huang et al., 2023a) are crawled from the web and may lack explicit text-to-image references. VLMs, thus might fail to handle user queries involving intricate text-to-image references.

**Hard to Understand the Relationships between Multiple Images**: There are often spatial, temporal, and logical relationships between multiple images, and correctly understanding them allows the model to handle user queries better. However, the pre-training data used by previous VLMs (Alayrac et al., 2022) are collected from the internet, lacking close connections among images, especially when these images are far apart on the same webpage. It hampers the ability of VLMs to understand the intricate relationships among the images and further limits their reasoning ability.

**Hard to Learn from In-Context Multi-Modal Demonstrations**: Previous studies have shown that pretrained LLMs can benefit from few in-context demonstrations (Brown et al., 2020; Dong et al., 2023). However, the ICL ability of current VLMs is rather limited, specifically: 1) VLMs like BLIP-2 (Li et al., 2023d), LLaVA (Li et al., 2023b) only support multi-modal prompts with a single image, hampering their abilities to use multiple multi-modal demonstrations to enhance their performance during the inference; 2)Although VLMs such as Flamingo (Alayrac et al., 2022) support multi-image inputs during pretraining and emerge ICL abilities, their context schemes fail to provide text-image references and closely related images. It inhibits them from offering sophisticated enough prompts to the VLMs, thereby limiting the effectiveness of their ICL ability. Besides, the lack of further supervised instruction tuning hinders their effectiveness across downstream tasks.

In this paper, to address the aforementioned limitations 1) We present MMICL, a new approach to allow VLMs to efficiently deal with multi-modal inputs, including relationships among multiple

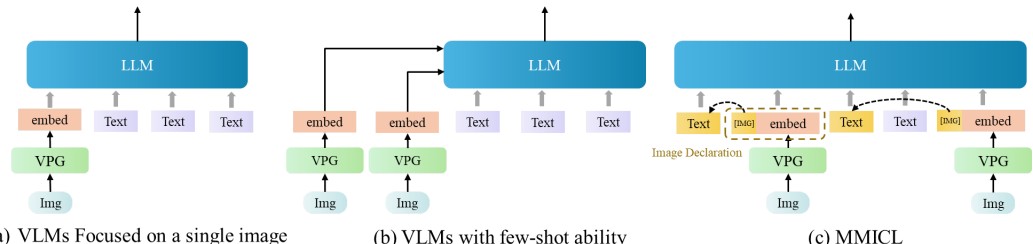

(a) VLMs Focused on a single image      (b) VLMs with few-shot ability      (c) MMICL

Figure 2: **Comparison of different VLM architectures:** VLMs focused on a single image, VLMs with few-shot ability, and MMICL with equal treatment of image and text representation.

images and text-to-image references. 2) We propose a novel context scheme in which incorporating an extra image declaration section, along with the inclusion of image proxy tokens, enhances the ICL ability of the VLM. 3) We construct a multi-modal in-context learning dataset in accordance with the proposed scheme. The dataset is adapted from a range of existing datasets and can be used to provide support for the training of more capable VLMs.

Our experiments show that MMICL achieves new state-of-the-art performance on various of vision-language benchmarks including MME (Fu et al., 2023) and MMBench (Liu et al., 2023c) [1]. Comprehensive examinations of the three limitations we aim to address reveal that MMICL exhibits exceptional ability in understanding text-to-image references (13-points improvement on the vision-language compositionality benchmark, Winoground (Thrush et al., 2022a)) and intricate relationships among images(12-points improvement on the multi-image reasoning benchmark, RAVEN (Huang et al., 2023a)). Moreover, MMICL demonstrates impressive multi-modal ICL performance across various tasks. We also observe that MMICL efficiently mitigates the language bias, which often causes VLMs to ignore visual contents when facing extensive textual contexts, leading to hallucinations.

## 2 MMICL

### 2.1 MODEL ARCHITECTURE

Most VLMs utilize Visual-Prompt Generators (VPG) (e.g., Resampler (Alayrac et al., 2022), Q-former (Li et al., 2023d)) to extract visual embeddings from the image features encoded by vision backbones and use visual embeddings to help LLMs understand visual inputs. The model architecture shown in Fig. 2.a belongs to VLMs that focus on prompts with a single image, such as Blip-2 (Li et al., 2023d), which always places the image at the top of the entire input and can not handle the inputs with multiple images. In Fig. 2.b, VLMs with few-shot ability, such as Flamingo (Alayrac et al., 2022), encode images into image embeddings with a fixed number of visual tokens and inserts new gated cross-attention layers into the LLM to inject visual features. Different from previous work, MMICL shown in Fig. 2.c treats image and text representations equally and establishes the reference between image and text via image declaration. It enables users to have the flexibility to input multiple images and text in any desired order, with no restrictions on the quantity or placement of images in contexts. As shown in Fig. 5, each given image is encoded by a vision encoder (e.g., ViT (Radford et al., 2021)) to get the image representation. Then, we use the Q-former as the VPG to encode images into embeddings understandable by the language model. We utilize a fully connected layer as the projection layer to convert each visual embedding to the same dimension as the text embedding of the LLM. Finally, we combine the visual and text embeddings into an interleaved style and feed them into the LLM. This design is a natural extension of the original attention mechanism in LLMs. We set the weights for mapping query and value vectors in the attention layer of LLM as learnable to better adapt to multi-modal prompts with multiple images. More details are presented in Appendix E.

### 2.2 THE DESIGN OF CONTEXT SCHEME OF MMICL

In this section, we outline the design of the Context Scheme for MMICL. The proposed scheme is devised to proficiently transform the interleaved image-text data into the training context for MMICL.

---

[1] Results of MMICL are submitted on August 28th, 2023.

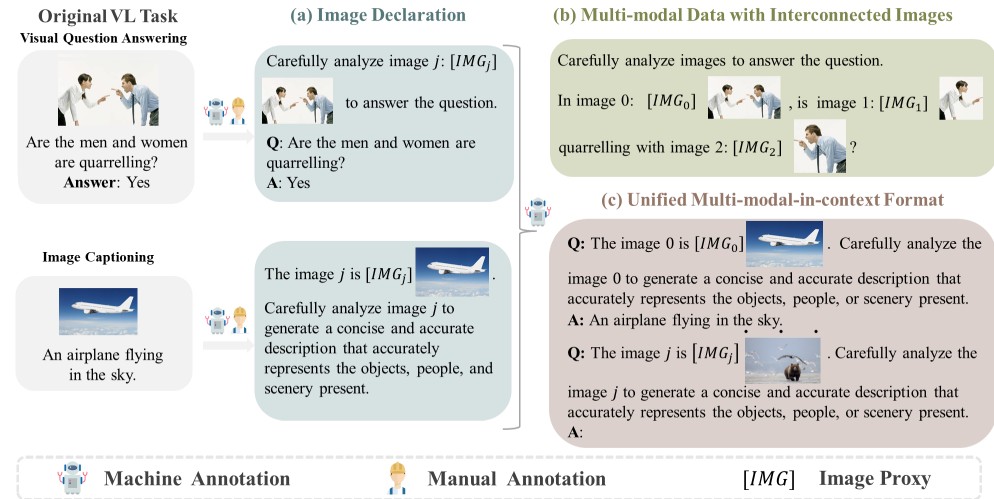

Figure 3: Context scheme for MMICL, which seamlessly transforms the interleaved image-text data into training context in a unified format.

### 2.2.1 IMAGE DECLARATION

Users may use textual descriptions to refer to particular images in their queries. Such reference can provide information about the visual content mentioned in the text to the VLM, allowing it to learn alignment between two modalities. To precisely link text and image, we form image declaration templates for each image in mixed inputs, as shown in Fig. 3.a. Firstly, we allocate a unique image proxy ([**IMG**$j$]) to reference the visual embedding of image $j$, which provides a unique identifier for VLMs to index and distinguish between visual and text embeddings. Then, we utilize natural language prompts to establish references between text and image. Incorporating the explicit text-to-image reference in the image declaration assists the model in correlating the text with the appropriate image. Meanwhile, the image declaration, maintained as textual content, can also preserve the flexibility to appear at any position within the prompt. Each instance $\mathbf{I}_i$ follows the structure, where the $\mathbf{X}_i$ symbolizes the set of image decorations that can be placed anywhere within the instance $\mathbf{I}_i$. $\mathbf{q}_i$ and $\mathbf{a}_i$ denote the question with instruction and corresponding answer, respectively.

$$\mathbf{I}_i = (\mathbf{X}_i, \mathbf{q}_i, \mathbf{a}_i) \tag{1}$$

### 2.2.2 MULTI-MODAL DATA WITH INTERCONNECTED IMAGES

To incorporate abundant multi-image information within the context scheme of MMICL, we generate interconnected multi-image data that includes spatial, logical, and temporal relationships. It aids MMICL in understanding the intricate relationships among images in user queries. Specifically, we derive frames from videos to build multi-image data. The frames extracted from video inherently sustain close temporal and spatial relations, which infuse spatial and temporal correlation information among images into the context scheme. Besides, we build multi-image data from images depicting multiple object interactions. We detect the objects within the image and generate bounding boxes for each object. We acquire multiple sub-images of different objects by cropping the image according to bounding boxes. We then replace the textual references to these objects with their corresponding cropped images, thus forming interleaved multi-modal data with logical and causal interconnected images, as delineated in Fig. 3.b and Fig. 4. Each instance $\mathbf{I}_i$ comprises a question-answer text pair along with $K$ images, where the $\mathbf{x}_{i,k} \in \mathbf{X}_i$ represents the image declaration for the $k$-th image.

$$\mathbf{I}_i = (\{\mathbf{x}_1, \mathbf{x}_2, \ldots, \mathbf{x}_k\}, \mathbf{q}_i, \mathbf{a}_i) \tag{2}$$

### 2.2.3 UNIFIED MULTI-MODAL IN-CONTEXT FORMAT FOR DIFFERENT TASKS

We propose a design for producing multi-modal in-context learning data for different tasks to enrich the context scheme of MMICL. It aims to improve the instruction-aware ability of VLM and expand its abilities for proficient multi-modal in-context learning. Specifically, we start by crafting diverse instructions for each task and generate different templates for the task utilizing these instructions. We then fill in the randomly selected template with the original task to assemble data equipped with instructions as Appendix G. Moreover, we convert the data into a multi-modal in-context format by

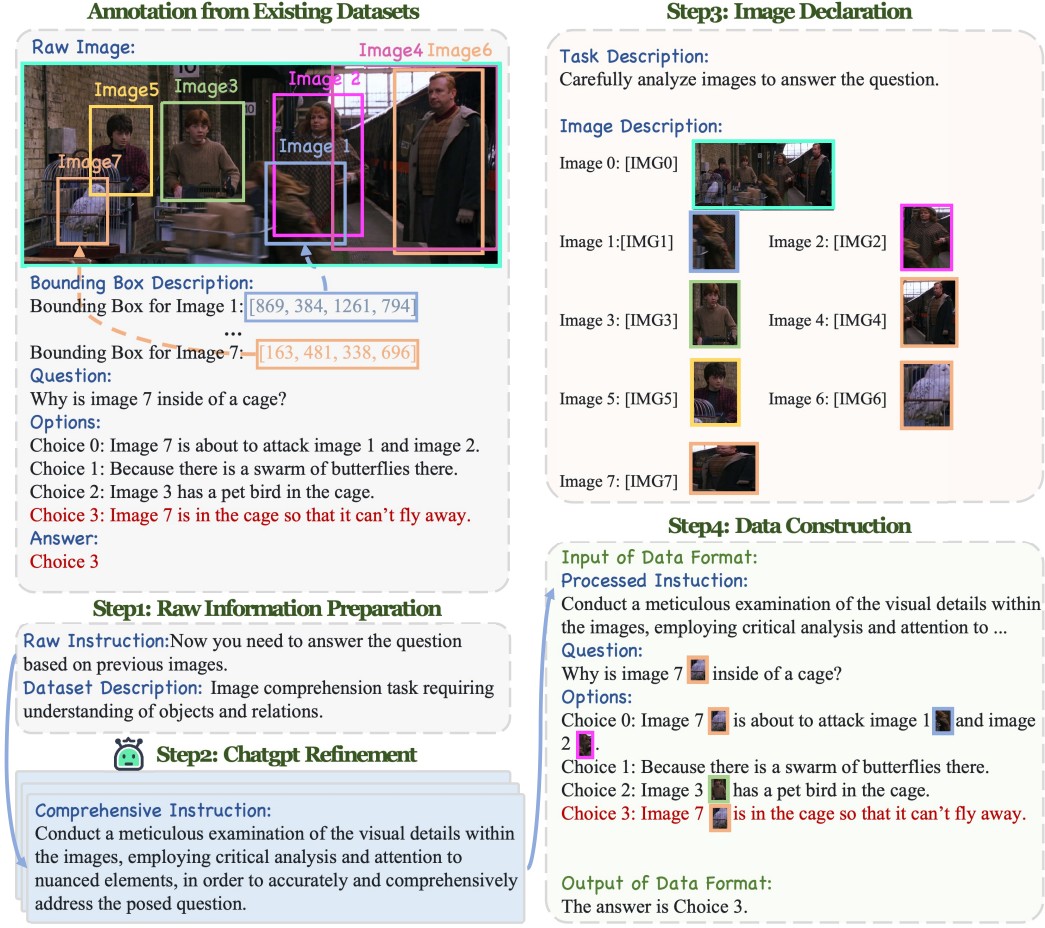

Figure 4: Illustration of automatic data construction pipeline for multi-model data with interconnected images. The automatic construction is based on the existing annotation of VCR dataset (Zellers et al., 2019) without human involvement. ChatGPT is used for instruction refinement.

constructing few-shot exemplars generated by sampling instances from the data. These exemplars are combined with the input instance to produce the multi-modal in-context data. In this way, we can transform all tasks into a unified multi-modal in-context format, as illustrated in Fig. 3.c. This method facilitates amassing an extensive amount of high-quality data from different tasks, enriching the context scheme of MMICL with an abundant diversity of multi-modal in-context data teeming with diverse instructions. Ultimately, this improves the model's ability to follow instructions and multi-modal in-context learning ability. Each instance $\mathbf{I}_i$ comprises N exemplars.

$$\mathbf{I}_i = (\{\mathbf{P}_1, \cdots, \mathbf{P}_N\}, \mathbf{X}_i, \mathbf{q}_i, \mathbf{a}_i) \tag{3}$$

Each exemplar $\mathbf{P}_j = (\mathbf{X}_j, \mathbf{q}_j, \mathbf{a}_j)$, $\mathbf{X}_j$ denotes the image declaration of the $j$-th exemplar. $\mathbf{q}_j$ and $\mathbf{a}_j$ denote the question and answer for the $j$-th exemplar, respectively.

## 2.3 MULTIMODALITY IN-CONTEXT LEARNING (MIC) DATASET CONSTRUCTION

To help VLMs understand the complex prompts, we construct MIC dataset by gathering data from public data resources and converting them based on the context scheme. It has three key aspects: 1) image declaration, 2) multi-modal data with closely related images, and 3) multi-modal in-context data for different tasks. Training set of MIC comes from 16 datasets across 8 categories, while the test set comes from 18 datasets across 10 categories.

Our dataset is automatically constructed based on existing datasets. Firstly, we created an image declaration for each instance in all datasets to produce datasets with explicit text-to-image reference. Secondly, we created an instruction template for each dataset and asked Chatgpt to rewrite instructions, filling in the data from the existing datasets to obtain a dataset with diverse instruction formats. Finally, we used those datasets with instructions to construct the MIC dataset according to our proposed

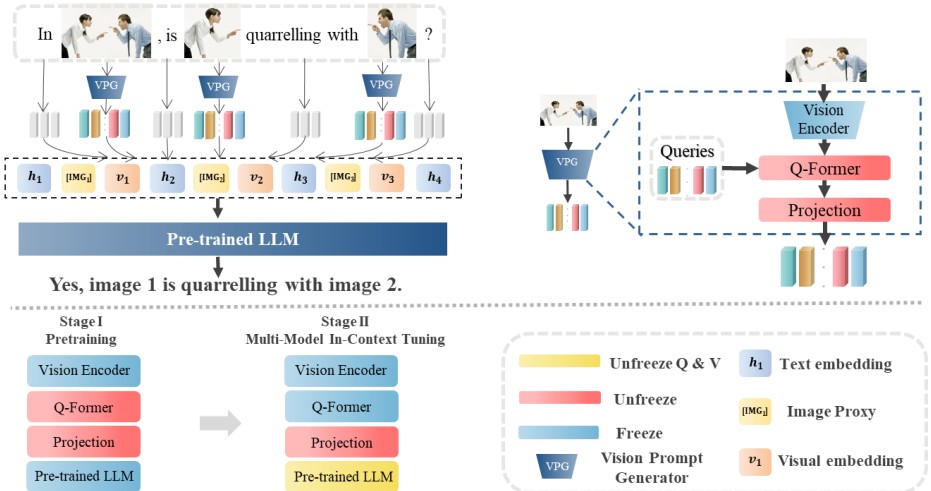

Figure 5: Illustration of MMICL architecture and training paradigm. The upper part denotes the overview of model architecture, and the bottom denotes the pipeline of two-stage training paradigm.

context scheme. For the example presented in Fig. 3 and Fig. 4 (e.g., two people quarreling with each other), we constructed the data based on existing annotations (i.e., bounding boxes and the relation between bounding boxes) provided by the VCR dataset (Zellers et al., 2019). Additionally, we also constructed an in-context learning dataset by sampling examples from the original dataset. We also extracted eight frames per video from video datasets to generate the multi-modal data with interconnected images. Details are presented at Appendix D.

We convert all data into a vision-language Q&A format to create high-quality multi-modal training data and accumulate 5.8M samples in MIC dataset. Due to resource constraints, we use approximately 10% of the data with the sampling strategy described in Appendix F to finetune MMICL. It is anticipated that a larger model trained on all of our data would yield a more promising result.

## 2.4 TRAINING PARADIGM

**Stage I: Pretraining.** This stage aims to assist the model in aligning the image and text embeddings. During this stage, both the vision encoder and the LLM remain frozen. The VPG (i.e., Q-Former) and projection layer are trained to learn visual embeddings that can be interpreted by the LLM.

**Stage II: Multi-Modal In-Context Tuning.** In this stage, we aim to address the aforementioned limitations and take our model a step further by extending it to multi-modal in-context learning. Specifically, we aim to make the model understand the intricate referential relationships between the text and images and the complex relationships among multiple images and ultimately acquire a proficient multi-modal in-context learning ability. Therefore, we perform multi-modal In-Context Tuning on MIC dataset. During the stage II, we freeze the image encoder, Q-former, and LLM while jointly training the projection layer and query and value vectors. Details can be found in Appendix H.

## 3 EXPERIMENT

### 3.1 EXPERIMENTAL SETUP

**Evaluation Setup.** We aim to develop general-purpose VLMs that can generally adapt to diverse, challenging multi-modal prompts. Therefore, we evaluate our models in several vision-language benchmarks, including tasks that involve images and videos. The metrics used in these benchmarks and further details are shown in Appendix M.

**Models and Baselines.** We provide two versions of MMICL: (1) MMICL (FLAN-T5) which uses BLIP-2 (Li et al., 2023d) as the backbone and (2) MMICL (Instruct-FLAN-T5) which uses Instruct-BLIP (Dai et al., 2023) as the backbone. We also adopt XL and XXL of FLANT5 (Chung et al., 2022) model for both versions. We compare MMICL with following strong baselines: **Flamingo** (Alayrac et al., 2022), **KOSMOS-1** (Huang et al., 2023a), **BLIP-2-FLAN-T5**, **InstructBLIP-FLAN-T5**,

| Model | Model Size | Cognition | | | | Perception | | | | | | | | | | Total Avg. |
|---|---|---|---|---|---|---|---|---|---|---|---|---|---|---|---|---|
| | | Comm. | Num. | Text. | Code. | Existen. | Count | Pos. | Color | OCR | Poster | Cele. | Scene | Land. | Art. | |
| LLaVA | 13B | 57.14 | 50.00 | 57.50 | 50.00 | 50.00 | 50.00 | 50.00 | 55.00 | 50.00 | 50.00 | 48.82 | 50.00 | 50.00 | 49.00 | 51.25 |
| MiniGPT-4 | 13B | 59.29 | 45.00 | 0.00 | 40.00 | 68.33 | 55.00 | 43.33 | 75.00 | 57.50 | 41.84 | 54.41 | 71.75 | 54.00 | 60.50 | 51.85 |
| MultiModal-GPT | 9B | 49.29 | 62.50 | 60.00 | 55.00 | 61.67 | 55.00 | 58.33 | 68.33 | 82.50 | 57.82 | 73.82 | 68.00 | 69.75 | 59.50 | 62.97 |
| VisualGLM-6B | 6B | 39.29 | 45.00 | 50.00 | 47.50 | 85.00 | 50.00 | 48.33 | 55.00 | 42.50 | 65.99 | 53.24 | 146.25 | 83.75 | 75.25 | 63.36 |
| VPGTrans | 7B | 64.29 | 50.00 | 77.50 | 57.50 | 70.00 | 85.00 | 63.33 | 73.33 | 77.50 | 84.01 | 53.53 | 141.75 | 64.75 | 77.25 | 74.27 |
| LaVIN | 13B | 87.14 | 65.00 | 47.50 | 50.00 | 185.00 | 88.33 | 63.33 | 75.00 | 107.50 | 79.59 | 47.35 | 136.75 | 93.50 | 87.25 | 86.66 |
| LLaMA-Adapter-V2 | 7B | 81.43 | 62.50 | 50.00 | 55.00 | 120.00 | 50.00 | 48.33 | 75.00 | 125.00 | 99.66 | 86.18 | 148.50 | 150.25 | 69.75 | 87.26 |
| mPLUG-Owl | 7B | 78.57 | 60.00 | 80.00 | 57.50 | 120.00 | 50.00 | 50.00 | 55.00 | 65.00 | 136.05 | 100.29 | 135.50 | 159.25 | 96.25 | 88.82 |
| InstructBLIP | 12.1B | 129.29 | 40.00 | 65.00 | 57.50 | 185.00 | 143.33 | 66.67 | 153.33 | 72.50 | 123.81 | 101.18 | 153.00 | 79.75 | 134.25 | 107.47 |
| BLIP-2 | 12.1B | 110.00 | 40.00 | 65.00 | 75.00 | 160.00 | 135.00 | 73.33 | 148.33 | 110.00 | 141.84 | 105.59 | 145.25 | 138.00 | 136.50 | 113.13 |
| Lynx | 7B | 110.71 | 17.50 | 42.50 | 45.00 | 195.00 | 151.67 | 90.00 | 170.00 | 77.50 | 124.83 | 118.24 | 164.50 | **162.00** | 119.50 | 113.50 |
| GIT2 | 5.1B | 99.29 | 50.00 | 67.50 | 45.00 | 190.00 | 118.33 | **96.67** | 158.33 | 65.00 | 112.59 | 145.88 | 158.50 | 140.50 | **146.25** | 113.85 |
| Otter | 9B | 106.43 | 72.50 | 57.50 | 70.00 | **195.00** | 88.33 | 86.67 | 113.33 | 72.50 | 138.78 | **172.65** | **158.75** | 137.25 | 129.00 | 114.19 |
| Cheetor | 7B | 98.57 | 77.50 | 57.50 | **87.50** | 180.00 | 96.67 | 80.00 | 116.67 | 100.00 | 147.28 | 164.12 | 156.00 | 145.73 | 113.50 | 115.79 |
| LRV-Instruction | 7B | 100.71 | 70.00 | 85.00 | 72.50 | 165.00 | 111.67 | 86.67 | 165.00 | 110.00 | 139.04 | 112.65 | 147.98 | 160.53 | 101.25 | 116.29 |
| BLIVA | 12.1B | 136.43 | 57.50 | 77.50 | 60.00 | 180.00 | 138.33 | 81.67 | **180.00** | 87.50 | **155.10** | 140.88 | 151.50 | 89.50 | 133.25 | 119.23 |
| MMICL | 12.1B | **136.43** | **82.50** | **132.50** | 77.50 | 170.00 | **160.00** | 81.67 | 156.67 | 100.00 | 146.26 | 141.76 | 153.75 | 136.13 | 135.50 | **129.33** |

Table 1: Evaluation results on the MME. Top two scores are highlighted and underlined, respectively.

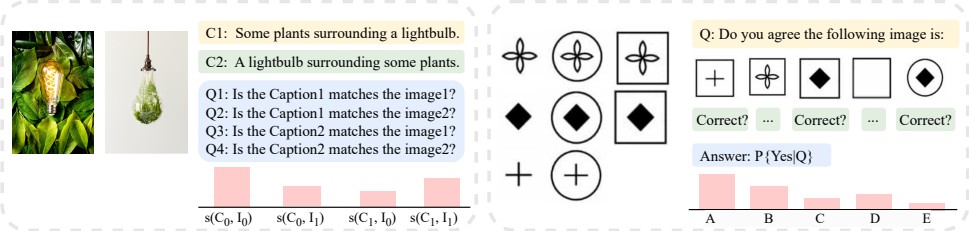

Figure 6: Illustration of two complex vision language reasoning tasks: **Winoground** (Thrush et al., 2022b) (Left) and **RAVEN** (Zhang et al., 2019) (Right).

**Shikra** (Chen et al., 2023a), **Otter** (Li et al., 2023a), **Ying-VLM** (Li et al., 2023e). The details of MMICL and baselines are shown in Appendix H, and Appendix O.

## 3.2 General Performance Evaluations

We evaluate the general performance of MMICL on both MME (Fu et al., 2023) and MMBench (Liu et al., 2023c) benchmarks[2]. MME evaluates VLMs with 14 sub-tasks that encompass cognition and perception abilities. Results in Table 1 show that MMICL can achieve the best average scores compared with current VLMs on cognition and perception tasks. MMICL also demonstrates outstanding performance and significantly surpasses other VLMs on the MMBench benchmark, which thoroughly evaluates the diverse skills of VLMs. The detailed results are presented in Table 22. See Appendix I and J for MMICL's evaluation detail and comparisons with other VLMs.

## 3.3 Performance Prob

### 3.3.1 Understanding Text-to-Image Reference

The Winoground (Thrush et al., 2022b) proposes a task of correctly matching two given images and captions, as depicted in the left of Fig. 6. The challenge lies in the fact that both captions consist of the exact same words, albeit in a different order. VLMs must compare both images and texts to discern their subtle differences and capture the implicit reference between them. Therefore, we select the Winoground to evaluate whether VLMs understand the text-to-

Table 2: Results on Winoground across text, image and group score metrics.

| Model | Text | Image | **Group** |
|---|---|---|---|
| MTurk Human | 89.50 | 88.50 | 85.50 |
| VQ2 (Yarom et al., 2023) | 47.00 | 42.20 | 30.50 |
| PALI (Chen et al., 2022) | 46.50 | 38.00 | 28.75 |
| Blip-2 (Li et al., 2023d) | 44.00 | 26.00 | 23.50 |
| *GPT4-V (Wu et al., 2023)* | *69.25* | *46.25* | *39.25* |
| MMICL (FLAN-T5-XXL) | 45.00 | 45.00 | **43.00** |

image reference. MMICL is given two images and two captions in each prompt during evaluation. Results in Table 2 demonstrate that MMICL captures the referential relationship between image and text, surpassing previous baselines.

---

[2]All the reported performance for the baseline methods is from the leaderboard of MME (Fu et al., 2023) and MMBench (Liu et al., 2023c). We report the result of MMICL with InstructBlip-FLANT5-XXL backbone.

| Model | Flickr 30K | WebSRC | VQAv2 | Hateful Memes | VizWiz |
|---|---|---|---|---|---|
| Flamingo-3B (Alayrac et al., 2022) (w/o ICL example) | 60.60 | - | 49.20 | 53.70 | 28.90 |
| Flamingo-3B (Alayrac et al., 2022) (w/ ICL examples (4)) | 72.00 | - | 53.20 | 53.60 | 34.00 |
| Flamingo-9B (Alayrac et al., 2022) (w/o ICL example) | 61.50 | - | 51.80 | 57.00 | 28.80 |
| Flamingo-9B (Alayrac et al., 2022) (w/ ICL examples (4)) | 72.60 | - | 56.30 | 62.70 | 34.90 |
| KOSMOS-1 (Huang et al., 2023b) (w/o ICL example) | 67.10 | 3.80 | 51.00 | 63.90 | 29.20 |
| KOSMOS-1 (Huang et al., 2023b) (w/ ICL examples (4)) | 75.30 | - | 51.80 | - | 35.30 |
| *w/o ICL example* | | | | | |
| BLIP-2 (Li et al., 2023d) (FLANT5-XL) | 64.51 | 12.25 | 58.79 | 60.00 | 25.52 |
| BLIP-2 (Li et al., 2023d) (FLANT5-XXL) | 60.74 | 10.10 | 60.91 | 62.25 | 22.50 |
| InstructBLIP (Dai et al., 2023) (FLANT5-XL) | 77.16 | 10.80 | 36.77 | 58.54 | 32.08 |
| InstructBLIP (Dai et al., 2023) (FLANT5-XXL) | 73.13 | 11.50 | 63.69 | 61.70 | 15.11 |
| *ICL example Evaluation* | | | | | |
| MMICL (FLAN-T5-XL) (w/o ICL example) | 83.47 | 12.55 | 62.17 | 60.28 | 25.04 |
| MMICL (FLAN-T5-XL) (w/ ICL examples (4)) | 83.84 | 12.30 | 62.63 | 60.80 | 50.17 |
| MMICL (FLAN-T5-XXL) (w/o ICL example) | 85.03 | 18.85 | 69.99 | 60.32 | 29.34 |
| MMICL (FLAN-T5-XXL) (w/ ICL examples (4)) | 89.27 | 18.70 | 69.83 | 61.12 | 33.16 |
| MMICL (Instruct-FLAN-T5-XL) (w/o ICL example) | 82.68 | 14.75 | 69.13 | 61.12 | 29.92 |
| MMICL (Instruct-FLAN-T5-XL) (w/ ICL examples (4)) | 88.31 | 14.80 | 69.16 | 61.12 | 33.16 |
| MMICL (Instruct-FLAN-T5-XXL) (w/o ICL example) | 73.97 | 17.05 | 70.30 | 62.23 | 24.45 |
| MMICL (Instruct-FLAN-T5-XXL) (w/ ICL examples (4)) | 88.79 | 19.65 | 70.56 | 64.60 | 50.28 |

Table 4: Main results illustrating the multi-modal in-context learning ability of MMICL across various vision-language tasks[4]. All metrics employed in the evaluations are introduced in Table 25.

| Model | Model Size | Average Performance | Don't Require Visual Infomation | Require Visual Infomation | Performance Gap |
|---|---|---|---|---|---|
| Random Guess | - | 35.50 | 35.80 | 34.90 | - |
| Ying-VLM (Li et al., 2023e) | 13.6B | 55.70 | 66.60 | 44.90 | 21.70 |
| InstructBLIP (Dai et al., 2023) | 12.1B | 71.30 | 82.00 | 60.70 | 21.30 |
| Otter (Li et al., 2023a) | 9B | 63.10 | 70.90 | 55.70 | 15.20 |
| Shikra (Chen et al., 2023a) | 7.2B | 45.80 | 52.90 | 39.30 | 13.60 |
| MMICL | 12.1B | 82.10 | 82.60 | 81.70 | 0.90 |

Table 5: Zero-shot performance of different VLMs on ScienceQA-IMG dataset in different split. MMICL outperforms other VLMs by successfully alleviating language bias.

### 3.3.2 UNDERSTANDING COMPLEX IMAGE-TO-IMAGE RELATIONSHIP

RAVEN (Zhang et al., 2019; Huang et al., 2023a) test is widely used to evaluate the nonverbal reasoning ability of VLMs. Each instance has 3 or 8 images as inputs and 6 candidate images with a unique answer, and the goal is to predict the right image as shown in the right of Fig. 6. It requires visual and logical skills to understand the relationships among images. We

Table 3: Zero-shot generalization on Raven IQ test.

| Model | Accuracy |
|---|---|
| Random Choice | 17 |
| InstructBlip (Dai et al., 2023) | 10.00 |
| Otter (Li et al., 2023a) | 22.00 |
| KOSMOS-1 (Huang et al., 2023a) | 22.00 |
| MMICL (FLAN-T5-XXL) | 34.00 |

conduct zero-shot experiments on the Raven test to evaluate VLM's ability to understand image-to-image relationships. The result in Table 3 shows that MMICL achieves 12 points improvement compared to KOSMOS-1. It indicates that MMICL is able to capture the complex image-to-image relationships and conduct nonverbal visual reasoning tasks.

### 3.4 LEARNING FROM IN-CONTEXT MULTI-MODAL DEMONSTRATIONS

As shown in Table 4, we evaluate the multi-modal in-context learning ability of MMICL across various vision-language tasks. MMICL outperforms other VLMs on both the held-in and held-out datasets and achieves the state-of-the-art few-shot performance. For example, few-shot evaluation (4-shot) of MMICL on the VizWiz benchmark outperforms the baseline Flamingo-9B (Alayrac et al., 2022) and KOSMOS-1 (Huang et al., 2023b) by 15.38 and 14.98 points, respectively. Since VizWiz has never been exposed in the training data, this superior suggests the ability of MMICL to generalize to new tasks with a few exemplars. The few-shot performance of Flickr30K decreases with examples given because the captions examples may provide noise for the VLM to finish the task(i.e., in-context exemplars generally do not provide hints for models to perform image captioning tasks).

---

[4]The anomalous score of InstructBlip on the Vizwiz dataset results from correct outputs not being calculated by the VQA accuracy metric due to the exact match failure.

| Model | VSR | IconQA text | VisDial | IconQA img | Bongard HOI |
|---|---|---|---|---|---|
| | | Stage I | | | |
| Stage I (Blip-2-FLANT5-XL) | 61.62 | 45.44 | 35.43 | 48.42 | 52.75 |
| Stage I (Blip-2-FLANT5-XXL) | 63.18 | 50.08 | 36.48 | 48.42 | 59.20 |
| Stage I (InstructBLIP-FLANT5-XL) | 61.54 | 47.53 | 35.36 | 50.11 | 53.15 |
| Stage I (InstructBLIP-FLANT5-XXL) | 65.06 | 51.39 | 36.09 | 45.10 | 63.35 |
| | | Stage I + Stage II | | | |
| Stage I + Stage II (BLIP-2-FLAN-T5-XL) | 62.85 | 47.23 | 35.76 | 51.24 | 56.95 |
| Stage I + Stage II (BLIP-2-FLAN-T5-XXL) | 64.73 | 50.55 | 37.00 | 34.93 | 68.05 |
| Stage I + Stage II (InstructBLIP-FLAN-T5-XL) | 70.54 | 52.55 | 36.87 | 47.27 | 74.20 |
| Stage I + Stage II (InstructBLIP-FLAN-T5-XXL) | 66.45 | 52.00 | 37.98 | 60.85 | 67.20 |

Table 6: Ablation study on Training Paradigm across five datasets: VSR (Liu et al., 2022), IconQA-text (Lu et al., 2021), VisDial (Das et al., 2017), IconQA-img, and Bongard-HOI (Jiang et al., 2022).

| Model | $MME_{Perception}$ | $MME_{Cognition}$ | Icon-QA | NLVR2 | Raven | Winoground |
|---|---|---|---|---|---|---|
| - w/o context scheme | 1238.99 | 316.79 | 52.80 | 56.65 | 8.00 | 6.00 |
| - w/o image declaration | 1170.87 | 341.07 | 47.15 | 61.00 | 18.00 | 3.00 |
| - w/o in-context format | 1141.02 | 345.36 | 51.95 | 62.63 | 28.00 | 20.00 |
| - w/o interrelated images | 1207.70 | 333.21 | 54.35 | 59.60 | 16.00 | 25.75 |
| MMICL | 1303.59 | 370.71 | 58.12 | 72.45 | 32.00 | 38.75 |

Table 7: Ablation study on Context Scheme across different tasks. We only report the overall perception and cognition score of MME and the Group score of Winoground.

### 3.5 HALLUCINATION AND LANGUAGE BIAS OF VLMS

Current VLMs have significant visual hallucinations (Li et al., 2023f), preventing VLMs from benefiting from multi-modal ICL. Especially when dealing with complex prompts with multiple images (e.g., multi-modal chain of thoughts (Zhang et al., 2023b)), VLMs often overlook visual content when facing extensive text. This language bias reduces their efficiency in answering questions that require both images and text. ScienceQA-IMG (Lu et al., 2022) is a challenging task that requires a model to use both modalities to answer the question. We manually split the dataset into two groups: questions needing images to answer and those not. Details are presented in Appendix Q. Extensive experiments in Table 5 demonstrate that MMICL effectively mitigates language bias as it performs equally well in both groups. We also examined object hallucination in MMICL in Appendix L, which shows impressive performance.

### 3.6 ABLATION STUDY

**Ablation Study on Training Paradigm:** We conduct an ablation study on various tasks to evaluate the effect of multi-modal in-context tuning. Table 6 displays a significant enhancement due to multi-modal in-context tuning. It can be observed across all types and sizes of models, especially for tasks that involve multiple images. This indicates that with the help of Stage II, MMICL can handle complex multi-modal prompts and accomplish challenging tasks with multiple images. Result in Appendix K also confirms this point with the outstanding performance of MMICL across video datasets.

**Ablation Study on Context Scheme:** We conducted ablation experiments using the InstructBLIP-FLANT5-XL as the backbone model in various settings to verify the effectiveness of the context scheme, aiming at pinpointing the exact source driving improvements in certain capabilities of our model. As shown in Table 7, the superiority of MMICL is driven by the collective impact of our design elements—removing any component cannot guarantee the superior performance of our model. Each component of our design significantly contributes to different aspects of our model. Details and further analysis are available in Appendix N.

## 4 CONCLUSION

In this paper, we highlight the limitations of VLMs handling the complex multi-modal prompts with multiple images, which makes VLMs less effective in downstream vision-language tasks. We introduce MMICL to address the aforementioned limitations and take our model a step further by extending it to multi-modal in-context learning. This breakthrough enables VLMs to better understand complex multi-modal prompts. Furthermore, MMICL sets a new state-of-the-art performance on the general VLM benchmarks and complex multi-modal reasoning benchmarks.

## 5 ACKNOWLEDGEMENT

We extend our heartfelt gratitude to the anonymous reviewers whose dedication and insightful feedback have significantly enhanced the caliber of this paper. Their constructive critiques and valuable suggestions were instrumental in refining our work. Additionally, we are deeply appreciative of the Program Chairs and Area Chairs for their meticulous handling of our submission and for their comprehensive and invaluable feedback. Their guidance has been pivotal in elevating the quality of our research.

This work is supported by the National Science Foundation of China under Grant No.61936012 and 61876004.

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

# A  RELATED WORK

## A.1  VISION-LANGUAGE PRETRAINING

| Model | Multi-Image Inputs | Multi-modal Instruction Tuning | Text-to-Image Reference |
| --- | --- | --- | --- |
| Flamingo | ✓ | ✗ | ✗ |
| Meta learner | ✓ | ✗ | ✗ |
| BLIP-2 | ✗ | ✗ | ✗ |
| LLAVA | ✗ | ✓ | ✗ |
| MiniGPT-4 | ✗ | ✓ | ✗ |
| InstructBLIP | ✗ | ✓ | ✗ |
| Shikra | ✗ | ✓ | ✓ |
| Kosmos-1 | ✓ | ✗ | ✗ |
| Otter | ✓ | ✓ | ✗ |
| MMICL | ✓ | ✓ | ✓ |

Table 8: Summary of Vision-Language Pre-Trained Models.

Our work is inspired by recent vision-language pre-training works (Zhu et al., 2023; Liu et al., 2023b; Li et al., 2022; 2023d), which have been proven effective for aligning visual inputs and frozen LLMs to obtain cross-modal generalization ability.

**BLIP-2**  BLIP-2 (Li et al., 2023d) bridges the modality gap with a lightweight Querying Transformer, which is pre-trained in two stages. The first stage bootstraps vision-language representation learning from a frozen image encoder. The second stage bootstraps vision-to-language generative learning from a frozen language model.

**InstructBLIP**  InstructBLIP (Dai et al., 2023) performs vision-language instruction tuning based on the pre-trained BLIP-2 models with converted multi-modal datasets and the LLaVA (Liu et al., 2023b) dataset generated by GPT-4.

**MiniGPT-4**  MiniGPT-4 (Zhu et al., 2023)aligns a CLIP visual encoder with a frozen Vincuna (Chiang et al., 2023) with an artificially collected dialog dataset

**Shikra**  Shikra (Chen et al., 2023a), a VLM which can handle spatial coordinate inputs and outputs in natural language. It makes Shikra excel at referential dialogue and general vision-language tasks, resulting in outstanding performance.

However, there is still less work focusing on VLMs with multi-image inputs.

**Flamingo**  Flamingo (Tsimpoukelli et al., 2021) achieves multi-visual inputs based on self-attention for images but performs poorly in downstream tasks. Flamingo supports Few-Shot Learning (FSL) in VLM via ICL by leveraging its robust capability to handle multi-visual inputs and uses cross-attention instead of self-attention to get better performance. However, it still suffers from the unableness to explicitly point images, so they introduce a hacky cross-attention mask.

**Kosmos-1**  Kosmos-1 (Huang et al., 2023a), is trained from scratch on billion-scale multi-modal corpora, including interleaved text-image web page data, image-text caption, and language-only instruction tuning data. It can multi-modal Few-Shot Learning and Chain-of-Thought processes, thereby achieving formidable performance.

**Otter**  Otter (Li et al., 2023a), an open-source implementation of flamingo and trained with multi-modal instruction in-context tuning data.

**Meta learner**  Najdenkoska et al. (2023) uses meta-learning objective to train an adapter that aggregates multiple image features so the original VLM and adapter become a better few-shot learner.

Recent VLMs (Zhu et al., 2023; Liu et al., 2023b; Li et al., 2022; Alayrac et al., 2022; Dai et al., 2023; Chen et al., 2024b) have been proven effective for aligning visual inputs and frozen LLMs to obtain

cross-modal generalization ability. However, previous works overlooked multi-image VLMs, mainly focusing on handling single-image prompts. Tsimpoukelli et al. (2021) supports multi-image inputs using self-attention for images but performs poorly in downstream tasks. Although Flamingo (Alayrac et al., 2022) supports Few-Shot Learning in VLMs and uses cross-attention to capture text-image relationships, it still suffers from exact reference to specific images.

## A.2 MULTI-MODAL INSTRUCTION TUNING

Pre-trained language models have achieved great success in both natural language understanding (Si et al., 2022b; 2023a; An et al., 2023; Hu et al., 2023a; Chen et al., 2023b) and natural language generation (Si et al., 2022a; 2023b; Cai et al., 2023; Liu et al., 2023d). Recently, instruction tuning (Kung & Peng, 2023; Li et al., 2024; Wei et al., 2022) has attracted more and more attention in order to enable LLMs to follow natural language instructions and complete real-world tasks. However, multi-modal instruction tuning still requires further exploration. Multiinstruct (Xu et al., 2023) introduces instruction tuning to enhance the performance of VLMs in instruction-following ability. Due to the architectural design, Multiinstruct still struggles with complex contexts containing multiple images. Otter (Li et al., 2023a) fine-tunes Openflamingo (Awadalla et al., 2023) to augment its instruction comprehension capabilities. However, Otter's dataset lacks text-to-image references and interconnected image-to-image data. This limitation hinders its capability to handle complex contexts that involve visual-textual relationships.

## A.3 IN-CONTEXT LEARNING

It has been well-explored to enable ICL in pre-trained language models (PLM). MetaICL (Min et al., 2021) proposes a meta-training framework for few-shot learning to tune a PLM to do in-context learning on a large set of training tasks. LM-BFF (Gao et al., 2020) studies few-shot fine-tuning of PLMs. However, ICL in VLM is still less explored. Recent works in VLM mainly focus on zero-shot evaluation with single image input.

## B MULTI-MODAL ICL DATA

We construct two training datasets, text-image interleaved data and in-context learning data, for the text-image relationship challenge and image-image relationship challenge, respectively. In this section, we will cover the data resources.

## C DATA RESOURCE

The data resource used in constructing the MIC dataset is displayed in Fig. 7. Our training dataset comes from 8 task categories and 16 datasets.

*Image Captioning* aims to produce descriptions of the given images according to different needs. Our training dataset includes MS COCO (Lin et al., 2014), DiffusionDB (Wang et al., 2022b), and Flickr 30K (Young et al., 2014).

*Knowledgeable Visual Question Answering (KVQA)* requires the model to make use of commonsense knowledge outside the input image to answer questions. Our training dataset includes OK-VQA (Marino et al., 2019).

*Image Question Answering (IQA)* requires the model to answer the questions based on the image correctly. Our training dataset includes VQAv2 (Goyal et al., 2017), ST-VQA (Biten et al., 2019), Text-VQA (Singh et al., 2019), WikiART (Saleh & Elgammal, 2015) and RefCOCO (Yu et al., 2016).

*Video Question Answering (VideoQA)* requires the model to answer questions based on the video correctly. We extract eight frames per video as visual inputs for Video QA tasks. Our training dataset includes MSRVTTQA (Xu et al., 2016).

*Video Captioning* requires the model to give the caption based on the video. We extract eight frames per video as visual inputs for Video Captioning tasks. Our training dataset includes MSRVTT (Xu et al., 2016).

| Task | Dataset | Used | #samples Train | Val | Test | License |
|------|---------|------|-------|-----|------|---------|
| Captioning | MS COCO (Lin et al., 2014) | Yes | 413,952 | 202,496 | 0 | Custom |
| | DiffusionDB (Wang et al., 2022b) | Yes | 19,963 | 0 | 0 | MIT |
| | Flickr (Young et al., 2014) | Yes | 144,896 | 4,864 | 4,864 | Custom |
| | NoCaps (Agrawal et al., 2019) | Yes | 0 | 45,000 | 0 | CC BY 2.0 |
| Classification | MiniImage (Russakovsky et al., 2015) | Yes | 38,400 | 9,600 | 12,000 | CC0 1.0 |
| VQA | VQA v2 (Goyal et al., 2017) | Yes | 592,998 | 26,173 | 25,747 | CC BY 4.0 |
| | ST-VQA (Biten et al., 2019) | Yes | 78,222 | 0 | 4,070 | CC BY 4.0 |
| | Text-VQA (Singh et al., 2019) | Yes | 34,602 | 0 | 0 | CC BY 4.0 |
| | NLVR2 (Suhr et al., 2018) | Yes | 86,373 | 6,982 | 6,967 | CC BY 4.0 |
| | RefCOCO (Yu et al., 2016) | Yes | 141,968 | 0 | 0 | Custom |
| KVQA | OK-VQA (Marino et al., 2019) | Yes | 9,009 | 5,046 | 0 | CC BY 4.0 |
| Reasoning | GQA (Hudson & Manning, 2019) | Yes | 943,000 | 132,062 | 12,578 | CC BY 4.0 |
| | VCR (Zellers et al., 2019) | Yes | 118,156 | 14,472 | 5,000 | Custom |
| | Winoground (Thrush et al., 2022a) | No | 0 | 0 | 400 | Custom |
| video | MSRVTT-QA (Xu et al., 2016) | Yes | 158,581 | 12,278 | 72,821 | MIT |
| | MSRVTT (Xu et al., 2016) | Yes | 130,260 | 9,940 | 59,800 | MIT |
| Others | WikiART (Saleh & Elgammal, 2015) | Yes | 13,000 | 5,500 | 0 | Custom |
| | LLAVA-Instruct-150K (Liu et al., 2023b) | Yes | 236,564 | 0 | 0 | CC BY 4.0 |

Table 9: Detailed task descriptions and statistics of our instruction tuning tasks, including all datasets in all types of tasks. The column "Used" indicates whether we use this dataset in the multi-modal in-context tuning stage.

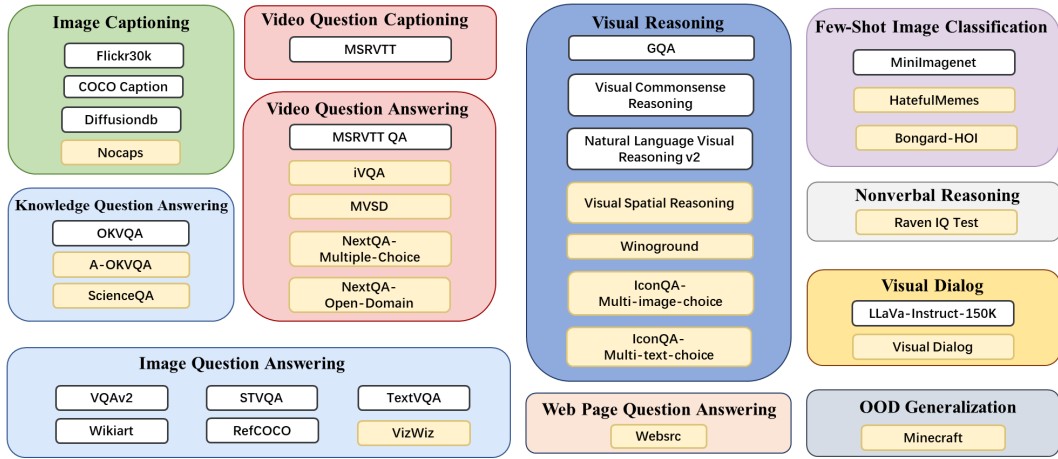

Figure 7: Illustration of the data resource used to construct MIC dataset. It consists of 11 tasks and 33 different datasets. The held-in datasets are indicated by white and the held-out datasets are indicated by yellow.

*Visual Reasoning* requires the model to correctly perform image reasoning and answer questions. Our training dataset includes GQA (Hudson & Manning, 2019), VCR (Zellers et al., 2019), and NLVR2 (Suhr et al., 2018).

*Image Classification* involves classifying an image based on a given set of candidate labels. Our training dataset includes MiniImage (Russakovsky et al., 2015).

*Visual Dialog* requires the model to hold a meaningful dialog about visual content with humans in natural, conversational language. Our training dataset includes LLAVA-Instruct-150K (Liu et al., 2023b).

Our testing dataset comes from 10 task categories and 18 datasets.

*Image Captioning* includes the Nocaps (Agrawal et al., 2019) dataset.

*Knowledgeable Visual Question Answering (KVQA)* includes the ScienceQA (Lu et al., 2022) and A-OKVQA (Schwenk et al., 2022) datasets.

*Image Question Answering (IQA)* includes the VizWiz (Bigham et al., 2010) dataset.

*Visual Reasoning* includes the Winoground (Thrush et al., 2022b), VSR (Liu et al., 2022) and IconQA (Lu et al., 2021) dataset. Winoground proposes a task of matching two given images and two captions correctly. The challenge of this task is that both captions contain a completely identical set of words, only in a different order. VSR describes the spatial relation of two individual objects in the image, and a VLM needs to judge whether the caption correctly describes the image (True) or not (False). The IconQA dataset has two sub-datasets: image question answering with multiple text choice and image question answering with multiple image choice.

*Web Page Question Answering (Web QA)* includes the Websrc (Chen et al., 2021a; Huang et al., 2023a) datasets. The model must answer questions based on the web image and the optional extracted texts. We sampled 2000 instances from Websrc for the evaluation. To align with KOSMOS-1 (Huang et al., 2023a), we only use the web image as input.

*Video Question Answering (VideoQA)* includes the iVQA (Yang et al., 2021), MVSD (Chen & Dolan, 2011), and NextQA (Xiao et al., 2021) dateset. The NextQA dataset has two sub-datasets: video question answering with multiple choice and open-domain video question answering.

*Few-shot Image Classification* includes the HatefulMemes (Kiela et al., 2020) and Bonard-HOI (Jiang et al., 2022) dataset. HatefulMemes requires the model to determine if a meme is hateful based on the image and explanation provided. Bonard-HOI is the benchmark for evaluating the model's ability in Few-Shot Visual Reasoning for Human-Object Interactions. It provides few-shot examples with challenging negatives, where positive and negative images only differ in action labels. The model is then asked whether the final image is positive or negative. We sampled 2000 instances from Bonard-HOI for the evaluation.

*Nonverbal Reasoning* includes the Raven IQ test (Huang et al., 2023a). Each instance in the Raven IQ test has 3 or 8 images as inputs and six candidate images with a unique correct completion, and the goal is to predict the next image from the candidates.

*Visual Dialog* includes the visual dialog dataset (Das et al., 2017). We use the question of the final dialogue as the question for instance and take all preceding dialogues as the context to perform open-domain image question answering.

*OOD Generalization* includes the Minecraft dataset that we construct using Minecraft (Cipollone et al., 2014) game which requires the VLM to identify whether an animal (i.e., cow, llama, chicken, donkey, and so on) is present in a picture.

More detailed task descriptions and statistics about the datasets are shown in Table 9.

## D  DATA CONSTRUCTION

Firstly, we create an image declaration per instance in all datasets to generate datasets with explicit text-to-image reference. We then have annotators scrutinize every dataset's samples and provide task instructions. This practice aids in gaining a comprehensive understanding of the task and helps craft high-quality templates.

Next, we employ ChatGPT[5] to rewrite the instructions to describe the key characteristics of each task accurately. After ChatGPT generates the instructions, we undergo a manual review to guarantee the high quality of the instructions.

We select ten suitable templates matching as candidates, then merge the original dataset's input into a randomly chosen template. We assemble demonstrations for each instance from the dataset by selecting a small amount of data and arranging them sequentially. These demonstrations are integrated with the input instance to generate multi-modal contextual data[6].

---

[5]We use the *gpt-3.5-turbo* version of ChatGPT.
[6]Except for the video datasets, vcr dataset, and LLaVa dataset.

We construct multi-image data by extracting eight frames per video from MSRVTT (Xu et al., 2016) and MSRVTTQA (Xu et al., 2016) datasets. We also crop images from the VCR (Zellers et al., 2019) dataset using object bounding boxes to produce intertwined multi-modal data with closely related images. We convert all data into a vision-language Q&A format to create high-quality multi-modal training data and accumulate 5.8M samples in MIC dataset. Due to resource constraints, we use approximately 10% of the data with the sampling strategy described in Appendix F to finetune MMICL. It is anticipated that a larger model trained on all of our data would yield a more promising result.

---

**Algorithm 1** Image Declaration

---

**Require:** Interleaved multi-modal input: $X$, containing visual embedding: $V = \{v_1, v_2, \ldots\}$ and text embedding $H = \{h_1, h_2, \ldots\}$, where $v_i$ represents the image embedding and $h_i$ represents the span between the image embeddings.

**Ensure:** Interleaved multi-modal input with image declaration: $\hat{X}$
1: **for** each interleaved multi-modal input $X$ **do**
2:     $n \leftarrow$ number of images in $X$
3:     Initialize image proxy tokens $[IMG_1], [IMG_2], \ldots$
4:     **for** each image $i$ in $X$ **do**
5:         $Ref_i \leftarrow$ "image $i$ is $[IMG_i]v_i$" or "image $i$: $[IMG_i]\,v_i$"
6:     **end for**
7:     $R \leftarrow \{Ref_1, Ref_2, \ldots\}$
8:     Replace $v_i$ in $X$ with $Ref_i$: $\hat{X} = [Ref_1, h_1, Ref_2, h_2, \ldots]$
9: **end for**

---

### D.1    MULTI-MODAL DATA WITH INTERCONNECTED IMAGES

The data construction framework for Multi-modal Data with Interconnected Images as Sec. 2.2.2 is developed through an automated process that leverages existing dataset annotations (i.e. Visual Commonsense Reasoning dataset (VCR) (Zellers et al., 2019)). This approach obviates the necessity for supplementary manual annotation efforts.

The original annotations of VCR dataset comprise raw images, bounding boxes, and a set of questions and answers that specifically reference these bounding boxes as the left part of Fig. 8. This structured annotation approach facilitates a comprehensive understanding of the image context. Leveraging the existing annotations, we develop an automated workflow, including extracting sub-images from raw images based on bounding boxes and employing ChatGPT to formulate a variety of instructions. Initially, we manually craft a set of instruction templates, which, alongside the dataset description, are inputted into ChatGPT. Subsequently, ChatGPT is asked to expand the templates, thereby yielding a diverse set of instruction templates.

### D.2    UNIFIED MULTI-MODAL IN-CONTEXT FORMAT FOR DIFFERENT TASKS

The data construction framework for the multi-modal in-context format data as Sec. 2.2.3 is developed through an automated process that leverages existing dataset annotations. The used dataset are presented as Appendix F.

Through selective sampling within the dataset, we have formulated in-context examples for each respective task. These examples encompass a range of elements, including images, questions, and answers. ChatGPT is used to formulate a variety of instructions. The formulated instruction templates are presented in Appendix G.

### D.3    MULTI-MODAL DATA WITH VIDEO FRAMES

The data construction framework for Video Data is developed through an automated process that leverages existing dataset annotations. The used dataset are MSRVTT (Xu et al., 2016) and MSRVTTQA (Xu et al., 2016) datasets as Appendix F.

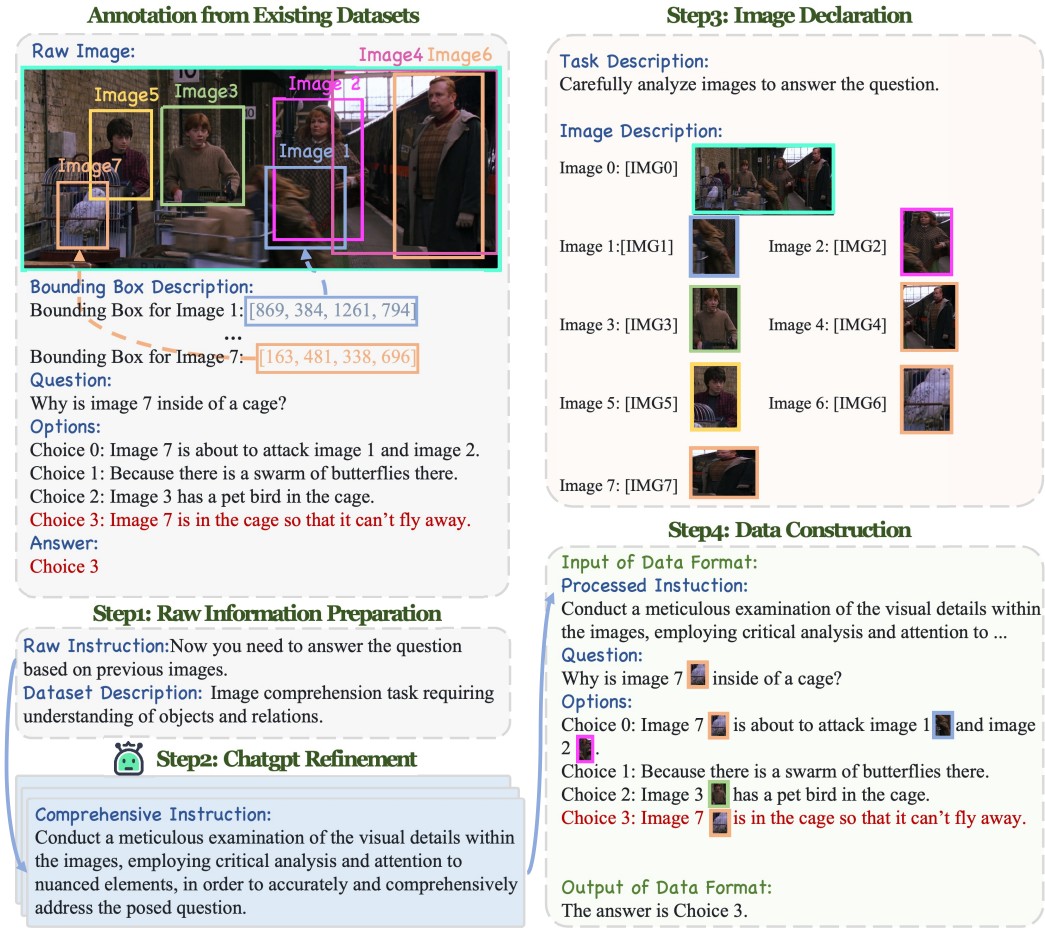

Figure 8: Illustration of automatic data construction pipeline for multi-model data with interconnected images. The automatic construction is based on existing annotation of VCR dataset (Zellers et al., 2019) without human involvement. ChatGPT is used for instruction refinement.

## E MODEL STRUCTURE

As shown in Fig. 11, MMICL treats the image and language representations equally and combines them into interleaved image-text representations, similar to the original input. Each given image is encoded by a vision encoder (e.g., ViT (Radford et al., 2021; Fang et al., 2023)) to get the vision representation of the image. Then, we use the Q-former as the VPG to extract the visual embedding. We utilize a fully connected layer as the projection layer to convert each visual embedding to the same dimension as the text embedding of the LLMs. This alignment helps the LLM to understand the images. Our approach treats the visual and text embedding equally, enabling a flexible combination of visual and textual content. Finally, we combine the visual embeddings of multiple images with text embeddings in an interleaved style and then feed them into the LLM. We set the weights for mapping query and value vectors in the attention layer of LLM as learnable to better adapt to the multi-modal context with multiple images. During the pre-training, we freeze the image encoder, Q-former, and the backbone LLM while jointly training the language projection and the query and value vectors of the LLM.

## F DATA BALANCE

Previous studies have shown that the data balance of training data could significantly influence the model performance (Dai et al., 2023). Mixing the training data of each dataset uniformly could

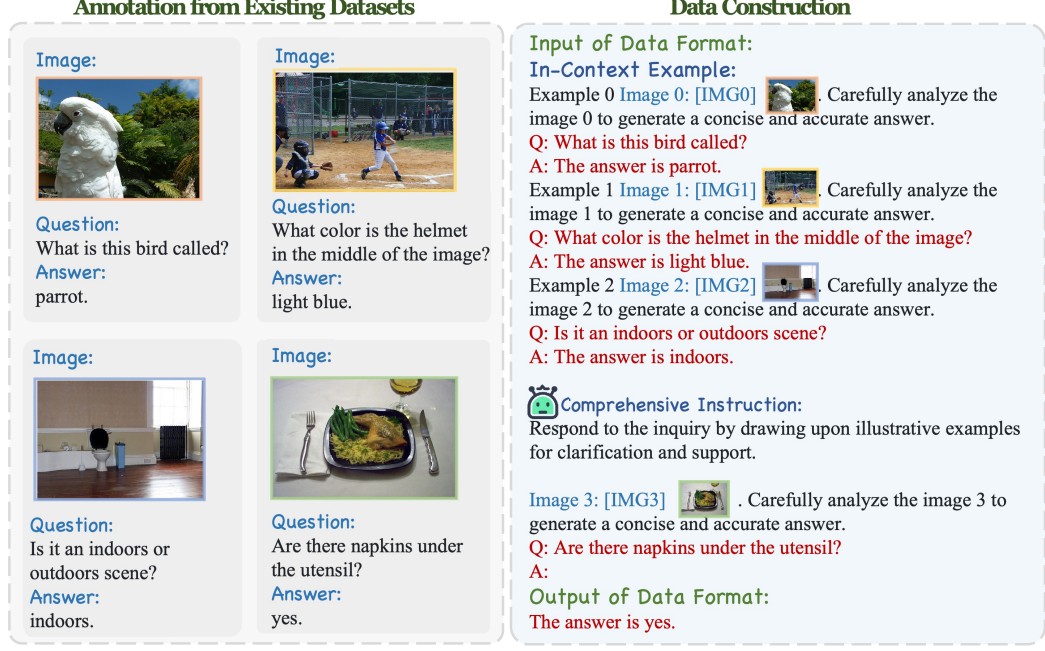

Figure 9: Illustration of automatic data construction pipeline for multi-model data with in-context examples. The automatic construction is based on existing annotation without human involvement. ChatGPT is used for instruction refinement. Data source can be found at Appendix F. The formulated instruction templates are presented in Appendix G.

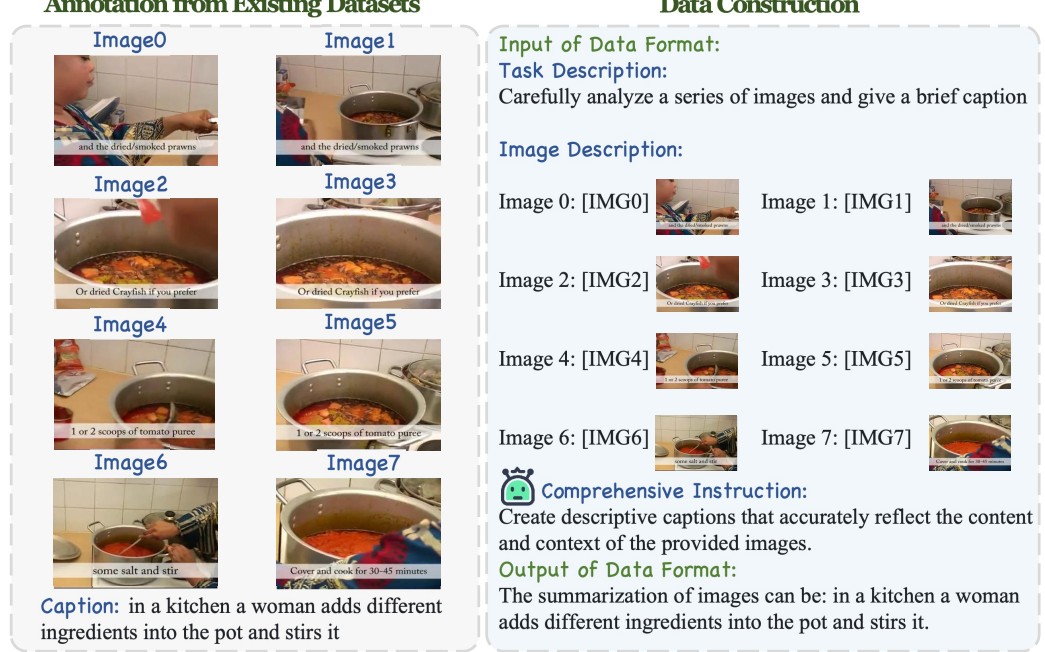

Figure 10: Illustration of automatic data construction pipeline for multi-model data with video. The automatic construction is based on existing annotation without human involvement. ChatGPT is used for instruction refinement.

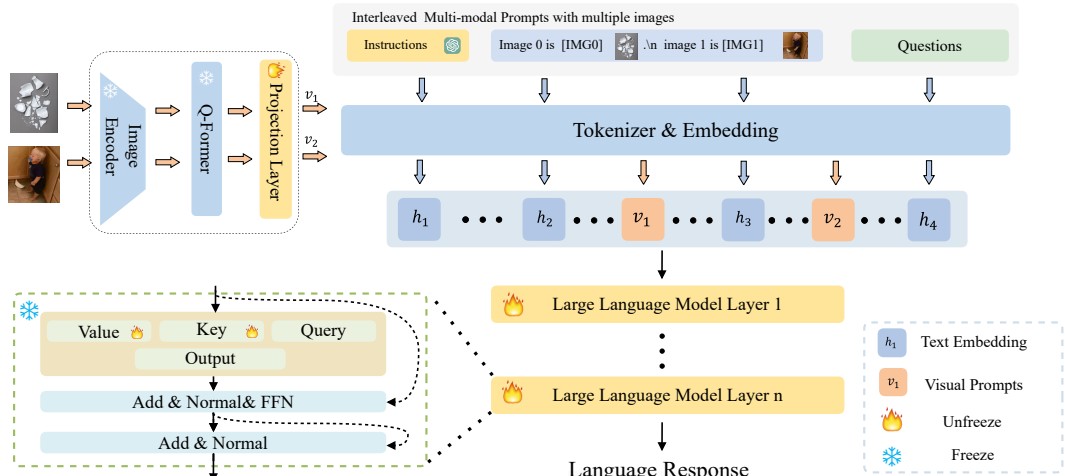

Figure 11: Illustration of the MMICL structure.

cause the model to overfit smaller datasets and underfit larger datasets, causing poor performance. In order to alleviate this problem, we employ a sampling strategy to sample datasets with probabilities proportional to the square root of the number of training samples following Dai et al. (2023). Formally, given $D$ datasets with $N$. training samples $\{N_1, N_2, \ldots, N_D\}$, the probability $p_d$ of data samples being selected from a dataset during training is as follows.

$$p_d = \frac{\sqrt{N_d}}{\sum_{i=1}^{D} \sqrt{N_i}} \tag{4}$$

## G    INSTRUCTION TEMPLATE FOR DATA CONSTRUCTION

As Sec. 2.2.3, the constructions of MIC require carefully designed templates. The instruction templates for each task are presented in this section. The templates for tasks MSCOCO, Flick30k, Nocaps, and Diffusiondb are presented in Table 10. The templates for tasks MiniImagenet are presented in Table 11. The templates for tasks VQAv2, S-VQA, WikiART, and RefCOCO are presented in Table 13. The templates for task OKVQA are presented in Table 14. The templates for task MSRVTT are presented in Table 15. The templates for tasks MSRVTTQA and MSVD are presented in Table 16.

## H    EXPERIMENT DETAILS

Following Chung et al. (2022), we use FLANT5-XL and FLANT5-XXL (Chung et al., 2022) as the backbone LLMs. In Stage I, we set the vision encoder and language model to be frozen and utilize the images from COCO(Lin et al., 2014), CC3M(Sharma et al., 2018), Visual Genome(Krishna et al., 2017), CC12M(Changpinyo et al., 2021), SBU(Ordonez et al., 2011) and LAION-400M data(Schuhmann et al., 2021), as well as the captions generate by $BLIP_{large}$ (Li et al., 2022) to perform feature alignment training on the Q-former. We keep the other part of the VLM frozen and jointly train the Q-former and projection layer. The Q-former's visual embedding output occurs in 32-token size of the backbone language model. To benefit from BLIP-2's significant visual representation extraction ability, we integrate its powerful vision encoder to initialize the Q-former and projection layer. [7]. In Stage II, we train the model for three epochs with a lower learning rate of $1e-5$. The weights of mapping query and value vectors in the attention layer of LLMs are learnable in this stage to better adapt to the multi-modal prompts with multiple images. In this stage, we freeze the visual encoder, Q-former, and the backbone LLM and jointly train the projection layer, the query vectors, and the value vectors of the LLM.

---

[7]In practice, we use checkpoint of BLIP-2 and InstructBlip as the backbone for MMICL, so Stage I is skipped.

| **Templates of Image Captioning (MSCOCO, Flick30k, Nocaps, Diffusiondb)** |
|---|
| (1) Carefully analyze image 0: [IMG0] {image} to generate a concise and accurate description that accurately represents the objects, people, and scenery present.
(2) Use clear and concise language that accurately describes the content of image 0: [IMG0] {image}.
(3) Your caption should provide sufficient information about image 0: [IMG0] {image} so that someone who has not seen the image can understand it.
(4) image 0 is [IMG0] {image}. Be specific and detailed in your description of image 0, but also try to capture the essence of image 0 in a succinct way.
(5) image 0 is [IMG0] {image}. Based on the image 0, describe what is contained in this photo. Your caption should be no more than a few sentences and should be grammatically correct and free of spelling errors.
(6) Include information in your caption that is specific to image 0: [IMG0] {image} and avoid using generic or ambiguous descriptions.
(7) image 0 is [IMG0] {image}. Based on the image 0, give a caption about this image. Think about what message or story image 0 is conveying, and try to capture that in your image caption.
(8) Based on the image 0, give a caption about this image. Your caption should provide enough detail about image 0: [IMG0] {image} to give the viewer a sense of what is happening in the image.
(9) Give a caption about this image. Avoid using overly complex language or jargon in your caption of image 0: [IMG0] {image} that might confuse the viewer.
(10) Be creative in your approach to captioning image 0: [IMG0] {image} and try to convey a unique perspective or story. |

Table 10: Instruction templates used for transforming datasets into instruction tuning data. (I) {image} denotes image embedding encoded by image encoder, image embedding will be concatenated with language embedding as input. <image $j$> denotes image token to exact reference the $j$-th image in an instance as described in Sec. 2.2.1.

| **Templates of Image Classification (MiniImagenet, etc)** |
|---|
| (1) image 0 is [IMG0] {image}. Please identify the object or concept depicted in image 0.
(2) image 0 is [IMG0] {image}. What is the main subject of image 0?
(3) image 0 is [IMG0] {image}. Can you recognize and label the object shown in image 0?
(4) image 0 is [IMG0] {image}. Identify the category or class to which image 0 belongs.
(5) image 0 is [IMG0] {image}. Based on the visual content, determine what image 0 represents.
(6) image 0 is [IMG0] {image}. What is the name or label of the item captured in image 0?
(7) image 0 is [IMG0] {image}. Please provide a description or identification of the subject in image 0.
(8) image 0 is [IMG0] {image}. From the visual cues, determine the object or entity depicted in image 0.
(9) image 0 is [IMG0] {image}. Can you recognize and name the primary element shown in image 0?
(10) image 0 is [IMG0] {image}. Identify the object or concept that best describes what is depicted in image 0. |

Table 11: Instruction templates used for transforming datasets into instruction tuning data. (I) {image} denotes image embedding encoded by image encoder, image embedding will be concatenated with language embedding as input. <image $j$> denotes image token to exact reference the $j$-th image in an instance as described in Sec. 2.2.1.

| **Templates of Knowledge Visual Question Answering (OK-VAQ)** |
|---|
| (1) Look at image 0 labeled [IMG0] {image} carefully and read question: question. Try to understand what is being asked before selecting an answer.
(2) image 0 is [IMG0] {image}. Consider all of the information in image 0 labeled [IMG0] when answering question. Look at objects, colors, shapes, and other details that may be relevant to question: question Answer:
(3) image 0 is [IMG0] {image}. Read each answer choice carefully and answers question : question based on the information provided in image 0.
(4) image 0 is [IMG0] {image}. Given the picture [IMG0], pay attention to the wording of question and answer the following question: question Answer:
(5) Read the question carefully and look at image 0 labeled [IMG0] {image}. Use your intuition and common sense when answering the question: question
(6) Consider all of the information in image 0 labeled [IMG0] {image} when answering the question: question
(7) Take your time when answering each question. Don't rush through the questions, and make sure you have carefully considered all of the information provided in image 0 labeled [IMG0] {image} and the question before making your selection. Question: question Answer:
(8) Make sure your answers are based on the information presented in the image 0: [IMG0] {image}. Question:question Answer:
(9) Carefully examine image 0 labeled [IMG0] {image} before answering the question. Question:question Answer:
(10) Please refer to image 0: [IMG0] {image} when answering the following questions: question Answer: |

Table 12: Instruction templates for task OKVQA.

| Templates of Image Question Answering (VQAv2, ST-VQA, WikiART, RefCOCO, etc) |
| --- |

**VQAv2**

(1) image 0 is [IMG0] {image}. For the question, carefully examine the image and use your knowledge to determine the correct answer. Question: question Answer:

(2) image 0 is [IMG0] {image}. Given the picture [IMG0], pay attention to the wording of question and answer the following question: question Answer:

(3) Read the question carefully and look at image 0 labeled [IMG0] {image}. Use your intuition and common sense when answering the question: question

(4) Answer each question based on the information presented in image 0: [IMG0] {image}. Given the picture [IMG0], what is the answer to the question: question Answer:

(5) Please refer to image 0: [IMG0] {image} when answering the following questions: question Answer:

(6) Questions is related to image 0: [IMG0] {image}. Please analyze the image and provide the correct answer for the question: question

(7) Read the question carefully and look at image 0 labeled [IMG0] {image}. Use your intuition and common sense when answering the question: question

(8) Consider all of the information in image 0 labeled [IMG0] {image} when answering the question: question

(9) Take your time when answering each question. Don't rush through the questions, and make sure you have carefully considered all of the information provided in image 0 labeled [IMG0] {image} and the question before making your selection. Question: question Answer:

(10) Use the image 0: [IMG0] {image} as a visual aid to help you answer the questions accurately. Question:question Answer:

**ST-VQA**

(1) Answer each question based on the information presented in image 0: [IMG0] {image}. Given the picture [IMG0], what is the answer to the question: question Answer:

(2) Please refer to image 0: [IMG0] {image} when answering the following questions: question Answer:

(3) Questions is related to image 0: [IMG0] {image}. Please analyze the image and provide the correct answer for the question: question

(4) For each question, use the image 0: [IMG0] {image} as a reference to answer the question: question

(5) Make sure your answers are based on the information presented in the image 0: [IMG0] {image}, and any OCR text associated with it. Question:question Answer:

(6) Answer the question as accurately as possible using the information provided in the image 0: [IMG0] {image}, and any OCR text associated with it. Question:question Answer:

(7) Please ensure that you are answering the question based on the information presented in the image 0: [IMG0] {image}.Question:question Answer:

(8) The image 0: [IMG0] {image} is the primary source of information for answering the questions. Please refer to it carefully when answering question: question Answer:

(9) Pay close attention to the details in image 0: [IMG0] {image}, as they may provide important information for answering the questions. Question:question Answer:

(10) Use the image 0: [IMG0] {image} as a visual aid to help you understand the context and answer the questions accurately. Question:question Answer:

**WikiART**

(1) image 0 is [IMG0] {image}. Please provide information about the artist, genre, and style of this artwork.

(2) image 0 is [IMG0] {image}. I would like to know the artist's name, the genre, and the specific style depicted in this painting.

(3) image 0 is [IMG0] {image}. Could you identify the artistic genre, the artist, and the style portrayed in this artwork?

(4) image 0 is [IMG0] {image}. In this painting, which genre does it belong to, who is the artist, and what is the predominant style?

(5) image 0 is [IMG0] {image}. Tell me about the artist, genre, and style associated with this particular artwork.

(6) image 0 is [IMG0] {image}. This piece of art seems intriguing. Can you provide details about the genre, the artist, and the style it represents?

(7) image 0 is [IMG0] {image}. Identify the genre, artist, and style of this captivating artwork, please.

(8) image 0 is [IMG0] {image}. I'm curious to learn about the artist's name, the genre, and the distinctive style showcased in this artwork.

(9) image 0 is [IMG0] {image}. Could you enlighten me about the genre, artist, and the artistic style that characterizes this beautiful piece?

(10) image 0 is [IMG0] {image}. In terms of genre, artist, and style, what information can you provide regarding this fascinating artwork?

**RefCOCO**

(1) image 0 is [IMG0] {image}.Given image 0, create a descriptive caption that accurately represents the content of the image, including the item located in the {quadrant} of the image.

(2) Use your knowledge of the image 0 and the {quadrant} location to generate a detailed and accurate caption that captures the essence of the scene. Keep in mind that image 0 is [IMG0] {image}.

(3) image 0 is [IMG0] {image}. When writing your caption, be sure to include specific details about the item located in the {quadrant} of the image 0, such as its size, shape, color, and position.

(4) Think about the intended audience for your caption and use appropriate language and tone. Consider the context of the image: [IMG0] {image} and the {quadrant} location when creating your caption, and make sure that it accurately reflects the content of the image.

(5) Your caption should be concise and to the point, while still capturing the essence of the image 0 and the item located in the {quadrant} of the image. Avoid including irrelevant information in your caption that detracts from the main content of the image. Remember that image 0 is [IMG0] {image}.

(6) image 0 is [IMG0] {image}. Check your caption for accuracy and grammatical errors before submitting. Be creative in your approach to captioning the image and the item located in the {quadrant}.

(7) image 0 is [IMG0] {image}. Given image 0, describe the item in the {quadrant} of the image.

(8) image 0 is [IMG0] {image}. Using image 0, provide a caption for the object located in the {quadrant} of the image.

(9) For image 0: [IMG0] {image}, describe the object in the {quadrant} of the image.

(10) Given the image 0: [IMG0] {image}. Generate a description for the item located in the {quadrant} of the image.

(11) image 0 is [IMG0] {image}. Using the provided image 0, describe the object located in the {quadrant} of the image.

Table 13: Instruction templates for tasks VQAv2, ST-VQA, WikiART, and RefCOCO.

**Templates of Video Question Captioning (MSRVTT)**

(1) image 0 is [IMG0] {image}. image 1 is [IMG1] {image}. image 2 is [IMG2] {image}. image 3 is [IMG3] {image}. image 4 is [IMG4] {image}. image 5 is [IMG5] {image}. image 6 is [IMG6] {image}. image 7 is [IMG7] {image}. Watch the images carefully and write a detailed description of what you see.

(2) image 0 is [IMG0] {image}. image 1 is [IMG1] {image}. image 2 is [IMG2] {image}. image 3 is [IMG3] {image}. image 4 is [IMG4] {image}. image 5 is [IMG5] {image}. image 6 is [IMG6] {image}. image 7 is [IMG7] {image}. After viewing the images, provide a summary of the main events or key points depicted.

(3) image 0 is [IMG0] {image}. image 1 is [IMG1] {image}. image 2 is [IMG2] {image}. image 3 is [IMG3] {image}. image 4 is [IMG4] {image}. image 5 is [IMG5] {image}. image 6 is [IMG6] {image}. image 7 is [IMG7] {image}. Pay close attention to the details in the images and provide accurate description to the images based on what you see.

(4) image 0 is [IMG0] {image}. image 1 is [IMG1] {image}. image 2 is [IMG2] {image}. image 3 is [IMG3] {image}. image 4 is [IMG4] {image}. image 5 is [IMG5] {image}. image 6 is [IMG6] {image}. image 7 is [IMG7] {image}. Utilize your comprehension skills to describe the context and events depicted in the images.

(5) image 0 is [IMG0] {image}. image 1 is [IMG1] {image}. image 2 is [IMG2] {image}. image 3 is [IMG3] {image}. image 4 is [IMG4] {image}. image 5 is [IMG5] {image}. image 6 is [IMG6] {image}. image 7 is [IMG7] {image}. Reflect on the images's narrative structure and identify any storytelling techniques or narrative devices used. Write a detailed description of what you see.

(6) image 0 is [IMG0] {image}. image 1 is [IMG1] {image}. image 2 is [IMG2] {image}. image 3 is [IMG3] {image}. image 4 is [IMG4] {image}. image 5 is [IMG5] {image}. image 6 is [IMG6] {image}. image 7 is [IMG7] {image}. Consider both the explicit and implicit information conveyed in the images to provide comprehensive description of the images.

Table 14: Instruction templates for task MSRVTT.

**Templates of Video Question Answering (MSRVTT QA, MSVD, etc)**

(1) image 0 is [IMG0] {image}. image 1 is [IMG1] {image}. image 2 is [IMG2] {image}. image 3 is [IMG3] {image}. image 4 is [IMG4] {image}. image 5 is [IMG5] {image}. image 6 is [IMG6] {image}. image 7 is [IMG7] {image}. Watch the provided images carefully and answer the following questions based on your understanding of the images content. Qusetion: {question}. Answer:

(2) image 0 is [IMG0] {image}. image 1 is [IMG1] {image}. image 2 is [IMG2] {image}. image 3 is [IMG3] {image}. image 4 is [IMG4] {image}. image 5 is [IMG5] {image}. image 6 is [IMG6] {image}. image 7 is [IMG7] {image}. Carefully analyze the visual elements of the images and answer the questions based on your observations. Qusetion: {question}. Answer:

(3) image 0 is [IMG0] {image}. image 1 is [IMG1] {image}. image 2 is [IMG2] {image}. image 3 is [IMG3] {image}. image 4 is [IMG4] {image}. image 5 is [IMG5] {image}. image 6 is [IMG6] {image}. image 7 is [IMG7] {image}. Pay close attention to the details in the images and provide accurate answers to the questions based on what you see. Qusetion: {question}. Answer:

(4) image 0 is [IMG0] {image}. image 1 is [IMG1] {image}. image 2 is [IMG2] {image}. image 3 is [IMG3] {image}. image 4 is [IMG4] {image}. image 5 is [IMG5] {image}. image 6 is [IMG6] {image}. image 7 is [IMG7] {image}. Utilize your comprehension skills to answer the questions based on the context and events depicted in the images. Qusetion: {question}. Answer:

(5) image 0 is [IMG0] {image}. image 1 is [IMG1] {image}. image 2 is [IMG2] {image}. image 3 is [IMG3] {image}. image 4 is [IMG4] {image}. image 5 is [IMG5] {image}. image 6 is [IMG6] {image}. image 7 is [IMG7] {image}. Consider the relationships between the images frames, scenes, and the provided questions to formulate accurate answers. Qusetion: {question}. Answer:

(6) image 0 is [IMG0] {image}. image 1 is [IMG1] {image}. image 2 is [IMG2] {image}. image 3 is [IMG3] {image}. image 4 is [IMG4] {image}. image 5 is [IMG5] {image}. image 6 is [IMG6] {image}. image 7 is [IMG7] {image}. Use your knowledge of the images's content to answer the questions by recalling specific details and events. Qusetion: {question}. Answer:

(7) image 0 is [IMG0] {image}. image 1 is [IMG1] {image}. image 2 is [IMG2] {image}. image 3 is [IMG3] {image}. image 4 is [IMG4] {image}. image 5 is [IMG5] {image}. image 6 is [IMG6] {image}. image 7 is [IMG7] {image}. Make logical inferences based on the information presented in the images to answer the questions with reasoned explanations. Qusetion: {question}. Answer:

(8) image 0 is [IMG0] {image}. image 1 is [IMG1] {image}. image 2 is [IMG2] {image}. image 3 is [IMG3] {image}. image 4 is [IMG4] {image}. image 5 is [IMG5] {image}. image 6 is [IMG6] {image}. image 7 is [IMG7] {image}. While answering the questions, consider both the explicit and implicit information conveyed in the images to provide comprehensive responses. Qusetion: {question}. Answer:

(9) image 0 is [IMG0] {image}. image 1 is [IMG1] {image}. image 2 is [IMG2] {image}. image 3 is [IMG3] {image}. image 4 is [IMG4] {image}. image 5 is [IMG5] {image}. image 6 is [IMG6] {image}. image 7 is [IMG7] {image}. Formulate your answers by considering the temporal context of the images and the chronological order of events. Qusetion: {question}. Answer:

(10) image 0 is [IMG0] {image}. image 1 is [IMG1] {image}. image 2 is [IMG2] {image}. image 3 is [IMG3] {image}. image 4 is [IMG4] {image}. image 5 is [IMG5] {image}. image 6 is [IMG6] {image}. image 7 is [IMG7] {image}. Take into account the emotions, actions, and interactions of the characters in the images when answering the questions. Qusetion: {question}. Answer:

Table 15: Instruction templates for task MSRVTT QA and MSVD.

| Templates of Visual Reasoning (GQA, VCR, NLVR v2, etc) |
| --- |

**GQA**

(1) image 0 is [IMG0] {image}. For the question, carefully examine the image and use your knowledge to determine the correct answer. Question: {question} Answer:

(2) image 0 is [IMG0] {image}. Given the picture [IMG0], pay attention to the wording of question and answer the following question: {question} Answer:

(3) Read the question carefully and look at image 0 labeled [IMG0] {image}. Use your intuition and common sense when answering the question: {question}

(4) Consider all of the information in image 0 labeled [IMG0] {image} when answering the question: {question}

(5) The image 0: [IMG0] {image} is the primary source of information for answering the questions. Please refer to it carefully when answering question: {question} Answer:

(6) Pay close attention to the details in image 0: [IMG0] {image}, as they may provide important information for answering the questions. Question:{question} Answer:

(7) image 0 is [IMG0] {image}. Make sure your answer is relevant to the question and the image 0. Question:{question} Answer:

(8) image 0 is [IMG0] {image}. Do not provide answers based on assumptions or personal opinions; only use the information presented in the image 0 and the question. Question:{question} Answer:

(9) Look at image 0 labeled [IMG0] {image} carefully and read question: {question}. Try to understand what is being asked before selecting an answer.

(10) image 0 is [IMG0] {image}. Consider all of the information in image 0 labeled [IMG0] when answering question. Look at objects, colors, shapes, and other details that may be relevant to question: {question} Answer:

**VCR**

(1) {prompt}. Given the options below, based on the photo [IMG0], select the most suitable answer for the following question: {question}. Options: {options}

(2) Please read the question and answer choices carefully. Select the option that best answers the question. {prompt}. Given the images, select the best option that answers the question from the available answer choices. Question: {question} Options: {options} Answer:

(3) Choose the answer that best fits the description or action in the image. {prompt}. Consider the scene depicted in the images, choose the answer that best fits the description or action in the image from the available answer choices. Question: {question} Options: {options} Answer:

(4) {prompt}. Examine the details in the pictures and use them to inform your answer to the question. Choose the best answer from the available options. Question: {question} Options: {options} Answer:

(5) Look closely at the images and think about what is happening in the scene. {prompt}. Given the pictures, carefully examine the images and select the best answer that describes what is happening in the scene from the available answer choices. Question: {question} Options: {options} Answer:

(6) Consider all of the details in the image and the wording of the question before making your selection. {prompt}. Given the pictures, consider all of the details in the image and the wording of the question before selecting the best answer choice from the available options. Question: {question} Options: {options} Answer:

(7) Remember to use your common sense and reasoning skills to choose the best answer. {prompt}. Think about the images, use your common sense and reasoning skills to select the best answer choice from the available options. Question: {question} Options: {options} Answer:

(8) {prompt}. Select the answer that most closely matches the description or action in images, based on the available options. Given the picture [IMG0], select the answer choice that most closely matches the description or action in the image from the available options. Question: {question} Options: {options} Answer:

(9) Choose the option that provides the most accurate and complete answer to the question, based on the available information. {prompt} Given the images, select the option that provides the most accurate and complete answer to the question from the available answer choices. Question: {question} Options: {options} Answer:

(10) {prompt}. Use the information in the images to help you make the best choice from the available answer options for the question Question: {question} Options: {options} Answer:

**NLVR v2**

(1) image 0 is [IMG0] {image}. Given the picture [IMG0], answer the following question: {question} Is this correct? True or False. Answer:

(2) For the question: {question}, carefully examine image 0: [IMG0] {image} and use your knowledge to determine if the statement is True or False.

(3) Please refer to image 0: [IMG0] {image} when answering the question: {question} Is this correct? True or False. Answer:

(4) Remember to consider both the question and the information presented in image 0: [IMG0] {image} when answering the True or False question: {question}

(5) image 0 is [IMG0] {image}.Answer the question: {question} based on the information presented in the image 0 and determine if the statement is True or False.

(6) Carefully examine the image 0: [IMG0] {image} and use your knowledge to determine whether the statement is True or False. Question: {question}

(7) Remember that the answer to each question is either True or False, so make sure you choose the correct option based on the information presented in image 0: [IMG0] {image}. Question: {question}

(8) Make sure your answers are based on the information presented in the image 0: [IMG0] {image}. Question:{question} Is this correct?True or False. Answer:

(9) Carefully examine image 0 labeled [IMG0] {image} before answering the question. Question:{question} True or False? Answer:

Table 16: Instruction templates for tasks GQV, VCR and NLVR v2.

| Model | Commonsense Reasoning | Numerical Calculation | Text Translation | Code Reasoning | Avg. |
|---|---|---|---|---|---|
| MiniGPT-4 (Zhu et al., 2023) | 59.29 | 45.00 | 0.00 | 40.00 | 36.07 |
| VisualGLM-6B (Du et al., 2021) | 39.29 | 45.00 | 50.00 | 47.50 | 45.45 |
| LLaVA (Liu et al., 2023b) | 57.14 | 50.00 | 57.50 | 50.00 | 53.66 |
| Lynx (Zeng et al., 2023) | 110.71 | 17.50 | 42.50 | 45.00 | 53.93 |
| MultiModal-GPT (Gong et al., 2023) | 49.29 | 62.50 | 60.00 | 55.00 | 56.70 |
| LLaMA-Adapter-V2 (Gao et al., 2023) | 81.43 | 62.50 | 50.00 | 55.00 | 62.23 |
| VPGTrans (Zhang et al., 2023a) | 64.29 | 50.00 | 77.50 | 57.50 | 62.32 |
| LaVIN (Luo et al., 2023) | 87.14 | 65.00 | 47.50 | 50.00 | 62.41 |
| GIT2 (Wang et al., 2022a) | 99.29 | 50.00 | 67.50 | 45.00 | 65.45 |
| mPLUG-Owl (Ye et al., 2023) | 78.57 | 60.00 | 80.00 | 57.50 | 69.02 |
| BLIP-2 (Li et al., 2023d) | 110.00 | 40.00 | 65.00 | 75.00 | 72.50 |
| InstructBLIP (Dai et al., 2023) | 129.29 | 40.00 | 65.00 | 57.50 | 72.95 |
| Otter (Li et al., 2023a) | 106.43 | 72.50 | 57.50 | 70.00 | 76.61 |
| Cheetor (Li et al., 2023c) | 98.57 | 77.50 | 57.50 | 87.50 | 78.02 |
| LRV-Instruction (Liu et al., 2023a) | 100.71 | 70.00 | 85.00 | 72.50 | 82.05 |
| BLIVA (Hu et al., 2023b) | 136.43 | 57.50 | 77.50 | 60.00 | 82.86 |
| MMICL | 136.43 | 82.50 | 132.50 | 77.50 | **107.23** |

Table 17: **Evaluation of cognition**. In the MME benchmark, each image will have two questions, with answers restricted to 'yes' or 'no'. The evaluation metrics for this benchmark include ACC and ACC+. ACC refers to the accuracy calculated for each question, while ACC+ represents the accuracy for each image, where both questions must be answered correctly. The Avg. metric denotes the average value across all numbers. It is important to note that all the reported figures for the baseline methods are obtained from the MME benchmark (Fu et al., 2023). We use the FLAN-T5-XXL version of MMICL to evaluate the performance.

All experiments are conducted with 6 NVIDIA A40 GPUs with the zero2-offload (Rajbhandari et al., 2020) of Deepspeed (Rasley et al., 2020) with the trainer of huggingface transformers (Wolf et al., 2020). The batch size is 10 and 4 for MMICL (FLAN-T5-XL) and MMICL (FLAN-T5-XXL), respectively. The largest MMICL (FLAN-T5-XXL) requires about two days for the Stage II.

# I  MME BENCHMARK

| Model | Existen. | Count | Pos. | Color | OCR | Poster | Cele. | Scene | Land. | Art. | Avg. |
|---|---|---|---|---|---|---|---|---|---|---|---|
| LLaVA | 50.00 | 50.00 | 50.00 | 55.00 | 50.00 | 50.00 | 48.82 | 50.00 | 50.00 | 49.00 | 50.28 |
| MiniGPT-4 | 68.33 | 55.00 | 43.33 | 75.00 | 57.50 | 41.84 | 54.41 | 71.75 | 54.00 | 60.50 | 58.17 |
| MultiModal-GPT | 61.67 | 55.00 | 58.33 | 68.33 | 82.50 | 57.82 | 73.82 | 68.00 | 69.75 | 59.50 | 65.47 |
| VisualGLM-6B | 85.00 | 50.00 | 48.33 | 55.00 | 42.50 | 65.99 | 53.24 | 146.25 | 83.75 | 75.25 | 70.53 |
| VPGTrans | 70.00 | 85.00 | 63.33 | 73.33 | 77.50 | 84.01 | 53.53 | 141.75 | 64.75 | 77.25 | 79.05 |
| LaVIN | 185.00 | 88.33 | 63.33 | 75.00 | 107.50 | 79.59 | 47.35 | 136.75 | 93.50 | 87.25 | 96.36 |
| LLaMA-Adapter-V2 | 120.00 | 50.00 | 48.33 | 75.00 | 125.00 | 99.66 | 86.18 | 148.50 | 150.25 | 69.75 | 97.27 |
| mPLUG-Owl | 120.00 | 50.00 | 50.00 | 55.00 | 65.00 | 136.05 | 100.29 | 135.50 | 159.25 | 96.25 | 96.73 |
| InstructBLIP | 185.00 | 143.33 | 66.67 | 153.33 | 72.50 | 123.81 | 101.18 | 153.00 | 79.75 | 134.25 | 121.28 |
| BLIP-2 | 160.00 | 135.00 | 73.33 | 148.33 | 110.00 | 141.84 | 105.59 | 145.25 | 138.00 | 136.50 | 129.38 |
| Lynx | 195.00 | 151.67 | 90.00 | 170.00 | 77.50 | 124.83 | 118.24 | 164.50 | 162.00 | 119.50 | 137.32 |
| GIT2 | 190.00 | 118.33 | 96.67 | 158.33 | 65.00 | 112.59 | 145.88 | 158.50 | 140.50 | 146.25 | 133.21 |
| Otter | 195.00 | 88.33 | 86.67 | 113.33 | 72.50 | 138.78 | 172.65 | 158.75 | 137.25 | 129.00 | 129.23 |
| Cheetor | 180.00 | 96.67 | 80.00 | 116.67 | 100.00 | 147.28 | 164.12 | 156.00 | 145.73 | 113.50 | 130.00 |
| LRV-Instruction | 165.00 | 111.67 | 86.67 | 165.00 | 110.00 | 139.04 | 112.65 | 147.98 | 160.53 | 101.25 | 129.98 |
| BLIVA | 180.00 | 138.33 | 81.67 | 180.00 | 87.50 | 155.10 | 140.88 | 151.50 | 89.50 | 133.25 | 133.77 |
| MMICL | 170.00 | 160.00 | 81.67 | 156.67 | 100.00 | 146.26 | 141.76 | 153.75 | 136.13 | 135.50 | **138.17** |

Table 18: **Evaluation of coarse-grained and fine-grained recognition and OCR**. The settings are the same as Table 17. It is important to note that all the reported figures for the baseline methods are obtained from the MME benchmark (Fu et al., 2023). We use the FLAN-T5-XXL version of MMICL to evaluate the performance.

| Model | Existence | | Count | | Position | | Color | | OCR | | Avg. |
|---|---|---|---|---|---|---|---|---|---|---|---|
| | ACC | ACC+ | ACC | ACC+ | ACC | ACC+ | ACC | ACC+ | ACC | ACC+ | |
| BLIP-2 | 86.67 | 73.33 | 75.00 | 60.00 | **56.67** | 16.67 | 81.67 | 66.67 | **70.00** | 40.00 | 62.67 |
| LLaVA | 50.00 | 0.00 | 50.00 | 0.00 | 50.00 | 0.00 | 51.67 | 3.33 | 50.00 | 0.00 | 25.49 |
| MiniGPT-4 | 75.00 | 60.00 | 66.67 | 56.67 | **56.67** | **33.33** | 71.67 | 53.33 | 62.50 | 35.00 | 57.08 |
| mPLUG-Owl | 73.33 | 46.67 | 50.00 | 0.00 | 50.00 | 0.00 | 51.67 | 3.33 | 55.00 | 10.00 | 34.00 |
| LLaMA-Adapter-V2 | 76.67 | 56.67 | 58.33 | 6.67 | 43.33 | 3.33 | 55.00 | 16.67 | 57.50 | 15.00 | 38.92 |
| VisualGLM-6B | 61.67 | 23.33 | 50.00 | 0.00 | 48.33 | 0.00 | 51.67 | 3.33 | 42.50 | 0.00 | 28.08 |
| Otter | 53.33 | 6.67 | 50.00 | 0.00 | 50.00 | 0.00 | 51.67 | 3.33 | 50.00 | 0.00 | 26.50 |
| Multimodal-GPT | 46.67 | 10.00 | 51.67 | 6.67 | 45.00 | 13.33 | 55.00 | 13.33 | 57.50 | 25.00 | 32.42 |
| PandaGPT | 56.67 | 13.33 | 50.00 | 0.00 | 50.00 | 0.00 | 50.00 | 0.00 | 50.00 | 0.00 | 27.00 |
| MMICL | **90.00** | **80.00** | **86.67** | **73.33** | 55.00 | 26.67 | **88.33** | **73.33** | 60.00 | **40.00** | **67.33** |

Table 19: Fine-grained result of MME benchmark

| Model | Poster | | Celebrity | | Scene | | Landmark | | Artwork | | Avg. |
|---|---|---|---|---|---|---|---|---|---|---|---|
| | ACC | ACC+ | ACC | ACC+ | ACC | ACC+ | ACC | ACC+ | ACC | ACC+ | |
| BLIP-2 | 79.25 | 62.59 | 58.53 | 37.06 | 81.25 | 64.00 | **79.00** | 59.00 | 76.50 | **60.00** | 66.72 |
| LLaVA | 50.00 | 0.00 | 48.82 | 0.00 | 50.00 | 0.00 | 50.00 | 0.00 | 49.00 | 0.00 | 24.78 |
| MiniGPT-4 | 49.32 | 19.73 | 58.82 | 24.71 | 68.25 | 45.50 | 59.75 | 30.50 | 56.25 | 27.00 | 44.00 |
| mPLUG-Owl | 77.89 | 57.14 | 66.18 | 34.12 | 78.00 | 57.50 | 86.25 | **73.00** | 63.25 | 33.00 | 62.63 |
| LLaMA-Adapter-V2 | 52.72 | 10.88 | 55.00 | 21.18 | 68.75 | 44.50 | 53.00 | 9.00 | 52.50 | 14.50 | 38.2 |
| InstructBLIP | 74.15 | 49.66 | 67.06 | 34.12 | **84.00** | 69.00 | 59.75 | 20.00 | **76.75** | 57.50 | 59.20 |
| VisualGLM-6B | 54.42 | 12.24 | 50.88 | 2.35 | 81.75 | 64.50 | 59.75 | 24.00 | 55.75 | 20.00 | 42.56 |
| Otter | 45.24 | 0.00 | 50.00 | 0.00 | 55.00 | 14.50 | 52.00 | 4.50 | 48.00 | 5.50 | 27.47 |
| Multimodal-GPT | 45.24 | 17.01 | 49.12 | 24.12 | 50.50 | 17.50 | 50.50 | 23.00 | 46.00 | 12.00 | 33.50 |
| PandaGPT | 56.80 | 19.73 | 46.47 | 10.59 | 72.50 | 45.50 | 56.25 | 13.50 | 50.25 | 1.00 | 37.26 |
| MMICL | **81.63** | **64.63** | **79.41** | **62.35** | 83.75 | **70.00** | 76.96 | 59.16 | 76.50 | 59.00 | **71.04** |

Table 20: Fine-grained result of MME benchmark

MME comprehensively evaluates VLMs with 14 sub-tasks that encompass perception and cognition abilities. Other than OCR, perception ability includes the recognition of coarse-grained and fine-grained objects. The former identifies the existence, count, position, and color of objects. The latter recognizes movie posters, celebrities, scenes, landmarks, and artworks. The cognition includes commonsense reasoning, numerical calculation, text translation, and code reasoning.

MME evaluates a wide range of multi-modal abilities. The compared baselines include LLaVA (Liu et al., 2023b), MiniGPT-4 (Zhu et al., 2023), MultiModal-GPT (Gong et al., 2023), VisualGPM-6B (Du et al., 2021), VPGTrans (Zhang et al., 2023a) , LaVIN (Luo et al., 2023), mPLUG-Owl (Ye et al., 2023), LLaMA-Adapter-V2 (Gao et al., 2023), InstructBLIP (Dai et al., 2023), Otter (Li et al., 2023a), BLIP-2 (Li et al., 2023d), LRV-Instruction (Liu et al., 2023a), Cheetor (Li et al., 2023c), GIT2 (Wang et al., 2022a), Lynx (Zeng et al., 2023), BLIVA (Hu et al., 2023b). We also provide more detail evaluation results for MMICL at Table 18, Table 19, Table 20, and Table 21. Results show that MMICL can achieve the best average scores in comparisons with current VLMs.

## J  MMBENCH BENCHMARK

MMBench (Liu et al., 2023c) is a thoughtfully designed benchmark that thoroughly evaluates the diverse skills of vision-language models. The results of all different VLMs from the test set are presented in Table 22.

## K  UNDERSTANDING MULTIPLE IMAGES IN THE MULTI-MODAL PROMPT

Videos contain more temporal information compared to static images. We test MMICL across different video-languages tasks to evaluate whether the MMICL is able to support the multiple images in the complex prompts. The result is present in Table 23. Our model, MMICL, achieved

| Model | Common. Reason. | | Numerical Calculation | | Text Translation | | Code Reason. | | Avg. |
|---|---|---|---|---|---|---|---|---|---|
| | ACC | ACC+ | ACC | ACC+ | ACC | ACC+ | ACC | ACC | |
| BLIP-2 | 68.57 | 41.43 | 40.00 | 0.00 | 55.00 | 10.00 | 55.00 | 20.00 | 36.25 |
| LLaVA | 49.29 | 11.43 | 50.00 | 0.00 | 52.50 | 5.00 | 50.00 | 0.00 | 27.27 |
| MiniGPT-4 | 58.57 | 34.29 | 47.50 | 20.00 | 42.50 | 15.00 | **67.50** | **45.00** | 41.30 |
| mPLUG-Owl | 59.29 | 24.29 | 50.00 | 10.00 | 60.00 | 20.00 | 47.50 | 10.00 | 35.14 |
| LLaMA-Ada.-V2 | 54.29 | 14.29 | **52.50** | 5.00 | 52.50 | 5.00 | 52.50 | 10.00 | 30.76 |
| InstructBLIP | 75.00 | 54.29 | 35.00 | 5.00 | 55.00 | 10.00 | 47.50 | 0.00 | 35.22 |
| VisualGLM-6B | 45.71 | 12.86 | 45.00 | 0.00 | 55.00 | 10.00 | 50.00 | 0.00 | 27.32 |
| Otter | 48.57 | 10.00 | 47.50 | 10.00 | 55.00 | 10.00 | 50.00 | 0.00 | 28.88 |
| MultiModal-GPT | 45.71 | 5.71 | 50.00 | 20.00 | 50.00 | 5.00 | 45.00 | 10.00 | 28.93 |
| PandaGPT | 56.43 | 17.14 | 50.00 | 0.00 | 52.50 | 5.00 | 47.50 | 0.00 | 28.67 |
| MMICL | **76.43** | **60.00** | 47.50 | **35.00** | **72.50** | **60.00** | 47.50 | 30.00 | **53.62** |

Table 21: Fine-grained result of MME benchmark

| Method | Language Model | Vision Model | Overall | LR | AR | RR | FP-S | FP-C | CP |
|---|---|---|---|---|---|---|---|---|---|
| MMGPT | LLaMA-7B | CLIP ViT-L/14 | 16.0 | 1.1 | 23.8 | 20.7 | 18.3 | 5.2 | 18.3 |
| MiniGPT-4 | Vincuna-7B | EVA-G | 12.0 | 13.6 | 32.9 | 8.9 | 28.8 | 11.2 | 28.3 |
| PandaGPT | Vincuna-13B | ImageBind ViT-H/14 | 30.6 | 15.3 | 41.5 | 22.0 | 20.3 | 20.4 | 47.9 |
| VisualGLM | ChatGLM-6B | EVA-CLIP | 33.5 | 11.4 | 48.8 | 27.7 | 35.8 | 17.6 | 41.5 |
| InstructBLIP | Vincuna-7B | EVA-G | 33.9 | 21.6 | 47.4 | 22.5 | 33.0 | 24.4 | 41.1 |
| LLaVA | LLaMA-7B | CLIP ViT-L/14 | 36.2 | 15.9 | 53.6 | 28.6 | 41.8 | 20.0 | 40.4 |
| G2PT | LLaMA-7B | ViT-G | 39.8 | 14.8 | 46.7 | 31.5 | 41.8 | 34.4 | 49.8 |
| Otter-I | LLaMA-7B | CLIP ViT-L/14 | 48.3 | 22.2 | 63.3 | 39.4 | 46.8 | 36.4 | 60.6 |
| Shikra | Vincuna-7B | CLIP ViT-L/14 | 60.2 | 33.5 | 69.6 | 53.1 | 61.8 | 50.4 | 71.7 |
| LMEye | Flan-XL | CLIP ViT-L/14 | 61.3 | 36.9 | 73.0 | 55.4 | 60.0 | 68.0 | 68.9 |
| mPLUG-Owl | LLaMA-7B | CLIP ViT-L/14 | 62.3 | 37.5 | 75.4 | 56.8 | 67.3 | 52.4 | 67.2 |
| JiuTian | FLANT5-XXL | EVA-G | 64.7 | 46.6 | 76.5 | 66.7 | 66.5 | 51.6 | 68.7 |
| MMICL | FLAN-T5-XXL | EVA-G | **65.24** | 44.32 | 77.85 | 64.78 | 66.5 | 53.6 | 70.64 |

Table 22: **Evaluation of MM benchmark dev set**. All the reported performance for the baseline methods is from the leaderboard of MM benchmark (Liu et al., 2023c). We use the FLAN-T5-XXL version of MMICL to evaluate the performance.

significant improvement of 10.86, 4.53, and 2.45 points for MSVD-QA (Chen & Dolan, 2011), NExT-QA (Xiao et al., 2021), and iVQA (Yang et al., 2021) respectively, when compared to the strongest baselines. It is important to note that our training dataset did not include any videos. This indicates that MMICL effectively enhances the model's ability to understand temporal information in videos.

## L  OBJECT HALLUCINATION EVALUATION

We test the following VLMs on the POPE benchmark to evaluate their object hallucination performance: MMICL, Shikra (Chen et al., 2023a), InstructBLIP (Dai et al., 2023), MiniGPT-4 (Zhu et al., 2023), LLaVA (Liu et al., 2023b), MM-GPT (Gong et al., 2023) and mPLUG-Owl (Ye et al., 2023). The result is present in the Table 24.

## M  DETAILS FOR EVALUATION

In this Section. we provide details for evaluation in our experiments as Sec. 3.

### M.1  EVALUATION METRICS

We provide evaluation metrics as Table 25

| Model | MSVD QA | NExT QA Multi-choice | iVQA |
|---|---|---|---|
| Flamingo-3B (Alayrac et al., 2022) (Zero-Shot) | 27.50 | - | 32.70 |
| Flamingo-3B (Alayrac et al., 2022) (4-Shot) | 33.00 | - | 35.20 |
| Flamingo-9B (Alayrac et al., 2022) (Zero-Shot) | 30.20 | - | 35.20 |
| Flamingo-9B (Alayrac et al., 2022) (4-Shot) | 36.20 | - | 37.70 |
| Flamingo-80B (Alayrac et al., 2022) (Zero-Shot) | 35.60 | - | 40.70 |
| Flamingo-80B (Alayrac et al., 2022) (4-Shot) | 41.70 | - | 44.10 |
| R2A (Pan et al., 2023) | 37.00 | - | 29.30 |
| BLIP-2 (Li et al., 2023d) (FLANT5-XL) | 33.70 | 61.73 | 37.30 |
| BLIP-2 (Li et al., 2023d) (FLANT5-XXL) | 34.40 | 61.97 | 49.38 |
| InstructBLIP (Dai et al., 2023) (FLANT5-XL) | 43.40 | 36.10 | 25.18 |
| InstructBLIP (Dai et al., 2023) (FLANT5-XXL) | 44.30 | 64.27 | 36.15 |
| MMICL (FLAN-T5-XL) | 47.31 | 66.17 | 41.68 |
| MMICL (FLAN-T5-XXL) | **55.16** | 64.67 | 41.13 |
| MMICL (Instruct-FLAN-T5-XL) | 53.68 | 65.33 | 49.28 |
| MMICL (Instruct-FLAN-T5-XXL) | 52.19 | **68.80** | **51.83** |

Table 23: Results of MMICL compared with other VLMs across different video-languages tasks. For Blip-2 and Instructblip, We concatenate the visual embeddings of all frames and place them on the top of the textual prompts following Dai et al. (2023).

Table 24: Performance result of different VLMs on the POPE benchmark

| Dataset | Metric | Models | | | | | | |
|---|---|---|---|---|---|---|---|---|
| | | MMICL | Shikra | InstructBLIP | MiniGPT-4 | LLaVA | MM-GPT | mPLUG-Owl |
| Random | Accuracy | 87.29 | 86.90 | 88.57 | 79.67 | 50.37 | 50.10 | 53.97 |
| | Precision | 94.63 | 94.40 | 84.09 | 78.24 | 50.19 | 50.05 | 52.07 |
| | Recall | 79.87 | 79.27 | 95.13 | 82.20 | 99.13 | 100.00 | 99.60 |
| | F1-Score | 86.62 | 86.19 | 89.27 | 80.17 | 66.64 | 66.71 | 68.39 |
| | Yes | 43.51 | 43.26 | 56.57 | 52.53 | 98.77 | 99.90 | 95.63 |
| Popular | Accuracy | 82.70 | 83.97 | 82.77 | 69.73 | 49.87 | 50.00 | 50.90 |
| | Precision | 85.11 | 87.55 | 76.27 | 65.86 | 49.93 | 50.00 | 50.46 |
| | Recall | 79.27 | 79.20 | 95.13 | 81.93 | 99.27 | 100.00 | 99.40 |
| | F1-Score | 82.08 | 83.16 | 84.66 | 73.02 | 66.44 | 66.67 | 66.94 |
| | Yes | 46.57 | 45.23 | 62.37 | 62.20 | 99.40 | 100.00 | 98.57 |
| Adversarial | Accuracy | 80.97 | 83.10 | 72.10 | 65.17 | 49.70 | 50.00 | 50.67 |
| | Precision | 81.88 | 85.60 | 65.13 | 61.19 | 49.85 | 50.00 | 50.34 |
| | Recall | 79.53 | 79.60 | 95.13 | 82.93 | 99.07 | 100.00 | 99.33 |
| | F1-Score | 80.69 | 82.49 | 77.32 | 70.42 | 66.32 | 66.67 | 66.82 |
| | Yes | 48.57 | 46.50 | 73.03 | 67.77 | 99.37 | 100.00 | 98.67 |

## M.2 VQA TOOLS

We use the same VQA Tools as the original VQA paper (Agrawal et al., 2016) and use it in all metrics using the VQA accuracy.

## N ABLATION STUDY

### N.1 ABLATION STUDY ON CONTEXT SCHEME

We conduct ablation experiments using the InstructBLIP-FLANT5-XL as the backbone model in the following settings to verify the effectiveness of our proposed context scheme:

| Dataset | Metrics |
|---|---|
| MSVD (Chen & Dolan, 2011) | Top-1 Acc. |
| iVQA (Yang et al., 2021) | iVQA Acc. |
| NExT-QA-multiple-choice (Xiao et al., 2021) | Top-1 Acc. |
| NExT-QA-opendomain (Xiao et al., 2021) | WUPS Score. |
| Hateful Memes (Kiela et al., 2020) | AUC Score |
| WebSRC (Chen et al., 2021b) | Exact Match |
| VSR (Liu et al., 2022) | Top-1 Acc. |
| *VQAv2 (Goyal et al., 2017) | VQA Acc. |
| VizWiz (Bigham et al., 2010) | VQA Acc. |
| IconQA-text (Lu et al., 2021) | Top-1 Acc. |
| IconQA-img (Lu et al., 2021) | Top-1 Acc. |
| ScienceQA-IMG (Lu et al., 2022) | Top-1 Acc. |
| Bongard-HOI (Jiang et al., 2022) | Top-1 Acc. |
| VisDial (Das et al., 2017) | Exact Match |
| NoCaps (Agrawal et al., 2019) | Cider Score |
| A-OKVQA (Agrawal et al., 2019) | Top-1 Acc. |
| *Flickr (Young et al., 2014) | Cider Score |
| Winoground (Thrush et al., 2022b) | Winoground mertic. |
| Raven IQ Test (Huang et al., 2023a) | Top-1 Acc. |
| Minecraft | Top-1 Acc. |

Table 25: Summary of the evaluation datasets and metrics. These datasets are used to validate the general design of MMICL. The datasets marked with * are the hold-in datasets, where their training set is used in training the MMICL.

| Winoground | Image-score | Text-score | Group-score |
|---|---|---|---|
| w/o context scheme | 12.25 | 17.25 | 6.00 |
| w/o image declaration | 4.00 | 22.25 | 3.00 |
| w/o in-context format | 31.25 | 25.75 | 20.00 |
| w/o interrelated images | 36.50 | 31.25 | 25.75 |
| MMICL | **46.00** | **39.25** | **38.75** |

Table 26: Ablation study on Context Scheme for different scores on Winoground.

- **w/o context scheme**: This model is trained only with instructions using all datasets from stage 2 of MMICL, without any additional design in context format and model architecture. It aims to ablate the contribution of the multi-modal instruction tuning.

- **w/o image declaration**: This model is trained without image declaration but includes multi-modal data with interconnected images and in-context data. It utilizes all datasets used in stage 2 of MMICL and intends to evaluate the impact of image declaration design.

- **w/o in-context format**: This model, devoid of in-context format data, comprises of multi-modal data with related images and image declaration. It uses all datasets found in stage 2 of MMICL with a single image-text pair and aims to assess the effectiveness of multi-modal in-context data design.

- **w/o interrelated images**: This version of the model is trained without any interconnected images. Its training pipeline contains multi-modal in-context data and image declaration. It utilizes all datasets used in stage 2 of MMICL, excluding the video/VCR datasets. It aims to evaluate the contribution of the design which incorporates multi-modal data with interrelated images

We use the equivalent amount of data as in the stage II of MMICL for the ablation models. We employ the ablation study on both the general benchmark(MME), Winoground (Thrush et al., 2022b) and multi-image datasets(IconQA img (Lu et al., 2021) & NLVR2 (Suhr et al., 2018), Raven IQ-test (Huang et al., 2023a)) to comprehensively evaluate the sources of the observed performance improvement in MMICL.

The ablation results in Table 7 affirm that the superiority of MMICL is driven by the collective impact of our design elements—removing any component cannot guarantee superior performance of our model. Each component of our design significantly contributes to different aspects of our model.

**Ablation of Context Scheme:** Our improved results demonstrate the effectiveness of our context scheme design rather than merely aggregating more pretraining data into the pipeline. The substantial improvements across all tasks when compared to the w/o context scheme further substantiate its effectiveness. Lacking the context scheme and our model architecture supporting the scheme, the model performed poorly across most datasets, except performance on the single image task focusing on specific object perception (such as Existence, Count). This is presumably due to the alignment of the additional single image-text pair training with the task's evaluation format.

**Ablation of Interrelated Images:** The exclusion of this design leads to a significant decrease in multi-image understanding ability, evidenced in the NLVR2 and Raven datasets. However, multi-modal in-context training pipeline still enabled the model to retain the multi-image understanding ability, achieving an advantage over datasets with multiple images (IconQA img, NLVR2, Raven) compared to the w/o context scheme. At the same time, the design of multi-modal in-context data still improves the model on single image understanding over the w/o in-context format, particularly in understanding the task about specific object perception (Perception), but performed poorly in more abstract tasks involving recognition and cognition(Cognition). When compared to the w/o in-context format, it demonstrates a superior understanding of paired image and text data, evident in the performance in Winoground. This suggests that multi-modal data with interrelated images provide some assistance in understanding multiple images. However, without the help of multi-modal ICL data, it will not display superior performance. It aligns with our initial insight.

**Ablation of In-context Format:** Excluding this design revealed that even with the introduction of extra datasets, performance improvement on the single image tasks was not achieved. Compared to other ablation settings, despite the w/o in-context format receiving a noticeable improvement in tasks related to comprehension and inference(Cognition) compared to other ablation settings, we observed a significant drop in the task about specific object perception (Perception). Training with multiple images enhanced its ability to better understand abstract logic information within a single image and to perform better in downstream tasks(Cognition). However, the model lost its ability to analyze and recognize specific objects in a single image, leading to a significant decrease in the perception score of MME. We found out that, by benefiting from learning on interrelated image data, its understanding ability on multiple images has not significantly declined, exhibiting its superiority over other settings. The performance in the MME also confirms this, indicating that multi-modal in-context data offers significant advantages for comprehending multiple images and enhancing the model's ability to understand abstract information in the signal image.

**Ablation of Image Declaration:** The removal of image declaration led to a stark decline in performance in understanding complex image-text relationships (Winoground) due to a lack of explicit modeling between image-text references. This phenomenon was also observed in tasks that involved understanding multiple image relations(Raven, IconQA img, NLVR2). However, it can be observed lacking interrelated images harms the model's multi-image understanding more than missing image declarations, given the relatively better performance of model without image declarations. This indicates the importance of image declaration in handling the multi-image and image-text relationships.

**Ablation of In-context Learning Ability:** The ablation results in Table 27 clarify the significance of the proposed context scheme for boosting the in-context learning ability of MMICL. The effectiveness of MMICL in improving the in-context learning ability of the VLM is attributed to the integration of all three designs in the context scheme, not merely the in-context learning format in the designed scheme. Notably, ICL-only achieves the most substantial enhancements compared to the other settings in the Vizwiz dataset. This highlights the efficacy of tuning with in-context format data to augment the model's comprehension of in-context instances. However, the modest advancements on the Flickr dataset indicate that, in the absence of multi-image instruction tuning, the model finds it challenging to effectively leverage in-context examples for enhancing its performance in tasks without fixed output formats, such as image captioning.

| Task | ICL example | MMICL | w/o context scheme | w/o in-context format | w/o interrelated images |
|------|-------------|-------|--------------------|-----------------------|-------------------------|
| VizWiz | w/o ICL example | 27.10 | 28.13 | 15.90 | 23.43 |
|  | w ICL example(5) | 42.66 | 36.23 | 28.33 | 37.13 |
|  | Δ | **15.56** | 8.10 | 12.43 | 13.70 |
| Flickr | w/o ICL example | 83.94 | 77.84 | 65.33 | 76.88 |
|  | w ICL example(5) | 88.11 | 79.38 | 65.95 | 77.09 |
|  | Δ | **4.17** | 1.54 | 0.62 | 0.18 |

Table 27: An Ablation Study of the Contribution of the Context Scheme to the Model's In-Context Learning Ability

## O  BASELINES

**Baselines**  We primarily compare MMICL with recently proposed powerful multi-modal approaches, including:

**(1) Flamingo** (Alayrac et al., 2022) where a VLM is trained on large-scale multi-modal- web corpora containing arbitrarily interleaved text and images;

**(2) KOSMOS-1** (Huang et al., 2023a) which is trained from scratch on web-scale multi-modal corpora;

**(3) BLIP-2-FLAN-T5** (Li et al., 2023d) where an instruction-tuned Flan-T5 (Chung et al., 2022) is connected with a powerful visual encoder to perform a series of multi-modal tasks;

**(4) InstructBLIP-FLAN-T5** (Dai et al., 2023), a recently proposed instruction tuning enhanced multi-modal agents with FLAN-T5 with converted multi-modal datasets and the LLaVA (Liu et al., 2023b) dataset generated by GPT-4 (OpenAI, 2023);

**(5) Shikra** (Chen et al., 2023a), a VLM that can handle spatial coordinate inputs and outputs in natural language without the need for extra vocabularies or external plugin models. All inputs and outputs of Shikra are in natural language form.

**(6) Otter** (Li et al., 2023a), an open-source implementation of flamingo (Alayrac et al., 2022). By utilizing multi-modal instruction in-context tuning data, Otter fine-tunes Openflamingo to augment its instruction comprehension capabilities while maintaining its ability to learn in context;

**(7) Ying-VLM** (Li et al., 2023e), a VLM model trained on Multi-Modal multilingual instruction tuning dataset, showcasing its potential to answer complex questions requiring world knowledge, generalize to unseen video tasks, and comprehend unseen instructions in Chinese.

## P  OOD GENERALIZATION TO UNSEEN DOMAIN

| Method | Shot | Top-1 Acc. |
|--------|------|------------|
| MiniGPT-4 (Vincuna-7B) | Zero-Shot | 35.10 |
| MiniGPT-4 (Vincuna-13B) | Zero-Shot | 48.40 |
| MMICL (FLAN-T5-XL) | Zero-Shot | 55.41 |
| MMICL (FLAN-T5-XL) | 4-Shot | 64.05 |
| MMICL (FLAN-T5-XXL) | 8-Shot | 65.41 |

Table 28: Results of generalization of MMICL to unseen domain in Minecraft. Results show that MMICL is able to generalize to unseen domains and tasks given a few examples.

In an unseen challenging domain with limited exemplars, analyzing regular patterns, reasoning, and learning new knowledge (OOD Generalization to unseen domain) is a great way to test multi-modal ICL ability.

We construct a task using Minecraft (Chen et al., 2024a; 2023c; Cipollone et al., 2014), which requires the VLM to identify whether an animal (i.e., cow, llama, chicken, donkey, and so on) is present in case (d) of Fig. 1.

| Model | rec | ocr | know | gen | spat | math | total |
|---|---|---|---|---|---|---|---|
| InstructBlip(Flant5-XXL) | 17.10 | 8.40 | 5.50 | 2.60 | 8.00 | 3.80 | 14.10 |
| MMICL(InstructBlip-Flant5-XXL) | **29.90** | **14.00** | **17.40** | **11.20** | **18.10** | **3.80** | **24.80** |

Table 29: Performance metrics for different models.

We collect 550 cases and transfer the task to a vision-to-text question-answering task to evaluate the performance of OOD generalization of MMICL. The results are shown in Table 28. Results demonstrate that MMICL is able to generalize to the Minecraft domain even if the images are extremely different compared to the images used by training at Stage I and II as Sec. 2.4.

## Q    DETAILS OF SCIENCEQA

In the test set of Science-QA, each item consists of a question and several choices. Out of 4221 items, we sampled 2017 instances that contain multimodal information incorporating both text and image, a subset we refer to as ScienceQA-IMG, while the remainder only includes text. However, we observe that not all questions within these 2017 items necessarily require answers based on image information. The standard for annotation is whether an understanding of the image sample is essential to answer the questions in the test set. [8]

To illustrate this, we provide two examples of questions and their corresponding choices:

- Question: Which animal is also adapted to be camouflaged among dead leaves? Choices: [ "strawberry poison frog", "Surinam horned frog" ];

- Question: Which property do these three objects have in common? Choices: ['shiny', 'slippery', 'opaque'].

Clearly, the first question can be answered without reference to image, while the second can not. As such, ScienceQA-IMG can be divided into two groups: questions that require images for answers (1012) and those that do not (1005).

## R    FREE-FORM ANSWERING EVALUATION

MM-VET (Yu et al., 2023) is an evaluation benchmark that examines VLMs on complex multi-modal tasks. It requires free-form answers and is evaluated by GPT-4, providing a different perspective on the model's capabilities.

It's worth noting that compared to the Flant5 backbone's instructblip model in table 29, MMICL, with the identical backbone, shows significant improvement on MM-VET. This demonstrates that our proposed method prevents the model from overfitting to a specific data structure and substantially enhances the model's free-form answering capability. Meanwhile, we can observe that the performance of MMICL on MM-VET is relatively low, which is due to MMICL using the Flant5 backbone model. As T5 is mainly fine-tuned on NLP benchmarks containing many multi-choice QA and classification datasets, it tends to output shorter content during free-form question answering, which is disadvantageous when evaluated with GPT4.

## S    PERFORMANCE COMPARISON ON THE MULTI-IMAGE DATASETS

We evaluated the MMICL model against other models with multi-modal instruction tuning on various multi-image datasets, including the Raven-IQ dataset and a random sample of 2000 instances from Nlvr2 and IconQA-img datasets. The query prompts for different methods are available in Table **??**. Our results in Table 30 demonstrate that our approach consistently outperforms others.

---

[8]The minimum hourly wage in the United States is near $7.5, which can be found at `https://www.worker.gov/`. On average, annotating an example takes 30 seconds. Consequently, the cost per annotator amounts to $262.5.

| Method | NLVR2 | IconQA-img | Raven |
|---|---|---|---|
| InstructionBlip | 53.95 | 45.10 | 10.00 |
| OTTER | 47.2 | 34.10 | 22.0 |
| MMICL | 66.6 | 60.85 | 34.00 |

Table 30: Performance comparison of different models on multi-image databases

| Method | Tasks | Query prompt |
|---|---|---|
| InstructionBlip | NLVR2 | <image><image>Carefully read the statement and evaluate the associated images, based on your analysis of the two images and statement, can you confirm if the statement: {Question} is correct? True or False? |
| | IconQA-img | <image>...<image> Use the images as a visual aid to help you answer the questions accurately. Given the options blow, based on the first image, from the following images, select the most suitable image for the question: {Question}. Options: {Options}. Select the right image. |
| | Raven | <image>...<image>User: Here are j images. Consider the relationships of the images to infer the pattern and make the correct decision. Is the last image follows the pattern. Yes or No? |
| | Winoground | <image><image>User: caption 0 is {Caption 0}.\n caption 1 is {Caption 1}.\n Given the images and captions, determine image 0 match caption 1? Yes or no? |
| Otter | NLVR2 | <image><image>User: Carefully read the statement and evaluate the associated images, based on your analysis of the two images and statement, can you confirm if the statement: {Question} is correct? True or False? GPT:<answer> |
| | IconQA-img | <image>...<image>User: Use the images as a visual aid to help you answer the questions accurately. Given the options blow, based on the first image, from the following images, select the most suitable image for the question: {Question}. Options: {Options}. Select the right image. GPT:<answer> |
| | Raven | <image>...<image>User: Here are j images. Consider the relationships of the images to infer the pattern and make the correct decision. Is the last image follows the pattern. Yes or No? GPT:<answer> |
| | Winoground | <image><image>User: caption 0 is {Caption 0}.\n caption 1 is {Caption 1}.\n Given the images and captions, determine image 0 match caption 1? Yes or no? GPT:<answer> |
| MMICL | NLVR2 | The image 0 is [IMG0]{image}.\n The image 1 is [IMG1]{image}.\n The image 0 is the on the left and the image 1 is on the right.\n Given the image 0 and image 1 as visual aid to answer the following question: {Question} is correct? True or False? |
| | IconQA-img | The image 0 is [IMG0]{image}.\n ... The image j is [IMGj]{image}.\n Given the options blow, based on the photo [IMG0], select the most suitable image for the following question: {Question}. Options: {Options}. Select the right image. |
| | Raven | Here are j images.\n image 0: [IMG0]{image}...image j: [IMGj]{image} Consider the relationships of images to infer the pattern and make correct decision whether the image 0: [IMG0] follows the pattern. Yes or No? |
| | Winoground | image 0 is [IMG0]{image}.\n image 1 is [IMG1]{image}.\n caption 0 is {Caption 0}.\n caption 1 is {Caption 1}.\n Given the images and captions, determine if image 0 matches caption 1? Yes or no? |
| | POPE | Questions is related to image 0: [IMG0]{image}. Please analyze the image 0 and pay attention to the objects in the image 0 for the question:{Question} |

Table 31: Query Prompt of different methods on different tasks

# T  LIMITATIONS

## T.1  POTENTIAL OVERFITTING:

We mitigate the risk of overfitting specific data structures by expanding the instruction templates with ChatGPT, ensuring diversity in the MIC dataset. The result in Table 28 demonstrates the generalization ability of MMICL to an unseen domain in Minecraft. This out-of-domain understanding ability demonstrates that MMICL does not overfit a specific data structure.

## T.2  CONTEXT LENGTH CONSTRAINTS:

The context length of the backbone language model imposes restrictions on the number of images that can be included in the input. This limitation renders MMICL unable to accommodate an unlimited number of images. Consequently, during MIC tuning, we integrate up to eight images per instance. Notably, MMICL demonstrates robust generalization capabilities beyond this constraint. For instance, in the context of video datasets, MMICL maintains excellent performance with inputs of up to 12/16 frames. We attribute this impressive generalization to the utilization of relative position embeddings in the architecture, as employed by Flant5 (Chung et al., 2022).

## T.3  MODEL EXPLORATION ON DECODER-ONLY ARCHITECTURES:

We acknowledge that our exploration has primarily focused on the T5 (Raffel et al., 2023) series models, and the effectiveness of our proposed approach with decoder-only architectures has not been comprehensively investigated.

