# OpenReview forum: "MMICL: Empowering Vision-language Model with Multi-Modal In-Context Learning"
_ICLR.cc/2024/Conference — ICLR 2024 poster_

### Official Review · Reviewer_eWon · 2023-10-27

**Soundness:** 2 fair
**Presentation:** 1 poor
**Contribution:** 2 fair
**Rating:** 3
**Confidence:** 4

**Summary:**

The paper introduces an interleaved image-text dataset built upon public VQA/captioning/Reasoning datasets such as Flickr/VQAv2/MSRVTT. It proposes several schemes to convert existing datasets into interleaved-image-text formats. The proposed dataset is used to train a VQA model that can take multiple images as input to answer questions (following user instructions). It shows state-of-the-art results on recent VQA benchmarks such as MME/MMBench, and demonstrated ability to improve downstream task performance using few-shot image-text samples. However, the writing of this paper is extremely bad, consisting of numerous typos, grammatical mistakes, and sentences that are difficult to understand. There are lots of missing details regarding dataset curation and evaluation protocols.

**Strengths:**

The proposed dataset with 5.8M samples could be useful for the multimodal community. The proposed MMICL model demonstrates top-tier performance on recent VQA benchmarks (MME/MMBench) compared to earlier models such as LLaVA and MiniGPT-4.

**Weaknesses:**

First, I find it **extremely hard to follow the methodology** in this paper. For example, as dataset curation is one of the most important contributions of this work, I am not able to understand what efforts the authors put in order to gather a massive 5.8M interleaved-image-text datasets. For instance, Could you explain how you manage to collect the sample as shown in Figure 3(b) and 4? In these two figures, it is shown two people quarreling with each other; this requires the original dataset to not just have bounding boxes for person1 and person2, but also the action relation (quarreling) between these two bounding boxes. I would appreciate if the authors could point us to the actual dataset used to gather this particular example, and explain if extra (and perhaps costly) human annotation is required to get such samples.

The authors should also provide more visual examples from each dataset and more detailed description on how you convert them into instruction-following formats. For example, the current Section 2.3 is very hard to understand. It says *“Next, we employ ChatGPT to rewrite the instructions to describe the key characteristics of each task accurately”* — what does it mean by *“key characteristics of each task”* and how did ChatGPT come up with them?

There are **lots of missing details** regarding model evaluation.

- For one example, in Table 4, how did the authors select the few-shot samples? Will different few-shot samples affect the performance? No variance/std is provided in Table4.

- How did you manually split ScienceQA-IMG into two groups that require images or not? Did you hire human annotators for this task? What rationales did the annotators employ?

- Table 8 shows that Flickr/VQAv2 are used in your training set. Then how could Table 4 claim that your model achieves good “zero-shot” and “few-shot” performance on Flickr/VQAv2?

- Table 8 attempted to show the licenses for all training datasets, but almost half are labeled with “Unknown” license. This is counter-intuitive because some widely used datasets such as Flickr have well-documented licenses listed on their websites.

- Do you use the same datasets during both stage-1 and stage-2 pre-training?

- As you show on Table 8, Winoground only has 800 samples. How could there be an image score of 44.99?

Finally, it is unclear if the performance boost really comes from having more interleaved-image-text data, because the current paper does not perform ablation on the selection of training data.

The paper has way too many typos. To list a few:
- Figure 1: missing a white space “MMICLtypically”.
- Page 2: fall should be fail? “their context schemes fall to”
- Page 3: missing a period “multiple images In Fig. 2.b”
- Figure 3: missing a period “unified format”
- Page 4: missing a white space “MMICLin”
- Page 8: hinds or hints “provide hinds”

**Questions:**

I think the current writing really hurts the paper. In addition to improve the writing quality, it is unclear what the key innovation / scientific insight is. The authors should ablate the use of training data in order to highlight the advantage of including interleaved image-text samples, which seems extremely costly to obtain.

---

> ### Author Response · Authors · 2023-11-19
> **Respond to the valuable advice and comments from Reviewer eWon (1/N)**
>
> Thanks for your constructive reviews and thoughtful feedback. It is encouraging to find your belief in our work's potential to contribute the multimodal community.
>
> Below, we provide detailed replies to your comments and hope we can resolve your major concerns.
>
> ### **Q1:**
>
> > First, I find it **extremely hard to follow the methodology** in this paper. For example, as **dataset curation** is one of the most important contributions of this work, I am not able to understand what efforts the authors put in order to gather a massive 5.8M interleaved-image-text datasets. For instance, Could you explain **how you manage to collect the sample as shown in Figure 3(b) and 4**? In these two figures, it is shown two people quarreling with each other; this requires the original dataset to not just have bounding boxes for person1 and person2, but also the action relation (quarreling) between these two bounding boxes. I would appreciate if the authors could point us to the actual dataset used to gather this particular example, and explain if extra (and perhaps costly) human annotation is required to get such samples.The authors should also provide more **visual examples from each dataset and more detailed description on how you convert them into instruction-following formats.** For example, the current Section 2.3 is very hard to understand. It says *“Next, we employ ChatGPT to rewrite the instructions to describe the key characteristics of each task accurately”* — what does it mean by *“key characteristics of each task”* and how did ChatGPT come up with them?
>
> ### **A1 :**
>
> We apologize for any confusion and appreciate your thorough review. We are committed to clarifying our methodology and dataset curation process. The relevant sections have been updated. Below, we provide more details for your convenience.
>
> > **Q1.1**: Dataset curation pipeline
> >
> - We have seamlessly integrated the MIC dataset using the open-source dataset into our data construction automatic pipeline at the **lowest cost**. The **only** manual **effort** required by humans is to **handcraft one instruction template** for each task. Then we expand **these templates with ChatGPT** to ensure diversity in the MIC dataset and prevent the MMICL from overfitting on specific datasets.
> - The newly added **Fig 4, Fig 9 and Fig 10** in our revised paper present a **detailed illustration of data construction**.
> - We have **included the detailed dataset curation pipeline** in **Sec 2.3 and Appendix D** in the revised paper.
>
> > **Q1.2**: How you manage to collect the sample as shown in Figure 3(b) and 4?
> >
> - As mentioned in Section 2.3, we obtain the kind of interleaved-image-text data you referred to above, with **bounding boxes and entity references** from th**e VCR dataset**.
>
>     We crop images from the VCR dataset using **object bounding boxes  provided** in the dataset to produce intertwined multi-modal data with closely related images. For each instance, it contains **an image**, several **object bounding boxes** for different entities appearing in the image, and the question for the image. All the **objects** have **been marked in the question as object reference labels**, so we can easily **replace** these labels with cropped images of the corresponding objects. Please kindly refer to **Fig 4** in our revised paper.
>
>     Specifically, take the instance you mentioned above as an example: in the original dataset, it has bounding boxes of the two people marked as **man_1** and **woman_1**. And the corresponding question：
>
>     > "Are the [man_1] and [woman_1] quarreling?"
>     >
>
> Therefore, simply replacing `man_1` and `women_1`  with the cropped images of them will result in an instance of our dataset.
>
> - We have made our tool **public**, which can be used to transform new datasets in accordance with our proposed context scheme through an automatic pipeline.
> - In summary, you can **create your own MIC dataset** using other open-source datasets by our released automatic pipeline. The only effort you need is to **write a few suitable instruction template**s for the datasets. And ask the ChatGPT to expand the instruction templates.

---

> ### Author Response · Authors · 2023-11-19
> **Respond to the valuable advice and comments from Reviewer eWon (2/N)**
>
> > **Q1.3**: Provide more visual examples from each dataset
> >
> - **Visual Examples and Dataset Curation:**
>
>     To enhance understanding, except for the example in Figure 3, we have provided **more visual examples in Fig.4, Fig.9, and Fig.10  in the revised paper**, illustrating how open-source datasets are converted into the MIC dataset in accordance with our proposed context scheme.
>
> > **Q1.4**: How you convert them into instruction-following formats?
> >
> - Allow me to provide you with a clearer overview and a **detailed description of the data collection pipeline**:
>     - **Hand-Crafted Instruction Templates:**
>         1. We scrutinize samples from each used dataset and handcraft instruction templates tailored to each dataset.
>     - **Templates Expansion by ChatGPT:**
>
>         We use ChatGPT to expand these instruction templates, adding diversity to the MIC dataset and preventing overfitting to specific data structures.
>
>         1. When expanding with ChatGPT, we give it the dataset description and our hand-craft instruction and ask chatgpt to **rewrite and** expand **these instruction templates**.
>         2. For each instance in the open-source dataset, we randomly select an expanded instruction template and fill in the data to obtain a dataset with diverse instruction formats. We aim to **enhance the diversity of MIC data** formats using these augmented instruction templates, thereby preventing model overfitting on specific data structures.
>     - **Data Construction Pipeline:**
>
>         We use the open-source datasets with instructions to construct the MIC dataset **according to our proposed context scheme**, involving **image declaration, interconnected images, and multi-modal in-context format** data:
>
>         1. **Image declaration**:
>
>             For each instance in every dataset, we create an "**image declaration" section**. This section takes the form of
>
>             > "image 0 is <image0>{visual embedding}" or "image 0: <image0>{visual embedding}".
>             >
>
>             Its purpose is to **establish a targeted connection** between language context and visual embedding.
>
>             As the number of images in an instance increases, the image IDs correspondingly increase.
>
>             Its placement is flexible. It can either appear at the start of the question concatenated with the original instance or surface within the text. This step is applied to every instance in the MIC dataset.
>
>         2. **Multi-modal data with interconnected images：**
>
>             **This step draws from two primary data sources:**
>
>             1. **Video Language Datasets (e.g., MSRVTT and MSRVTTQA)**: We extract eight frames per video from these datasets, transforming the original video-language task into a multi-image task.
>             2. **VCR dataset:** This dataset provides object bounding boxes and entity references for each object in related questions and answers. We crop images from the VCR dataset using these bounding boxes, replacing entity references in the context with the cropped images. The original image is also included at the beginning of the input context.
>         3. **Multi-modal in-context format data:** For each instance in a specific dataset, we sample questions and corresponding answers from the dataset to construct few-shot exemplars, ensuring no overlap between exemplars and instances. This process results in the creation of Multi-modal In-Context Format Data. The number of exemplars varies randomly, ranging from 0-shot to eight-shot.
> - We have **open-sourced a tool** to transform new datasets using our proposed context scheme, allowing others to create their own MIC datasets at minimal cost. The tool utilizes the expanded instruction templates and original dataset samples to generate instances following our format.
> - We have revised the whole **Sec 2.3 and provided more details about the dataset construction in Appendix D** in the revised paper.
>
> We hope this detailed explanation provides clarity on the dataset curation process and methodology used in our work. We appreciate your valuable feedback, and we **have revised the paper** to better articulate these aspects.

---

> ### Author Response · Authors · 2023-11-19
> **Respond to the valuable advice and comments from Reviewer eWon (3/N)**
>
> ### **Q2:**
> > For one example, in Table 4, how did the authors select the few-shot samples? Will different few-shot samples affect the performance? No variance/std is provided in Table4.
>
> ### **A2** :
> - Thank you for your valuable feedback, and we appreciate the opportunity to provide a more comprehensive explanation and address this concern.
>
>     We **randomly sample the  few-shot samples** in the dataset where the task belongs for all given tasks during the training and evaluation and ensure there is no gap between sampled examples and the question.
>
>     Using VQAv2 as an example, we evaluate MMICL on a 2000-instance VQAv2 dataset. We assess the robustness of MMICL performance using different seeds for random sampling of in-context learning examples. This demonstrates the robustness of MMICL.
>
>     | Seed | 0-example | 1024 | 4096 | 16384 | std |
>     | --- | --- | --- | --- | --- | --- |
>     | VQA accuracy | 69.54 | 69.98 | 69.94 | 70.21 | **0.1186** |
> - Results show that sampling of few-shot samples does not greatly affect the performance.
>
> ### **Q3:**
> > How did you manually split ScienceQA-IMG into two groups that require images or not? Did you hire human annotators for this task? What rationales did the annotators employ?
>
> ### **A3** :
> - We appreciate your inquiry and are glad to clarify it for you.
>
>     In the test set of Science-QA, each item consists of a question and several choices. Out of **4221 items**, we sampled **2017 instances that contain multimodal information** incorporating both text and image, a subset we refer to as **ScienceQA-IMG**, while the remainder only includes text. However, we observe that not all questions within these 2017 items **necessarily require answers** based on image information.
>
>     Consequently, we **engage eight locally sourced, highly-educated individuals,** **all pursuing Master's or Ph.D. degrees, to annotate the subset**. The standard for annotation is whether an understanding of the image sample is essential to answer the questions in the test set.
>
> - To illustrate this, we **provide two examples of questions** and their corresponding choices:
>
>     > a) Question: Which animal is also adapted to be camouflaged among dead leaves?  Choices: [ "strawberry poison frog", "Surinam horned frog" ];
>     >
>
>     > b) Question: Which property do these three objects have in common? Choices: ['shiny', 'slippery', 'opaque'].
>     >
>
>     Clearly, the first question can be answered without reference to the image, while the second can not. As such, ScienceQA-IMG can be divided into two groups: questions that require images for answers(**1012**) and those that do not(**1005**).
>
>     When considering these two groups, MMICL achieves the highest and least significant performance (**82.60 vs. 82.00 on Don't Require Visual Information** & **81.70 vs. 60.70 on Require Visual Information** when compared with InstructBLIP).
>
>
> This clearly demonstrates that our model successfully addresses the issue of **language bias hallucination** in Visual Language Models (VLMs), which are prone to overlooking visual content.
>
> - We have included the necessary information about the ScienceQA-img dataset **in Appendix Q** in the revised paper.
>
> ### **Q4:**
> > Table 8 shows that Flickr/VQAv2 are used in your training set. Then how could Table 4 claim that your model achieves good “zero-shot” and “few-shot” performance on Flickr/VQAv2 ?
>
> ### **A4 :**
> - We sincerely apologize for any confusion caused by the use of inappropriate terminology. In our context, the terms "zero-shot" and "few-shot" were not ideal choices to describe the model's performance.
> - Rather than emphasizing the model's zero-shot capabilities, our primary focus is on **contrasting performance with and without ICL examples.** These examples serve to validate that, for MMICL, providing a **certain quantity of ICL examples enhances performance.**
> - To avoid any further misunderstanding, we have revised the terminology as follows in **Table 4**:
>     - **"zero-shot" -> "w/o ICL examples"**
>     - **"4-shot" -> "w/ ICL examples (4)"**
>
> We believe these adjustments accurately reflect our intentions and clarify the model's performance in the specified scenarios. We hope this clarification adequately addresses your problems.
>
> ### **Q5:**
> > Table 8 attempted to show the licenses for all training datasets, but almost half are labeled with “Unknown” license. This is counter-intuitive because some widely used datasets such as Flickr have well-documented licenses listed on their websites.
>
> ### **A5** :
> - In response to your observation regarding the "Unknown" license in Table 8, we fully recognize the importance of transparency in dataset licenses. We have taken your feedback seriously, and **now completed the missing license information** for the datasets in the revised paper.
>
> We are committed to upholding the standards of the open-source community and ensuring the clarity of licensing details.

---

> ### Author Response · Authors · 2023-11-19
> **Respond to the valuable advice and comments from Reviewer eWon (4/N)**
>
> ### **Q6:**
>
> > Do you use the same datasets during both stage-1 and stage-2 pre-training?
>
> ### **A6** :
>
> - As outlined in our paper in **section 3.1 and detailed in Appendix G**, during stage-1 pre-training, we adhere to the settings of Blip-2, employing the **COCO captioning data and LAION-400M data**. In contrast, during stage-2 pre-training, we exclusively utilize the MIC dataset. Consequently, **apart from** the inclusion of the **COCO caption dataset** in both stages, there is **no overlap** in the datasets employed during stage-1 and stage-2 pre-training.
>
> We hope this clarification addresses your query adequately.
>
> ### **Q7:**
>
> > As you show on Table 8, Winoground only has 800 samples. How could there be an image score of 44.99?
>
> ### **A7** :
>
> We sincerely appreciate your diligence in pointing out the discrepancy in the number of samples and the reported image score for Winoground. We acknowledge this discrepancy, which seems to be a problem encountered across multiple works [1][2][3].
>
> - To ensure a fair comparison, we adhered to the evaluation code of [1]. Upon investigation, we identified that the issue arises from the **use of bfloat16** in the tensor calculation, which can lead to a **loss of precision**.
> - We compute and save the image-text score using the **bfloat16 data format**, and when we calculate the image score using tensor calculation and the **tensor.item()** function, there is a loss of precision when the data is cast into **float32**.
>
> We apologize for this misrepresentation and have rectified the error by updating the result to **45.00**.
>
> [1]Wang, T., Lin, K., Li, L., Lin, C. C., Yang, Z., Zhang, H., ... & Wang, L. (2023). Equivariant Similarity for Vision-Language Foundation Models. *arXiv preprint arXiv:2303.14465*.
>
> [2]Dou, Z. Y., Xu, Y., Gan, Z., Wang, J., Wang, S., Wang, L., ... & Zeng, M. (2022). An empirical study of training end-to-end vision-and-language transformers. In *Proceedings of the IEEE/CVF Conference on Computer Vision and Pattern Recognition* (pp. 18166-18176).
>
> [3]Wu, Y., Zhang, P., Xiong, W., Oğuz, B., Gee, J.C., & Nie, Y. (2023). The Role of Chain-of-Thought in Complex Vision-Language Reasoning Task.
>
> ### **Q8:**
>
> > Finally, it is unclear if the performance boost really comes from having more interleaved-image-text data, because the current paper does not perform ablation on the selection of training data.
>
> ### **A8** :
>
> Thank you for your valuable feedback, and we appreciate the opportunity to provide a more comprehensive explanation and address this concern.
>
> - In response to your suggestion, we have **conducted an ablation study** on the training data to elucidate the performance boost attributed to our proposed scheme design.
> - Specifically, we conduct ablation experiments using the Instructblip-xl as the backbone model in the following settings to verify the effectiveness of **interleaved image-text** during multi-modal ICL tuning:
>     - **MMICL(Instructblip-xl):** original model
>     - **Paired-only**: model only trained on the **single image-text pairs data**
>     - **Multi-image-only:** model only trained on the **interleaved image-text data pairs**
>     - **ICL-only:** model only trained on the **in-context learning format data**
>
>     We use **the equal amount** of data as in stage 2 of MMICL for the  ablation models.
>
>     - We employ the ablation study on both the **general benchmark(MME)** and **multi-image reasoning datasets(Icon-QA & NLVR2)** to comprehensively evaluate the sources of the observed performance improvement in MMICL. The ablation results affirm that our **interleaved image-text data scheme is a key factor** in observed improvements, providing clarity on its contribution to MMICL's performance.
>
>     | Task | MMICL | Paired-only | Multi-image-only | ICL-only |
>     | --- | --- | --- | --- | --- |
>     | MME_Perception(Overall) | **1303.59** | 1238.99 | 1141.02 | 1207.70 |
>     | MME_Cognition (Overall) | **370.71** | 316.79 | 345.36 | 333.21 |
>     | Icon-QA | **58.12** | 52.80 | 51.95 | 54.35 |
>     | NLVR2 | **72.45** | 56.65 | 62.63 | 59.60 |
> - The ablation study clearly demonstrates that the observed improvement in MMICL's performance is indeed **attributable to our proposed scheme and the incorporation of multi-modal ICL tuning.** Single image-text pair data for **SFT does not yield comparable enhancements** in the targeted areas, underscoring the unique value of our proposed method.
> - We have included the ablation study and analysis in the second sub section of **Sec. 3.6** in the revised paper.
>
> We trust that this additional information addresses your concern regarding the impact of interleaved-image-text data on performance. If you have any further questions or require additional details, please do not hesitate to reach out.  We hope that our response has addressed your concerns and kindly ask if you could consider increasing your score accordingly.

---

> ### Author Response · Authors · 2023-11-19
> **Respond to the valuable advice and comments from Reviewer eWon (5/N)**
>
> ### **Q9:**
>
> > The paper has way too many typos. To list a few:
> >- Figure 1: missing a white space “MMICLtypically”.
> >- Page 2: fall should be fail? “their context schemes fall to”
> >- Page 3: missing a period “multiple images In Fig. 2.b”
> >- Figure 3: missing a period “unified format”
> >- Page 4: missing a white space “MMICLin”
> >- Page 8: hinds or hints “provide hinds”.
> >
>
> ### **A9** :
>
> Thanks for pointing them out. All fixed. We've conducted a comprehensive typo-check to ensure a typo-free final version.
>
>
> ### **Q10:**
>
> > I think the current writing really hurts the paper. In addition to improve the writing quality, it is unclear what the **key innovation / scientific** insight is. The authors should **ablate the use of training data** in order to highlight the advantage of including interleaved image-text samples, which seems **extremely costly** to obtain.
>
> ### **A10** :
>
> We deeply appreciate your feedback. We appreciate your meticulous review of our paper, and please allow us to provide a more comprehensive explanation:
>
> > **Q10.1**:  In addition to improve the writing quality, it is unclear what the key innovation / scientific insight is.
> >
> - We believe the **key innovation insight** is that we present a **novel context scheme** in which incorporating three different designs and a multi-modal in-context-tuning **together augmenting the ICL ability of the VLM (we've highlighted this in the introduction).**
>
>     The proposed context scheme contains three designs:
>
>     - **image declaration**, which helps the VLM to **understand text-to-image reference**. It makes our model outperform others by 13-points on Winoground.
>     - **multiple interconnected images format**, which helps the VLM to better **understand the relationship between multiple images**. It makes our model perform multiimage reasoning and get 12-points improvement on Raven
>     - **unified multi-modal ICL data format**, which helps the VLM to better **learn from multi-modal in-context demonstrations**. It makes our model show impressive multi-modal ICL performance across various tasks.
>
>     These three designs address the limitations of VLMs when dealing with understanding complex multi-modal prompts and make our model outperform others.
>
> - We've done the ablation of the training data to demonstrate the advantage of our proposed scheme design. The ablation shows that the **improvement of the MMICL came from our proposed scheme and multi-modal ICL tuning. You may find these results in our response A8.**
> - Further, the results of the ScienceQA-Img experiment displayed in Sec. 3.6 demonstrates that MMICL effectively helps tackle the **language bias hallucination issue** inherent to the VLM, which tends to ignore visual content in the presence of extensive text. This characteristic enables the MMICL to outperform other models, even in single-image tasks. Our intuition is the interconnected image format in MMICL training forces the model to reason across text and multiple images, therefore mitigating the hallucination issue.
>
> > **Q10.2**: The authors should ablate the use of training data in order to highlight the advantage of including interleaved image-text samples, which seems extremely costly to obtain.
> >
> - We've conducted an ablation study on training data. Please take a look at the response A8 above.
> - It's worth noting that, as you mentioned earlier, the interleaved data we used for ICL tuning is "**extremely costly** to obtain." **on the contrary**, we have seamlessly integrated the MIC dataset using the open-source dataset into our automatic data construction pipeline **at minimal cost.**
>
> We obtained the interleaved-image-text data you referred to and **bounding boxes and entity references** from **the existing dataset**. We successfully incorporated them into our MIC dataset **at a low cost**.
>
> Please kindly refer to **Fig 4, Fig 9, and Fig 10.**
>
> - For example, in the VCR dataset, a raw image is provided along with two bounding boxes for two people labeled as `man_1` and`woman_1`(**Figure 3**). The corresponding question for the image is
>
>     > "Are the [man_1] and [woman_1] quarreling ?"
>     >
>
>     To process this, we first need to crop the sub-images from the raw image based on the bounding boxes. Finally, replacing `man_1` and`woman_1` with cropped images of them will result in an instance of our dataset.
>
> - We have **open-sourced a tool** that can be used to transform new datasets in accordance with our proposed context scheme at the lowest cost. You can create MIC data through an **automatic pipeline**.
>
> We have made modifications to our paper to better highlight our contributions and cost-effectiveness in data collection.

---

> > ### Author Response · Authors · 2023-11-21
> >
> > Dear Reviewer eWon,
> >
> > We appreciate your constructive feedback and observations! Kindly inform us if you find our response acceptable, and we'll be eager to address any additional concerns you might still harbor.
> >
> > Sincerely,
> >
> > Paper4878 Authors

---

> > > ### Comment · Reviewer_eWon · 2023-11-21
> > >
> > > Thank you for the detailed response -- your response is extremely long so it took time for me to look over both the comments and the updated paper.
> > >
> > > I think the authors are dedicated to answer my questions throughly and used reviewers' feedbacks to improve their paper. Here are some of my remaining comments.
> > >
> > > 1. I had misunderstood the contribution of this paper -- I thought the authors spent human efforts to curate a novel pretraining dataset, but it seems like the only effort (quoted from author response) is to design a single instruction template for each task. I think this weakens the contribution of the paper, as most of the heavy lifting are done by the original dataset curators such as VCR that provides both comprehensive bounding boxes and entity relations. While the author claims that the key insight is the **novel context scheme**, I saw no ablation on different ways to construct such multimodal in-context datasets. This weakens the scientific value of the paper.
> > > 2. I don't find the current ablation on selection of training data to be satisfactory. The current results only show that **using all data** will lead to the best performance, which is not particularly insightful. I think detailed ablations must be done to answer **what kinds of interleaved image-text data are most helpful, and for what tasks**. For example, is the VCR-based multimodal in-context data crucial for the downstream performance (and on what tasks)? I think this kind of ablation will have more scientific value to the community and help readers understand why the proposed method is effective.
> > > 3. I am also not satisfied with the attached few-shot in-context learning ablation. First, I believe the authors should have done this ablation for all datasets and provide std in main paper. And will different numbers of shot affect the results?
> > > 4. Training details for stage-1 vs. stage-2 is still not clear. I didn't find any details regarding to datasets used in Appendix G (as referred to by the authors), which currently seems to be discussing template construction. Also, I am confused because BLIP-2 did not use LAION400M during training, as they used BLIPv1-generated synthetic captions on a subset of LAION (~114M).
> > >
> > > One writing comment:
> > > 1. You should clarify that the Winoground evaluation of MMICL is different from BLIPv2's result, which uses 1 single (image, text) pair without looking at both images and both texts.
> > >
> > >
> > > In conclusion, I think the current paper is mostly **engineering** efforts (including prompt engineering), and do not provide enough **scientific** ablations. I also agree with Reviewer 7yvn's comments. Given that the model already shows impressive performance across benchmarking tasks, I would strongly encourage the authors to explain what are the key insights of the dataset/prompt design, and perform proper ablations to justify your claims. Even if the real key insight is just to aggregate more training data, I think the paper could still benefit from ablating the selection of training data and show their performance impact on different types of downstream tasks.

---

> ### Author Response · Authors · 2023-11-22
> **Response to Reviewer eWon (Q1)**
>
> Thanks for your constructive reviews.
>
> Regarding the questions you raised, we are willing to provide the following clarifications:
> ### Q1
>
> > I had misunderstood the contribution of this paper -- I thought the authors spent human efforts to curate a novel pretraining dataset, but it seems like the only effort (quoted from author response) is to design a single instruction template for each task. I think this weakens the contribution of the paper, as most of the heavy lifting are done by the original dataset curators such as VCR that provides both comprehensive bounding boxes and entity relations. While the author claims that the key insight is the novel context scheme, I saw no ablation on different ways to construct such multi-modal in-context datasets. This weakens the scientific value of the paper.
>
> ### A1:
>
> - The **contribution of the paper**
>
> Our **initial insight is quite straightforward**. We believe that the ability of VLMs to understand complex multi-modal contexts (e.g., multi-image reasoning, text-to-image reference) could emerge from the ability to learn from multi-modal demonstrations in context (ICL). Since ICL requires the model to refer to the examples in context to solve the current question, which is quite similar to the complex multi-modal context scenarios.
>
> Nonetheless, earlier research, including Flamingo, LLava, and Otter, either fails to encompass multiple images or lacks the capacity to manage the relationship between multiple images. Consequently, the **novelty of our paper** is in putting forth a method that addresses several critical shortcomings of present VLMs when dealing with intricate multi-modal context scenarios. This includes but is not limited to, image-text references, multi-image comprehension, and multi-modal ICL capabilities.
>
> The **context scheme** you referenced is, in fact, the approach we have advocated for. We have **reduced the burden of manual data collection**, compared to Llava and Otter, and instead, integrated our context scheme into existing open-source datasets to create the MIC dataset. This dataset spans 16 different datasets and covers 8 tasks. **We've added more datasets to our MIC dataset, which is now publicly available.** We have fully exploited each dataset's unique characteristics to generate an ICL format that aligns with our requirements. For instance, the Visual Commonsense Reasoning (VCR) dataset, provides both comprehensive bounding boxes and entity relations.
>
> Compared to prior works that **performed multi-modal instruction tuning with existing datasets**, the multi-modal contexts we have proposed can evidently address the aforementioned limitations and are better suited to real-world scenarios.
>
> In conclusion, we believe that our research has significantly expanded the community's comprehension of multi-modal ICL.

---

> > ### Author Response · Authors · 2023-11-22
> > **Response to Reviewer eWon (Q2)**
> >
> > ### Q2:
> >
> > > I don't find the current ablation on selection of training data to be satisfactory. The current results only show that using all data will lead to the best performance, which is not particularly insightful. I think detailed ablations must be done to answer what kinds of interleaved image-text data are most helpful, and for what tasks. For example, is the VCR-based multi-modal in-context data crucial for the downstream performance (and on what tasks)? I think this kind of ablation will have more scientific value to the community and help readers understand why the proposed method is effective.
> > >
> >
> > ### A2:
> >
> > Thank you for your comprehensive review. We have carried out additional ablation studies, particularly focusing on the selection of training data, aiming at pinpointing the exact source driving improvements in certain capabilities of our model.
> >
> > First, we would like to clarify the settings for our ablation study and assign an appropriate designation for each:
> >
> > 1. MMICL (Instructblip-xl): This is the standard model.
> > 2. **w/o context scheme**: This model is trained only with instructions using all datasets from stage 2 of MMICL, without any additional design in context format and model architecture. It aims to ablate the contribution of the multi-odal instruction tuning.
> > 3. **w/o image declaration**: This model is trained without image declaration but includes multi-modal data with interconnected images and in-context data. It utilizes all datasets used in stage 2 of MMICL and intends to evaluate the impact of image declaration design.
> > 4. **w/o in-context format**: This model, devoid of in-context format data, comprises multi-modal data with related images and image declaration. It uses all datasets found in stage 2 of MMICL with a single image-text pair and aims to assess the effectiveness of multi-modal in-context data design.
> > 5. **w/o interrelated images**: This version of the model is trained without any interconnected images. Its training pipeline contains multi-modal in-context data and image declaration. It utilizes all datasets used in stage 2 of MMICL, excluding the video/VCR datasets. It aims to evaluate the contribution of the design which incorporates multi-modal data with interrelated images
> >
> > We conduct ablation experiments with equal data amounts used in stage two of MMICL, employing the Instructblip-xl as the backbone model for these ablation models. The ablation study aims to perform accurate justifications of our designs based on our initial insights.
> >
> > | Task | MMICL | w/o context scheme | w/o image declaration | w/o in-context format | w/o interrelated images |
> > | --- | --- | --- | --- | --- | --- |
> > | MME_Perception | 1303.59 | 1238.99 | 1170.87 | 1141.02 | 1207.70 |
> > | MME_Cognition | 370.71 | 316.79 | 341.07 | 345.36 | 333.21 |
> > | Icon-QA | 58.12 | 52.80 | 47.15 | 51.95 | 54.35 |
> > | NLVR2 | 72.45 | 56.65 | 61.0 | 62.63 | 59.60 |
> > | Raven | 32.00 | 8.00 | 18.00 | 28.00 | 16.00 |
> >
> > | Winoground | Image-score | Text-score | Group-score |
> > | --- | --- | --- | --- |
> > | MMICL(Instructblip-xl) | 46.00 | 39.25 | 38.75 |
> > | w/o context scheme | 12.25 | 17.25 | 6.00 |
> > | w/o image declaration | 4.00 | 22.25 | 3.00 |
> > | w/o in-context format | 31.25 | 25.75 | 20.00 |
> > | w/o interrelated images | 36.50 | 31.25 | 25.75 |

---

> ### Author Response · Authors · 2023-11-22
> **Continue Response**
>
> Findings from the ablation study reveal:
>
> The superiority of MMICL is **driven by the collective impact of our design elements**—removing any component cannot guarantee the superior performance of our model. Each component of our design significantly **contributes to different aspects** of our model.
>
> 1. **Ablation of Context Scheme:**
>     1. For the training datasets, we followed the similar dataset selection to the Ying-VLM and Multiinstruct. Our improved results demonstrate the effectiveness of our **context scheme design** rather than merely aggregating more pretraining data into the pipeline.
>     2. The substantial improvements across all tasks when compared to the model **w/o context scheme** further substantiate its effectiveness. Lacking the context scheme and our model architecture supporting the scheme, the model performed **poorly across most datasets,** except performance on the single image task focusing on specific object perception (MME_Perception, such as Existence). This is presumably due to the alignment of the additional single image-text pair training with the task's evaluation format.
> 2. **Ablation of Interrelated Images:**
>     1. The introduction of video/VCR data was critical to the scheme design of the **multi-modal data with interrelated images**. We remove this scheme design as well as corresponding datasets to evaluate its contribution. The exclusion of this design leads to a significant **decrease in multi-image understanding** ability, evidenced in the **NLVR2 and Raven** datasets.
>     2. However, the **in-context training pipeline** still enabled it to **retain** some multi-image understanding ability, retaining an advantage over datasets with **multiple images (Icon-QA, NLVR2, Raven**) compared to the **Model w/o context scheme.**
>     3. At the same time, the design of ICL data still improves the model on single image understanding over the **model w/o in-context format**, particularly in understanding the task about specific object perception (MME_Perception), but performed poorly in more abstract tasks involving recognition and cognition(MME_Cognition).
>     4. When compared to the model that lacks in-context format, it demonstrates a superior understanding of paired image and text data, evident in the **model w/o interrelated images** surpasses the model **w/o in-context format** across all scores in Winoground. This suggests that multi-modal data with interrelated images provide some assistance in understanding multiple images. However, without the help of multi-modal ICL data, it will not display superior performance. It aligns with our initial insight.
> 3. **Ablation of In-context Format:**
>     1. Excluding this design revealed that even with the introduction of extra datasets, performance improvement on the single image tasks was not achieved. Compared to other ablation settings, despite the **w/o in-context format** receiving a noticeable improvement in tasks related to comprehension and inference(MME_Cognition) compared to other ablation settings, we observed a significant drop in the task about specific object perception (MME_Perception).
>     2. Training with multiple images enhanced its ability to better understand abstract logic information within a single image and to perform better in downstream task(MME_Cognition). However, the model lost its ability to analyze and recognize specific objects in a single image, leading to a significant decrease in the perception score of MME. The performance of the **w/o image declaration** on MME also provides evidence for that.
>     3. We found out that, by benefiting from learning on interrelated image data, its understanding ability on multiple images has not significantly declined, exhibiting its superiority over other settings. The performance in the MME also confirms this, indicating that multi-modal icl data offers significant advantages for comprehending multiple images and enhancing the model's ability to understand abstract information in the signal image.
> 4. **Ablation of Image Declaration:**
>     1. The removal of image declaration led to a stark decline in performance in understanding complex image-text relationships (Winoground) due to a lack of explicit modeling between image-text references.
>     2. This phenomenon was also observed in tasks that involved understanding multiple image relations(Raven, Iconqa, NLVR2). However, it can be observed lacking interrelated images harms the model's multi-image understanding more than missing image declarations, given the relatively better performance of the model **w/o image declarations.**
>     3. This indicates the importance of image declaration in handling the multi-image and image-text relationships.
>
> To make our work more comprehensive, we plan to delve deeper into the impacts of removing different datasets from various tasks (such as all the QA datasets). We comprehend that this task demands a lot of resources, but we believe it would provide valuable insights.

---

> ### Author Response · Authors · 2023-11-22
> **Response to Reviewer eWon (Q3-Q5)**
>
> ### Q3:
>
> > I am also not satisfied with the attached few-shot in-context learning ablation. First, I believe the authors should have done this ablation for all datasets and provide std in main paper. And will different numbers of shot affect the results?
> >
>
> ### A3:
>
> We acknowledge your desire for a comprehensive study showing the steady performance of the few-shot sampling strategy across all datasets. Nevertheless, due to time constraints, we were unable to perform this ablation study for all datasets and model types in a timely manner. Besides the VQAv2 experiment we provide above, we have **provided the ablation study of other datasets in Table 4,** and we aim to include additional ablation results of the std in the camera-ready version of the paper.
>
> It's important to note, however, that for most models, variations in the number and types of examples could have varying impacts on downstream tasks, and some level of performance fluctuation is to be expected. Therefore, we've chosen to report the experimental results using a random-sample strategy to create in-context examples, which does not significantly affect our model's performance. Our experiments on different datasets indicate that this approach yields steady performance.
>
> Additionally, it can be observed that for the dataset without a fixed format (Flickr, Websrc), their performance stability is worse than others (Hatefulmeme).
>
> | seed | Flickr | Websrc | VQAv2 | Hatefulmeme |
> | --- | --- | --- | --- | --- |
> | 1024 | 88.00 | 19.25 | 69.98 | 64.60 |
> | 4096 | 88.49 | 19.85 | 69.94 | 64.47 |
> | 16384 | 87.21 | 19.65 | 70.21 | 64.58 |
> | std | **0.5289** | **0.2494** | **0.1186** | **0.0565** |
>
> Your observation is indeed correct. The **number of examples** or samples used for training the model surely **affects** model performance.
>
> In Table 27 in **Appendix P**, MMICL further enhances its understanding of **Minecraft objects with the inclusion of ICL examples**. With 4-shot exemplars, MMICL demonstrates an **8.6-point improvement over the model without ICL examples**. This improvement is **more pronounced with increased examples and model size**, resulting in a **10-point** increase compared to the model without ICL examples.
>
> Specifically, we take the  Vizwiz datasets as an example with different numbers of examples, we notice a distinctive upward trend. As the number of **samples increases**, the model's performance **improves**. And when the number of icl examples **increases to 12-shot**, the performance of MMICL **begins to decrease**.
>
> Moreover, it's observable that multi-modal in-context learning significantly aids tasks with a fixed format. The performance of MMICL on Vizwiz increased significantly even with a single example. This enhancement is derived from both the output format and the prediction accuracy. On the other hand, for caption tasks like nocaps which are still significant, the level of improvements received from multi-modal ICL is much lower compared to VizWiz.
>
> |  | 0 | 1 | 2 | 4 | 8 | 12 |
> | --- | --- | --- | --- | --- | --- | --- |
> | Vizwiz | 24.45 | 42.01 | 50.11 | 50.28 | **51.02** | 50.27 |
> | Nocaps | 98.00 | 99.92 | 103.04 | 103.36 | **105.23** | 103.8 |
>
> ### Q4:
>
> > Training details for stage-1 vs. stage-2 is still not clear. I didn't find any details regarding to datasets used in Appendix G (as referred to by the authors), which currently seems to be discussing template construction. Also, I am confused because BLIP-2 did not use LAION400M during training, as they used BLIPv1-generated synthetic captions on a subset of LAION (~114M).
> >
>
> ### A4:
>
> We acknowledge the confusion caused by the reference to dataset details in Appendix G. The original reference is Appendix G.
>
> Because we revised the paper and added a new appendix, it made the original **Appendix G** become **Appendix H.**
>
> In response to your confusion about dataset usage, since the BLIP-2 paper uses the  "115M images from the LAION400M dataset (Schuhmann et al., 2021)" and "captions using the BLIPlarge captioning model." So we noted it as LAION-400M, we will provide clarity on the dataset utilization in our revised paper.
>
> ### Q5:
>
> > One writing comment:
> >
> > 1. You should clarify that the Winoground evaluation of MMICL is different from BLIPv2's result, which uses 1 single (image, text) pair without looking at both images and both texts.
>
> ### A5:
>
> We appreciate your suggestion. We have **addressed this concern** in the revised paper that BLIPv2 employs a single (image, text) pair without considering both images and both texts for a fair comparison. This clarification has been incorporated to ensure transparency in the evaluation methodology.
>
> We believe that the revised version will be more clear and sound. We trust that this additional scientific ablation addresses your concerns and justifies our claims.
> We also hope that you can kindly increase your score if our response has helped address your concerns.

---

### Official Review · Reviewer_QUmJ · 2023-10-31

**Soundness:** 3 good
**Presentation:** 3 good
**Contribution:** 3 good
**Rating:** 8
**Confidence:** 4

**Summary:**

The author proposed a new kind of in-context learning, called multi-modal in-context learning, to deal with multi-modal inputs. To train model, the author proposed a novel context scheme and constructed the Multi-modal In-Context Learning (MIC) dataset. Experiments are conducted to show the advantages on several well known datasets.

**Strengths:**

1. The targeting problem is interesting.

2. Improvements are achieved on several well-known datasets.

3. Extensive experimental results are provided.

**Weaknesses:**

1. It seems that the explicit meaning of MMICL is not mentioned in this paper. Some explanations might be helpful.

2. The experiments are conducted on T5 family models, which are encoder-decoder architecture models. Why not use the decoder-only models?

3. Model sizes should be included in some tables. Otherwise, it is not easy to compare the competing models.

4. It seems that bigger model does not always have better performance. Is there any explanations for that?

**Questions:**

See the above section.

**Details Of Ethics Concerns:**

no such concerns.

---

> ### Author Response · Authors · 2023-11-19
> **Respond to the valuable advice and comments from Reviewer QUmJ**
>
> We feel very excited to receive such a high rating from you for our paper. It makes us feel that our efforts are worthwhile! It is encouraging to see you find our work targeting an interesting problem.
>
> Regarding your concerns:
>
> ### **Q1:**
>
> > It seems that the explicit meaning of MMICL is not mentioned in this paper. Some explanations might be helpful.
>
>
>
> ### **A1** :
>
> - We sincerely appreciate your thoughtful feedback. In this paper, the term MMICL stands for "vision-language **M**odel with **M**ulti-modal **I**n-**C**ontext **L**earning"
>
> The overarching goal of MMICL is to empower Vision-Language Models (VLMs) by incorporating multi-modal in-context learning. This approach aims to enhance the model's ability to understand and generate responses in complex, multi-modal scenarios.
>
> We appreciate your observation, and we have included the explicit meaning of MMICL in the revised paper.
>
>
> ### **Q2**:
>
> > The experiments are conducted on T5 family models, which are encoder-decoder architecture models. Why not use the decoder-only models?
>
> ### **A2** :
>
> - Thank you for your inquiry. We selected the T5 family models as the backbone for our experiments. At the time of conducting this study, some decoder-only models, like Vicuna, had not been released. Consequently, we focused our experiments on the Flan backbone model.
> - Since then, we **have developed and released** the **Vicuna-based MMICL model**.
>
>     We believe that incorporating experiments on the Vicuna-based MMICL model further highlights the novelty and robustness of our proposed method. Later, we will add the evaluation results of the Vicuna-based MMICL model in the final version.
>
>
> If you have any additional questions or specific areas you would like further details on, please feel free to let us know.
>
>
> ### **Q3:**
>
> > Model sizes should be included in some tables. Otherwise, it is not easy to compare the competing models.
>
> ### **A3**:
>
> - Thank you for your insightful observation. Including the model size in the tables makes comparing the competing models easier. We have duly noted your suggestion and taken immediate action to address this concern.
> - In the revised version, we have **included the model sizes** alongside each corresponding model in **Table 1 and Table 5**.
>
> We appreciate your diligence in reviewing our work and providing constructive feedback. Your advice improves our work! Thank you!
>
> ### **Q4:**
>
> > It seems that bigger model does not always have better performance. Is there any explanations for that?
>
> ### **A4** :
>
> We appreciate your inquiry and are pleased to provide clarification on this phenomenon.
>
> - The phenomenon of **smaller models outperforming larger counterparts** in certain tasks is not unique to MMICL; it has been observed **across various VLMs** [1] [2].
>
>     For instance, the xl version of Blip2 demonstrates superior performance compared to the larger XXL version across **multiple datasets** such as Nocaps, Iconqa, and VizWiz [1].
>
> - Our method, trained on the **basis of Blip2 and InstructBlip**, appears to **inherit** this characteristic.
> - **Similar trends** are evident in other models like **Flamingo**, where the smaller version exhibits more advanced few-shot abilities than the larger model **across diverse datasets** such as STAR, FLICKR, and iVQA [3].
>
> Notably, this phenomenon is particularly observable in tasks such as image captioning and open-domain question answering.
>
> The nuanced dynamics between model size and performance represent an intriguing area for exploration. In our future work, we plan to delve deeper into this phenomenon, conducting a more comprehensive analysis to unravel the underlying factors.
>
> We appreciate your thoughtful consideration of this aspect, and your feedback continues to enrich our understanding and future research directions.
>
> [1] Liu, H., Li, C., Wu, Q., & Lee, Y. J. (2023). Visual instruction tuning. *arXiv preprint arXiv:2304.08485*.
>
> [2] Yu, W., Yang, Z., Li, L., Wang, J., Lin, K., Liu, Z., ... & Wang, L. (2023). Mm-vet: Evaluating large multimodal models for integrated capabilities. *arXiv preprint arXiv:2308.02490*.
>
> [3]Alayrac, J. B., Donahue, J., Luc, P., Miech, A., Barr, I., Hasson, Y., ... & Simonyan, K. (2022). Flamingo: a visual language model for few-shot learning. *Advances in Neural Information Processing Systems*, *35*, 23716-23736.

---

> > ### Author Response · Authors · 2023-11-21
> >
> > Dear Reviewer QUmJ,
> >
> > Thank you for your valuable feedback and comments! We appreciate your recognition of our work. Please let us know if you are satisfied with our response, and we will be happy to address any remaining concerns you may have.
> >
> > Sincerely,
> >
> > Paper4878 Authors

---

### Official Review · Reviewer_d1dY · 2023-11-01

**Soundness:** 3 good
**Presentation:** 2 fair
**Contribution:** 2 fair
**Rating:** 6
**Confidence:** 5

**Summary:**

The paper introduces a context learning scheme for multimodal large language models and constructs a instruction dataset. It demonstrates the model's comprehension ability when dealing with complex multi-modal prompts and uncovers its potential for in-context learning. Overall, this work leans more towards a technical report, but it is comprehensive.

**Strengths:**

1. Compared to open-/flamingo, kosmos, the model can more easily handle multiple image inputs and their relationships.

2. Through prompt engineering, it unifies more tasks.

3. More instruction tuning data has been obtained through chatgpt.

**Weaknesses:**

The synthetic dataset, training paradigms, and model structure have not undergone an ablation study, so the specific gain is unclear.

This work has expanded further on the data level compared to works such as InstrutBlip and LLaVA, and has achieved better results, which I am not surprised by. However, its design is slightly complex, and it contains a lot of prompt engineering. I believe this work is above the borderline, but it may not be classified as a 'good' paper here.

**Questions:**

See weakness.

---

> ### Author Response · Authors · 2023-11-19
> **Respond to the valuable advice and comments from Reviewer d1dY (1/N)**
>
> Thanks for your constructive reviews. It is encouraging to see you find our work to be comprehensive.
> Below, we provide detailed replies to your comments and hope we can resolve your major concerns.
>
> ### **Q1:**
>
> > The synthetic dataset, training paradigms, and model structure have not undergone an ablation study, so the specific gain is unclear.
>
> ### **A1 :**
>
> Thank you for your insightful comment, and we genuinely appreciate your constructive feedback.
>
> We commit to providing a clearer explanation of ablation the present in our paper.
>
> - **In Section 3.6**, we have **performed an ablation study on training paradigms** to assess the impact of multi-modal in-context tuning. The results, as depicted in Table 6, showcase significant improvements **across all types and sizes of models**. Notably, tasks involving multiple images, such as IconQA-img and Bongard-HOI, demonstrate substantial performance gains (15.75 and 21.05 points improvement, respectively). This underscores the effectiveness of our proposed ICL tuning in handling complex multi-modal prompts and accomplishing challenging tasks.
> - In addition to the training paradigms, we have extended our ablation study to **include training data and model structure**.
>     - Specifically, we conduct ablation experiments using the Instructblip-xl as the backbone model in the following settings to verify the effectiveness of **our proposed interleaved image-text data pairs and in-context learning data:**
>        - **MMICL**: This is the standard model.
>        - **w/o context scheme**: This model is trained only with instructions using all datasets from stage 2 of MMICL, without any additional design in context format and model architecture.
>        - **w/o image declaration**: This model is trained without image declaration but includes multi-modal data with interconnected images and in-context data.
>        - **w/o in-context format**: This model, devoid of in-context format data, comprises multi-modal data with related images and image declaration.
>        - **w/o interrelated images**: This version of the model is trained without any interconnected images. Its training pipeline contains multi-modal in-context data and image declaration.
>
>     We use the **equivalent amount** of data as in stage 2 of MMICL for the ablation models.
>
>     - We employ the ablation study on the **general benchmark(MME)** and **multi-image reasoning datasets(Raven, Icon-QA & NLVR2)** to comprehensively evaluate the sources of the observed performance improvement in MMICL. The ablation results affirm that our **interleaved image-text data scheme is a key factor in** observed improvements, clarifying its contribution to MMICL's performance. The superiority of MMICL is driven by the collective impact of our design elements—removing any component cannot guarantee the superior performance of our model. Each component of our design significantly contributes to different aspects of our model.
>
>    | Task | MMICL | w/o context scheme | w/o image declaration | w/o in-context format | w/o interrelated images |
>    | --- | --- | --- | --- | --- | --- |
>    | MME_Perception | **1303.59** | 1238.99 | 1170.87 | 1141.02 | 1207.70 |
>    | MME_Cognition | **370.71** | 316.79 | 341.07 | 345.36 | 333.21 |
>    | Icon-QA | **58.12** | 52.80 | 47.15 | 51.95 | 54.35 |
>    | NLVR2 | **72.45** | 56.65 | 61.0 | 62.63 | 59.60 |
>    | Raven | **32.00** | 8.00 | 18.00 | 28.00 | 16.00 |
>
>     The comprehensive ablation study emphasizes that the observed performance improvement in MMICL is **attributable to our proposed scheme, model structure, and the incorporation of multi-modal ICL tuning.**
>
>     Additional single image-text pair data for  SFT(supervised fine tuning) does not yield comparable enhancements in the targeted areas, underscoring the unique value of our proposed method
>
> - We have **included the ablation experiments** and analysis in the **second section of Sec 3.6** in the revised paper.
>
> We trust that this additional information addresses your concerns, and we welcome any further inquiries or feedback you may have.

---

> ### Author Response · Authors · 2023-11-19
> **Respond to the valuable advice and comments from Reviewer d1dY (2/N)**
>
> ### **Q2:**
>
> > This work has expanded further on the data level compared to works such as InstrutBlip and LLaVA, and has achieved better results, which I am not surprised by. However, its design is slightly complex, and it contains a lot of prompt engineering. I believe this work is above the borderline, but it may not be classified as a 'good' paper here.
>
> ### A2 :
>
> We deeply appreciate your comprehensive feedback and would like to provide a detailed explanation to address your concerns：
>
> - Firstly, it's important to clarify that our work **is not reliant on prompt engineering.** MMICL **consistently demonstrates robust results** across various experiments using a single, standardized prompt:
>
> > "Use the image 0: <image0>{visual token} as the visual aid to help you answer the question accurately.{Question}."
>
>
> We futher evaluate the performance of MMICL in different datasets under different seeds to sample fewshot-examplars. Due to time constraints, we evaluate MMICL on a 2000-instance VQA dataset.
>
> We employed different randomly sampled fewshot-examplars for each instance during evaluation, and it still showcases stable and consistent results. Additionally, it can be observed that for the dataset without a fixed format (Flickr, Websrc), their performance stability is worse than others (Hatefulmeme).
>
> | seed | Flickr | Websrc | VQAv2 | Hatefulmeme |
> | --- | --- | --- | --- | --- |
> | 1024 | 88.00 | 19.25 | 69.98 | 64.60 |
> | 4096 | 88.49 | 19.85 | 69.94 | 64.47 |
> | 16384 | 87.21 | 19.65 | 70.21 | 64.58 |
> | std | **0.5289** | **0.2494** | **0.1186** | **0.0565** |
>
> - Moreover, in the collection of the MIC dataset for ICL tuning, we employed an **automatic pipeline** to effortlessly generate MIC-format data.
>
>     To enhance the diversity of instruction templates in the MIC dataset, we leveraged ChatGPT to **expand instructional formats** based on our handwritten template. To further underscore our commitment to **avoiding prompt engineering**, we want to highlight that the utilization of ChatGPT aimed to **enhance the diversity** of the MIC dataset's instructions. These templates were not employed as prompt engineering tactics to artificially boost our model's performance.
>
>     All the instruction template we used in the collection of the MIC dataset is detailed in **Appendix G**.
>
> - Our primary innovation lies in the introduction of a **novel context scheme**, which incorporates three distinct designs and integrates **multi-modal in-context tuning** to enhance the ICL ability of VLM.
>
>
>     The proposed context scheme contains three designs:
>
>     1.  image declaration, which helps the VLM  to **understand text-to-image reference**. Demonstrates superior performance, surpassing other models by 13 points on Winoground.
>      2. multiple interconnected images format, which helps the VLM to **better understand the relationship between multiple images.** Enables our model to excel in multi-image reasoning, achieving a 12-point improvement on Raven.
>      3. unified multi-modal ICL data format, which helps the VLM to better l**earn from multi-modal in-context demonstrations.** Shows impressive multi-modal ICL performance across various tasks.
>
>     The experiments underscore the success of our proposed context scheme in **effectively tackling the limitations.**
>
>
> - Furthermore, the ablation study above demonstrates that **using additional data to facilitate SFT cannot adequately overcome these limitations**, further asserting the significance and value of our suggested strategy.
> - It is crucial to note that these designs are **not merely instances of prompt engineering**; rather, they **address the limitations** of Vision-Language Models in handling complex multi-modal prompts.
>
>     The experiments, including ablations across synthetic datasets, training paradigms, and model structures, affirm that the observed improvements in MMICL indeed stem from our proposed scheme and Multi-modal ICL tuning.
>
> - Moreover, the ScienceQA-Img experiment results in **Sec. 3.5** showcases MMICL's effectiveness in **mitigating the hallucination issue** inherent in VLMs. By strategically leveraging ICL tuning, MMICL outperforms other models, even in single-image tasks. Intuition: The context format of MMICL during training forces the model to reason across text and multiple images, therefore helping to combat the hallucination issue.
>
> We hope this detailed explanation clarifies the rationale behind our design choices and underscores the significance of our contributions. If you have any further questions or require additional clarification, please feel free to reach out.

---

> > ### Author Response · Authors · 2023-11-21
> >
> > Dear Reviewer d1dY,
> >
> > Thank you once again for your valuable feedback and comments! We would greatly appreciate it if you could inform us of your satisfaction with our response. We are more than willing to address any remaining concerns you may have.
> >
> > Sincerely,
> >
> > Paper4878 Authors

---

> > > ### Author Response · Authors · 2023-11-23
> > >
> > > Dear Reviewer d1dY,
> > >
> > > I hope this message finds you well. Your detailed review and valuable feedback is greatly appreciated, as it has substantially enhanced the quality of our paper.
> > >
> > > Your insightful feedback is immensely valuable to me, and I wish to ensure my responses adequately tackle the problems you have had.
> > >
> > > Furthermore, we hope that you will consider a favourable adjustment to your score if our response has mitigated your concerns.
> > >
> > > Best,
> > >
> > > Authors.

---

### Official Review · Reviewer_iVLP · 2023-11-07

**Soundness:** 4 excellent
**Presentation:** 2 fair
**Contribution:** 3 good
**Rating:** 6
**Confidence:** 4

**Summary:**

The paper explores a pioneering context schema, designed to bolster the VLM's proficiency in effectively handling multimodal inputs and facilitating in-context learning. This method mainly aims to construct the Text-Image Interleaved Dataset (MIC dataset) with the objective of enhancing the VLM's in-context learning capabilities. There are three distinct formats in the MIC dataset, including single image format, multiple interconnected images format, and in-context format, which respectively are tailored for different instances of various vision-language tasks. In experiments, the authors demonstrate that the proposed method (MMICL) outperforms baselines on a wide range of general vision-language tasks, especially on MME and MMBench benchmarks. Moreover, MMICL also shows in-context learning ability on some traditional vision-language tasks in the few-shot setting.

**Strengths:**

+ This paper offers a novel perspective on how to construct image-text interleaved instruction datasets. It highlights how VLM (Vision-Language Model) can simultaneously acquire in-context learning ability during the visual instruction tuning process.
+ The MIC dataset proposed by this paper covers a wide range of vision-language tasks, including Image Captioning, Video Caption, VQA, VideoQA, Visual Reasoning, Visual Dialogue, and Image Classification.
+ The experiments conducted on a wide range of general vision-language tasks are very comprehensive. The average results on the MME and MMBench benchmarks demonstrate the impressive zero-shot performance of MMICL compared with other baselines.

**Weaknesses:**

+ The few-shot performance improvement of MMICL appears to be somewhat limited. For instance, in Table 4, except for the VizWiz dataset, in some cases (e.g., FLAN-T5-XXL and Instruct-FLAN-T5-XL), MMICL does not demonstrate significant improvements on other test datasets when adding in-context examples.
+ The lack of exploration of various interleaved image-text data formats for VLM's in-context learning ability is evident. It is unclear whether only the in-context format (equation 3) can enhance in-context learning ability or if the multiple interconnected images format (equation 2) can also play a certain role.

**Questions:**

+ In Table 4, I observe that MMICL with different backbones exhibits significant performance differences in certain tasks. For instance, the performance gap between MMICL (Instruct-FLAN-T5-XL) and MMICL (Instruct-FLAN-T5-XXL) on Flickr 30k is as large as 34.6. A similar trend can be seen in the VizWiz dataset, with InstructBLIP (Flan-XL) scoring 32.08, while InstructBLIP (Flan-XXL) drops to 15.11. This phenomenon is absent in InstructBLIP. What could be the reason for this?
+ In Table 4, MMICL (FLAN-T5-XL) shows an improvement of approximately 25.13 in the 4-shot setting compared to 0-shot, whereas MMICL (FLAN-T5-XXL) only improves by 3.82. It's perplexing that increasing the backbone's size leads to a significant decline in few-shot performance. What could possibly be the underlying reason for this?
+ How many visual tokens does each image take up? Does the input of multiple images take up a lot of LLM's input context length?

---

> ### Author Response · Authors · 2023-11-19
>
> Thanks for your constructive reviews. It is encouraging to see you find our work offering a novel perspective on enhancing the ICL ability of VLM.
>
> Below, we provide detailed replies to your comments and hope we can resolve your major concerns.
>
> ### **Q1:**
> > The few-shot performance improvement of MMICL appears to be somewhat limited. For instance, in Table 4, except for the VizWiz dataset, in some cases (e.g., FLAN-T5-XXL and Instruct-FLAN-T5-XL), MMICL does not demonstrate significant improvements on other test datasets when adding in-context examples.
> ### **A1 :**
> We appreciate your thorough examination of MMICL's few-shot performance and are committed to addressing your concern to  bolster the validity of our claims.
>
> - Firstly, we sincerely appreciate your pinpointing this issue, enhancing our understanding of MMICL's in-context-learning capabilities.
>
>     Upon further examination, we identified nuances in the evaluation process on the Flickr30k dataset. Specifically, random-sampled in-context-learning examples introduce a prefix and ":" in the answer by using such template:
>
>     > Please provide a description or identification of the subject in image 0:<image0>{embedding}. :: The answer to the question is : {caption} ::
>     >
>     These in-context-learning examples affected MMICL's output in the in-context learning setting, causing it to be more likely to output with a similar prefix and an extra ":".
>
>     - And the abnormal performance of MMICL(Instructblip-xxl) is also due to its repetition of the question at the beginning of the response and the additional ":" at the end of the answer.
>
>      Unfortunately, the standard preprocessing tools were unable to filter out this additional context, which reduces the cider score of the MMICL in both zero-example and 4-example scenarios.
>
>     This means that, despite the fact that MMICL has **actually improved through in-context learning**, inaccurate cider score calculations instead result in an abnormal performance decrease for MMICL.
>
>     Therefore, by recalculating the model outputs with the appropriate preprocessing and filtering out extraneous information, the **abnormal** performance of MMICL **disappears** and the MMICL despite **significant in-context learning ability.**
>
>     The revised evaluation results, shown in the table below, depict the accurate performance of MMICL on this dataset.
>
>     | Model | Flickr 30K | WebSRC | VQAv2 | Hateful Memes | VizWiz |
>     | --- | --- | --- | --- | --- | --- |
>     | MMICL(FLAN-T5-XL) (Zero-Shot) | **83.47** | 12.55 | 62.17 | 60.28 | 25.04 |
>     | MMICL(FLAN-T5-XL) (4-Shot) | **83.84** | 12.30 | 62.63 | 60.80 | 50.17 |
>     | MMICL(FLAN-T5-XXL) (Zero-Shot) | **85.03** | 18.85 | 69.99 | 60.32 | 29.34 |
>     | MMICL(FLAN-T5-XXL) (4-Shot) | **89.27** | 18.70 | 69.83 | 61.12 | 33.16 |
>     | MMICL(Instruct-FLAN-T5-XL) (Zero-Shot) | **82.68** | 14.75 | 69.13 | 61.12 | 29.92 |
>     | MMICL(Instruct-FLAN-T5-XL) (4-Shot) | **88.31** | 14.80 | 69.16 | 61.12 | 33.16 |
>     | MMICL(Instruct-FLAN-T5-XXL) (Zero-Shot) | **73.97** | 17.05 | 70.30 | 62.23 | 24.45 |
>     | MMICL(Instruct-FLAN-T5-XXL) (4-Shot) | **88.79** | 19.65 | 70.56 | 64.60 | 50.28 |
>
>     Notably, MMICL's performance on the Flickr30k dataset improves significantly with in-context-learning examples, as demonstrated by the revised scores.
>
>     We have also **revised the Table 4** in the revised paper. We sincerely appreciate your careful review and feedback. Your feedback has made our paper better.
>
> - Secondly, MMICL exhibits substantial zero-shot performance improvements compared to baseline methods (Kosmos-1 and Flamingo) across various datasets.
>
>     The reason why the inclusion of examples in MMICL does not lead to a two-digit improvement is that in scenarios where performance is **already high**, the assistance provided by these examples **is quite limited**.
>
> - **Multi-Modal ICL Capability in Out-of-Domain Scenarios:**
>
>     Apart from the experiments in Sec. 3.6, **Table 25 in Appendix P** highlights MMICL's impressive generalization to an unseen domain in **Minecraft**, even if the images are extremely different compared to the images used by training. MMICL further enhances its understanding of **Minecraft objects with the inclusion of ICL examples**. With 4-shot exemplars, MMICL demonstrates an **8.6-point improvement over the model without ICL examples**. This improvement is **more pronounced with increased examples and model size**, resulting in a 10-point increase compared to the model without ICL examples. The nuanced performance gains in the few-shot setting demonstrate the adaptability and generalization capabilities of MMICL.
>
>
> We hope this detailed explanation successfully addresses your concern. Your advice and scrutiny have significantly contributed to the quality of our paper, and we are sincerely grateful for your time and expertise.

---

> ### Author Response · Authors · 2023-11-19
>
> ### **Q2:**
>
> > The lack of exploration of various interleaved image-text data formats for VLM's in-context learning ability is evident. It is unclear whether only the in-context format (equation 3) can enhance in-context learning ability or if the multiple interconnected images format (equation 2) can also play a certain role.
>
> ### **A2 :**
>
> Thank you for your insightful comment and we genuinely appreciate your constructive feedback.
>
> We commit to providing a clearer explanation of ablation the present in our paper.
>
> - **Extended Ablation Study:**
>
>     To address your concern, we have extended our **ablation study** using Instructblip-xl as the backbone model to investigate how the multiple **interconnected image format** (equation 2) and the **in-context format (equation 3)** contribute to the in-context learning (ICL) ability of MMICL.
>
> - **Ablation Models:**
>
> We sample 2000 instances of Vizwiz and 1000 instances of Flickr for the **ablation study.**
>
> - **MMICL**: This is the standard model.
> - **w/o context scheme**: This model is trained only with instructions using all datasets from stage 2 of MMICL, without any additional design in context format and model architecture.
> - **w/o in-context format**: This model, devoid of in-context format data, comprises multi-modal data with related images and image declaration.
> - **w/o interrelated images**: This version of the model is trained without any interconnected images. Its training pipeline contains multi-modal in-context data and image declaration.
>
> We use the **equivalent amount** of data as in stage 2 of MMICL for the ablation models.
>
> - **Comprehensive Evaluation:**
>
>     We employ the ablation study on both Vizwiz and Flickr under the five and zero of ICL examples. We also evaluate the zero-shot three different settings on the **general benchmark(MME)** and **multi-image reasoning datasets(Icon-QA & NLVR2)** to comprehensively evaluate the sources of the observed performance improvement in MMICL.
>
> - **Results and Conclusions:**
>     - The ablation results affirm that our **context scheme**, which **combines** image declaration **(eq1)**, interconnected images **(eq2)**, and ICL examples **(eq3)**, is a **key factor** in the observed improvements, providing clarity on **its contribution to MMICL's in-context learning ability**.
>
>         It can be conclude that the magic of MMICL on **enhancing ICL ability is a combination of eq1&2&3**, rather than just eq3.
>
>     - Notably, ICL-only achieves the most significant improvements compared to the other two settings in the Vizwiz dataset. This highlights that tuning with in-context format data effectively enhances the model's understanding of in-context samples. The relatively lower improvement on the Flickr dataset indicates that, in the absence of multi-image instruction tuning, the model struggles to effectively utilize in-context examples to improve its performance on tasks without fixed output formats, such as image captioning.
> - **Unique Value of Our Proposed Method:**
>     - The additional evaluation of zero-example experiments on the general benchmark and multi-image datasets further **validates the superior performance** of MMICL, which stems from the **combination of eq1, eq2, and eq3**.
>     - Our designs of the context scheme together help the VLM to show this superior performance. Meanwhile, the additional single image-text pair data for SFT(supervised fine tuning) does not yield comparable enhancements in the targeted areas, underscoring the unique value of our proposed method.
>
> | Task | ICL example | MMICL | w/o context scheme | w/o in-context format | w/o interrelated images |
> | --- | --- | --- | --- | --- | --- |
> | VIZWIZ | zero-example | 27.10 | 28.13 | 15.90 | 23.43 |
> |  | 5-example | 42.66 | 36.23 | 28.33 | 37.13 |
> |  | Δ | **15.56** | 8.1 | 12.43 | 13.7 |
> | Flickr | zero-example | 83.94 | 77.84 | 65.33 | 76.88 |
> |  | 5-example | 88.11 | 79.38 | 65.95 | 77.09 |
> |  | Δ | **4.17** | 1.54 | 0.62 | 0.18 |
> | MME_Perception(Overall) | zero-example | 1303.59 | 1238.99 | 1141.02 | 1207.70 |
> | MME_Cognition (Overall) | zero-example | 370.71 | 316.79 | 345.36 | 333.21 |
> | Icon-QA | zero-example | 58.12 | 52.80 | 51.95 | 54.35 |
> | NLVR2 | zero-example | 72.45 | 56.65 | 62.63 | 59.60 |
> | Raven | zero-example | 32.00 | 8.00 | 28.00 | 16.00 |
>
> We trust that this additional information addresses your concerns, and we welcome any further inquiries or feedback you may have. We hope that our response has addressed your concerns.

---

> > ### Author Response · Authors · 2023-11-19
> >
> > ### **Q3:**
> >
> > > In Table 4, I observe that MMICL with different backbones exhibits **significant performance differences** in certain tasks. For instance, the performance gap between MMICL (Instruct-FLAN-T5-XL) and MMICL (Instruct-FLAN-T5-XXL) on Flickr 30k is as large as 34.6. A similar trend can be seen in the VizWiz dataset, with InstructBLIP (Flan-XL) scoring 32.08, while InstructBLIP (Flan-XXL) drops to **15.11**. This phenomenon is absent in InstructBLIP. What could be the reason for this?
> >
> > ### **A3 :**
> >
> > We appreciate your meticulous observation and welcome the opportunity to provide a detailed explanation and analysis of this phenomenon.
> >
> > 1. **VizWiz Dataset Performance Drop**:
> >
> >     The observed performance drop in the VizWiz dataset can be attributed to our use of **VQA-accuracy** as the metric, which relies on the **exact match** for all possible answers.
> >
> >     To ensure a fair comparison, we employed the same instruction when querying the model: Use the image as the visual aid to answer the question accurately.
> >
> >     And we observed that InstructBLIP (Flan-XXL) sometimes tends to output longer responses than the candidate answers, leading to **a failure in the exact match** and subsequent **score reduction**.
> >
> >     However, by including instances that were indeed correct but not considered due to the exact match calculation strategy, we observed a clear performance improvement.
> >
> >     Specifically, instead of using the exact match mechanism to calculate potential answers in the VQA metric, we utilized partial matching, which **scores the prediction if the candidate answers are present in the prediction result** for the VQA metric.
> >
> >     Regarding the anomalous **15.11** score of InstructBLIP (Flan-XXL), after employing **partial matching** for the VQA metric, we achieved an improved score of **38.54**. The performance drop **no longer persists**.
> >
> >     We have **marked this abnormal result in Table 4 and provided the reason** in the revised paper.
> >
> >     - For your reference, we provide results on the VizWiz dataset with this new metric in the following table:
> >
> >
> >         | Model | vizwiz |
> >         | --- | --- |
> >         | Blip2-xl | 0.3743 |
> >         | Blip2-xxl | 0.3588 |
> >         | Instructblip-xl | 0.3680 |
> >         | Instructblip-xxl | 0.3854 |
> >         | MMICL(Flant5-xl)-zeroshot | 0.3086 |
> >         | MMICL(Flant5-xl)-fewshot | **0.5765** |
> >         | MMICL(Flant5-xxl)-zeroshot | 0.4506 |
> >         | MMICL(Flant5-xxl)-fewshot | **0.5007** |
> >         | MMICL(Instruct-Flant5-xl)-zeroshot | 0.4502 |
> >         | MMICL(Instruct-Flant5-xl)-fewshot | **0.4671** |
> >         | MMICL(Instruct-Flant5-xxl)-zeroshot | 0.3948 |
> >         | MMICL(Instruct-Flant5-xxl)-fewshot | **0.6125** |
> > 2. **Performance Decline on Flickr**
> >
> >     After a detailed analysis of the model prediction preprocessing and subsequent modifications to Table 4, the performance decline in the Flickr dataset **is no longer present**.
> >
> >     Following a thorough review of the model prediction preprocessing and subsequent modifications to Table 4, the performance decline in the Flickr dataset **is no longer present**.
> >
> >     Specifically, we manually removed the additional prefix of each model output, retained only the image caption provided by the model, deleted the ":" separator at the end of each sentence, and added ":" to the Stop Words list to enable the preprocessing tool to filter such abnormal patterns.
> >
> >     The processed model output was then sent for cider score calculation.
> >
> >     As a result, we have **updated Table 4 in the revised paper accordingly.**
> >
> >
> > We sincerely appreciate your careful review and valuable feedback, which have significantly contributed to the improvement of our paper.

---

> > > ### Author Response · Authors · 2023-11-19
> > >
> > > ### **Q4:**
> > >
> > > > In Table 4, MMICL (FLAN-T5-XL) shows an improvement of approximately 25.13 in the 4-shot setting compared to 0-shot, whereas MMICL (FLAN-T5-XXL) only improves by 3.82. It's perplexing that increasing the backbone's size leads to a significant decline in few-shot performance. What could possibly be the underlying reason for this?
> > > >
> > >
> > > ### **A4 :**
> > >
> > > We appreciate your inquiry and are pleased to provide clarification on the performance gap across different model sizes.
> > >
> > > - The phenomenon of **smaller models outperforming larger counterparts** in certain tasks is not unique to MMICL; it has been observed **across various models** [1] [2].
> > >
> > >     For instance, the xl version of Blip2 demonstrates superior performance compared to the larger XXL version across **multiple datasets** such as Nocaps, Iconqa, and VizWiz [1].
> > >
> > > - Our method, trained on the **basis of Blip2 and Instruction-Blip**, appears to **inherit** this characteristic. Our hypothesis is that the "smaller model does better" trend in the zero-shot setting of [1][2] has been transferred to the few-shot setting of MMICL.
> > > - **Similar trends** are evident in other models like **Flamingo**, where the smaller version exhibits more advanced few-shot abilities than the larger model **across diverse datasets** such as STAR, FLICKR, and iVQA [3].
> > >
> > > We posit that, in specific tasks, the superior few-shot performance of smaller models **may be attributed** to their **increased effectiveness in leveraging in-context learning.** Notably, this phenomenon is particularly observable in tasks such as image captioning and open-domain question answering.
> > >
> > > The nuanced dynamics between model size and few-shot capabilities represent an intriguing area for exploration. In our future work, we plan to delve deeper into this phenomenon, conducting a more comprehensive analysis to unravel the underlying factors.
> > >
> > > We appreciate your thoughtful consideration of this aspect, and your feedback continues to enrich our understanding and future research directions.
> > >
> > > [1] Liu, H., Li, C., Wu, Q., & Lee, Y. J. (2023). Visual instruction tuning. *arXiv preprint arXiv:2304.08485*.
> > >
> > > [2] Yu, W., Yang, Z., Li, L., Wang, J., Lin, K., Liu, Z., ... & Wang, L. (2023). Mm-vet: Evaluating large multimodal models for integrated capabilities. *arXiv preprint arXiv:2308.02490*.
> > >
> > > [3]Alayrac, J. B., Donahue, J., Luc, P., Miech, A., Barr, I., Hasson, Y., ... & Simonyan, K. (2022). Flamingo: a visual language model for few-shot learning. *Advances in Neural Information Processing Systems*, *35*, 23716-23736.
> > >
> > > ### **Q5:**
> > >
> > > > How many visual tokens does each image take up? Does the input of multiple images take up a lot of LLM's input context length?
> > >
> > > ### **A5 :**
> > >
> > > We sincerely appreciate your feedback, and we're dedicated to giving a clear explanation.
> > >
> > > - Following the setting of Blip-2, each image o**ccupies 32 visual tokens** in the input context. During MIC tuning， we incorporate up to 8 images per instance. Given that the maximum context of the language model backbone is constrained, the input of multiple images indeed impacts the Long-Short Term Memory (LLM) model's input context length.
> > > - In our paper, we utilize Flant5 as the backbone language model, with a **maximum context length of 512.** However, during inference, we have observed that MMICL exhibits robust generalization beyond the initially specified eight images. For instance, in evaluations on the Bonground-HOI dataset, MMICL achieves remarkable performance even with a **context of 14 images**. Additionally, in the context of video datasets, MMICL maintains a state-of-the-art performance with input sequences of up to **12/16 frames.**
> > > - We attribute this remarkable generalization to the inherent relative position encoding employed by Flant5 in its architecture.

---

> > > > ### Author Response · Authors · 2023-11-21
> > > >
> > > > Dear Reviewer iVLP,
> > > >
> > > > Thank you again for your invaluable insights and observations! We would be immensely grateful if you could share your satisfaction level regarding our response. We will be happy to address any remaining concerns.
> > > >
> > > > Sincerely,
> > > >
> > > > Paper4878 Authors

---

> > > > > ### Comment · Reviewer_iVLP · 2023-11-23
> > > > >
> > > > > Thank you for the comprehensive response. Due to the length of the authors' response, I spent considerable time reviewing all replies, including those to other reviewers.
> > > > >
> > > > > The rebuttal has addressed some of my concerns, but I still have several questions regarding the results.
> > > > >
> > > > > > I observed a significant decrease in performance without the in-context format in zero-shot performance. This suggests that the authors have demonstrated a substantial improvement in zero-shot performance by introducing few-shot prompt templates. However, this is confusing to me because both observations in the NLP community [1,2] and my experiments on VLMs indicate that incorporating few-shot prompt templates does not lead to such a substantial enhancement in zero-shot performance. In fact, such settings often result in a decline in zero-shot performance.
> > > > >
> > > > > I would appreciate it if the authors could provide detailed analysis to help me understand and resolve this confusion.
> > > > >
> > > > > [1] Iyer, Srinivas et al. “OPT-IML: Scaling Language Model Instruction Meta Learning through the Lens of Generalization.” ArXiv abs/2212.12017 (2022): n. pag.
> > > > >
> > > > > [2] Longpre, S. et al. “The Flan Collection: Designing Data and Methods for Effective Instruction Tuning.” International Conference on Machine Learning (2023).

---

> ### Author Response · Authors · 2023-11-23
> **Response to iVLP**
>
> Thank you for your thorough feedback.
>
> - The significant decrease in zero-shot performance without in-context format is due to the **model w/o in-context format** only incorporates **multi-modal data with interrelated images** into the training pipeline.
>
> While it enhances the model's ability to comprehend multiple images, it results in a decline in performance when it comes to understanding single image tasks.
>
> To clarify your query about the ablation studies concerning fewshot performance, we present the table from **Answer 2** below :
>
> 1. MMICL (Instructblip-xl): This is the standard model.
> 2. **w/o context scheme**: This model is trained only with instructions using all datasets from stage 2 of MMICL, without any additional design in context format and model architecture.
> 3. **w/o in-context format**: This model, devoid of in-context format data, comprises **multi-modal data with interrelated images and image declaration**.
> 4. **w/o interrelated images**: This model is trained without any interconnected images. Its training pipeline contains multi-modal in-context data and image declaration.
>
> | Task | ICL example | MMICL | w/o context scheme | w/o in-context-format | w/o interrelated images |
> | --- | --- | --- | --- | --- | --- |
> | VIZWIZ | zero-example | 27.10 | 28.13 | **15.90** | 23.43 |
> |  | 5-example | 42.66 | 36.23 | 28.33 | 37.13 |
> |  | Δ | 15.56 | 8.1 | 12.43 | 13.7 |
> | Flickr | zero-example | 83.94 | 77.84 | **65.33** | 76.88 |
> |  | 5-example | 88.11 | 79.38 | 65.95 | 77.09 |
> |  | Δ | 4.17 | 1.54 | 0.62 | 0.18 |
> | MME_Perception(Overall) | zero-example | 1303.59 | 1238.99 | 1141.02 | 1207.70 |
> | MME_Cognition (Overall) | zero-example | 370.71 | 316.79 | 345.36 | 333.21 |
> | Icon-QA | zero-example | 58.12 | 52.80 | 51.95 | 54.35 |
> | NLVR2 | zero-example | 72.45 | 56.65 | **62.63** | 59.60 |
> | Raven | zero-example | 32.00 | 8.00 | **28.00** | 16.00 |
>
> We believe that the **significant decrease in zero-shot performance without in-context format**(marked in bold) is due to the **impact of interrelated images setting** in the **model w/o in-context-format**. The discrepancy between training and evaluation results in this decline in zero-shot performance of the **w/o in-context-format**.
>
> The **interrelated images setting** maintain the model's comprehension ability on multiple images, outperforming other setups in the multi-image reasoning datasets (e.g., NLVR2 and Raven). However, this leads to a decrease in its **zero-shot performance**.
>
> In conclusion, our ablation study indicates that the **zero-shot performance improvement** is not due to the incorporation of few-shot prompt data. Instead, the **elimination of in-context format** makes the model only trained on multi-modal data with interrelated images,** causing a **performance decline in zero-shot settings**.
>
> The comparsion with the **w/o interrelated images** and **w/o context scheme**, indicates that the imbalance in data quantity, between interrelated and single images in the training data, results in trade-offs between the multi-image understanding and single image comprehension. The inclusion of multi-modal in-context data may alleviate this situation.
>
> In the **w/o interrelated images**, we completely removed the multi-modal data with interrelated images. The **results reflect your prediction**—incorporating few-shot prompt templates does not significantly affect the zero-shot performance.
>
> We also hope that you can kindly increase your score if our response has helped address your concerns.

---

### Official Review · Reviewer_7yvn · 2023-11-08

**Soundness:** 3 good
**Presentation:** 3 good
**Contribution:** 3 good
**Rating:** 5
**Confidence:** 5

**Summary:**

# Summary
The paper introduces MMICL (Multi-modal In-Context Learning), a novel approach designed to enhance Vision-Language Models (VLMs) in understanding complex multi-modal prompts, which include multiple images and text. The authors highlight that while Large Language Models (LLMs) excel in in-context learning from text prompts, VLMs lag behind, especially when dealing with prompts that involve intricate text-to-image references and relationships among multiple images. To overcome these limitations, the paper proposes a new context scheme that includes an image declaration section and image proxy tokens to improve the in-context learning capabilities of VLMs. Additionally, the authors have constructed a new dataset tailored to train VLMs on complex multi-modal prompts. The experimental results suggest that MMICL sets a new state-of-the-art in zero-shot performance on various vision-language tasks and benchmarks, demonstrating a significant improvement in understanding text-to-image references and relationships between images. MMICL also shows a reduction in language bias, which often leads VLMs to overlook visual content.

**Strengths:**

# Strengths

1. **Enhanced Multi-modal Understanding:** MMICL's approach to handling complex prompts with multiple images and text could significantly improve VLMs' performance on downstream tasks.

2. **State-of-the-Art Performance:** The paper reports new benchmarks in zero-shot performance on vision-language tasks, indicating a substantial advancement over existing models. On MME, MMICL seems to achieve the best average scores compared with current VLMs on cognition and perception tasks, indicating a strong performance. On MMBench, they also achieved SOTA performance which demonstrate the prominent ability of MMICL.

3. **Reduction in Language Bias:** MMICL reduces the chances of VLMs ignoring visual content, which is crucial for accurate multi-modal reasoning. Their experiments provided promising results.

**Weaknesses:**

# Weaknesses

Although I can spot many advantages in MMICL and I truly believe its a wonderful model. This paper also did a lot of experiments to demonstrate its ability. But here I should address few weaknesses and the authors should better clarify it to make the work's claims more sound.

1. The paper emphasizes the use of interleaved image-text data pairs and in-context learning data for training, suggesting this approach is beneficial for understanding complex multi-modal prompts. However, without experimental comparisons, it's unclear if interleaved pairs offer a substantial advantage over single image-text pairs. It's possible that similar results could be achieved without the interleaved structure. The authors should clarify whether the interleaved structure is a key contributor to performance improvements or if it primarily serves to enhance demonstrations and applicability to real-world scenarios.

2. The paper's focus on MME and MMBench, which involve fixed-format questions, might not fully represent the model's ability to handle freeform answers. Benchmarks like MM-VET[1], which require freeform answers and are evaluated by GPT-4, could provide a different perspective on the model's capabilities. A more diverse set of benchmarks, including those requiring freeform answers, would offer a more comprehensive evaluation of the model's performance and generalizability.

3. The paper does not fully discuss why MMICL's architecture is superior to the Flamingo or other paradigms[2] or other existing methods, which leaves the comparison incomplete. Many claims and designs in Section 2.1/2.2 seems just the author's considerations and are lacking sufficient reasons. Although conducting more comparative experiments may be challenging, the author should at least provide more explanations in these aspects to make the arguments in this paper more substantial. Instead of presenting a final result by combining all designs together, the author should provide further explanations to justify their choices.

For the dataset and training methodology to be validated, the authors should ideally show that the interleaved image-text pairs lead to better model performance than traditional single image-text pairs, across a range of tasks that include both fixed-format and freeform response requirements. Additionally, the paper should discuss any limitations of the proposed methods, such as potential overfitting to a specific data structure or benchmark, to provide a balanced and transparent evaluation of the approach.

[1] Yu, Weihao, et al. "Mm-vet: Evaluating large multimodal models for integrated capabilities." arXiv preprint arXiv:2308.02490 (2023).

[2] Yao, Zhewei, et al. "DeepSpeed-VisualChat: Multi-Round Multi-Image Interleave Chat via Multi-Modal Causal Attention." arXiv preprint arXiv:2309.14327 (2023).

**Questions:**

Most of my considerations are at the Weaknesses part. The authors may refer it and consider to address these questions.

1. Why using interleaved image-text data for instruction tuning an VLM would be considered beneficial? In terms of the model performance, is there any experimental comparison to support this claim?

2. Is there any quantitative results to support the models ability in free-form answering (evaluated by GPT-4)?

3. Why the MMICL's design is better for image-text interleaved chatting than Flamingo-based or other architectures?

---

> ### Author Response · Authors · 2023-11-19
>
> Thanks for your constructive reviews. It is encouraging to see you believe our work is wonderful. We sincerely thank you for your time and constructive comments. Below, we provide detailed replies to your comments to resolve your concerns.
>
> ### **Q1:**
>
> > The paper emphasizes the use of interleaved image-text data pairs and in-context learning data for training, suggesting this approach is beneficial for understanding complex multi-modal prompts. However, without experimental comparisons, **it's unclear if interleaved pairs offer a substantial advantage over single image-text pairs.** It's possible that similar results could be achieved without the interleaved structure. The authors should clarify whether the interleaved structure is a key contributor to performance improvements or if it primarily serves to enhance demonstrations and applicability to real-world scenarios.
> >
>
> > For the dataset and training methodology to be validated, the authors should ideally show that the **interleaved image-text pairs lead to better model performance than traditional single image-text pairs**, across a range of tasks that include both fixed-format and freeform response requirements.
> >
>
> > Why using **interleaved image-text data** for instruction tuning an VLM would be considered beneficial? In terms of the model performance, is there any **experimental comparison** to support this claim?
> >
>
> ### **A1** :
>
> - **Using interleaved image-text data outperforms single image-text data by a large margin, according to ablation experiments.**
>
> We conduct ablation experiments using the Instructblip-xl as the backbone model in the following settings to verify the effectiveness of interleaved image-text during multi-modal ICL tuning:
>
>   -  **MMICL**: This is the standard model.
>
>   -   **w/o context scheme**: This model is trained only with instructions using all datasets from stage 2 of MMICL, without any additional design in context format and model architecture.
>
>   -  **w/o image declaration**: This model is trained without image declaration but includes multi-modal data with interconnected images and in-context data.
>
>   -  **w/o in-context format**: This model, devoid of in-context format data, comprises multi-modal data with related images and image declaration.
>
>   -  **w/o interrelated images**: This version of the model is trained without any interconnected images. Its training pipeline contains multi-modal in-context data and image declaration.
>
> We use the **equivalent amount** of data as in stage 2 of MMICL for the ablation models.
>
> - We employ the ablation study on the **general benchmark(MME)** and **multi-image reasoning datasets(Raven, Icon-QA & NLVR2)** to comprehensively evaluate the sources of the observed performance improvement in MMICL. The ablation results affirm that our **interleaved image-text data scheme is a key factor in** observed improvements, clarifying its contribution to MMICL's performance. The superiority of MMICL is driven by the collective impact of our design elements—removing any component cannot guarantee the superior performance of our model. Each component of our design significantly contributes to different aspects of our model.
>
> | Task | MMICL | w/o context scheme | w/o image declaration | w/o in-context format | w/o interrelated images |
> | --- | --- | --- | --- | --- | --- |
> | MME_Perception | **1303.59** | 1238.99 | 1170.87 | 1141.02 | 1207.70 |
> | MME_Cognition | **370.71** | 316.79 | 341.07 | 345.36 | 333.21 |
> | Icon-QA | **58.12** | 52.80 | 47.15 | 51.95 | 54.35 |
> | NLVR2 | **72.45** | 56.65 | 61.0 | 62.63 | 59.60 |
> | Raven | **32.00** | 8.00 | 18.00 | 28.00 | 16.00 |
>
> The results confirm that the **performance improvement** in MMICL arises from **our proposed interleaved image-text data pairs and in-context learning data**. Additional single image-text pair data for  SFT(supervised fine tuning) does not yield comparable enhancements in the targeted areas, underscoring the unique value of our proposed method.
>
> We have **included the ablation experiments** and analysis in the **second section of Sec 3.6** in the revised paper.

---

> > ### Author Response · Authors · 2023-11-19
> > **Continue Response for Question 1**
> >
> > ### **Continue A1**:
> > - Using the interleaved image-text data for instruction tuning a VLM would **enhance MMICL in several aspects:**
> >     - **Improved Multi-Modal Abilities:**
> >
> >         Our approach achieves superior performance in **text-to-image reference, complex multi-image reasoning, and multi-modal in-context learning**.
> >
> >         In the paper, we use two experiments to show this: On the one hand, Raven test is widely used to evaluate nonverbal reasoning and complex multi-image understanding ability. Table 3 (**Sec. 3.3.2**) shows that MMICL gains a 12-point improvement compared to the KOSMOS-1, which is **trained on interleaved image-text data**.
> >
> >         On the other hand, MMICL achieves a 13-point improvement compared to Blip2, which is trained on single image-text data on the Winoground (**Sec.3.3.1**).
> >
> >     We further compared the model with **multi-modal instruction tuning**(**otter**) on **multi-image** datasets (Nlvr2 and iconqa-img), and our method consistently outperforms, as shown in the provided table.
> >
> >     |  | MMICL | OTTER | InstructionBlip |
> >     | --- | --- | --- | --- |
> >     | NLVR2 | **66.6%** | 47.2% | 53.95% |
> >     | iconqa-img | **60.85%** | 34.1% | 45.10% |
> >     | Raven | **34.00%** | 22.0% | 10.0% |
> >     - **Enhanced General Benchmark Performance**
> >     - The proposed scheme significantly contributes to **improved general performance** across diverse benchmarks by addressing the language bias hallucination problem. Table 5 **(Sec. 3.5)** presents results from the ScienceQA-Img experiment, demonstrating MMICL's effectiveness in **overcoming** VLM's **hallucination** issue. This advantage extends to single-image tasks, showcasing the robustness of our approach.
> >     - Generally, compared to methods that utilize single **image-text pairs multi-modal Instruction Tuning** (LLAVA, MiniGPT-4, InstructBLIP, Shikra), MMICL demonstrates superior performance in general **benchmarks** (Sec. 3.2, Appendix I), **in-context learning ability**(Sec. 3.4), **mitigating language bias hallucination**(Sec. 3.5), **multi-image understanding**(Sec. 3.6) and **Out of domain generalization** **ability**(Appendix P).
> >     - Similarly, when compared to other methods that **use interleaved image-text data pairs for training** (like Otter, Kosmos-1), MMICL also shows superior performance in general **benchmarks** (Sec. 3.2, Appendix I) **and multi-image reasoning(**Sec 3.3.2**).**

---

> ### Author Response · Authors · 2023-11-19
>
> ### **Q2**:
>
> > The paper's focus on MME and MMBench, which involve fixed-format questions, might not fully represent the model's ability to handle **freeform answers**. Benchmarks like **MM-VET**[1], which require freeform answers and are evaluated by GPT-4, could provide a different perspective on the model's capabilities. A more diverse set of benchmarks, including those requiring freeform answers, would offer a more comprehensive evaluation of the model's performance and generalizability.
> >
> >
> > Is there any **quantitative results** to support the models ability in **free-form answering** (evaluated by GPT-4)?
> >
>
> ### **A2** :
>
> MMICL shows **superior performance in free-form answering tasks**, such as **WebSRC and Vizwiz**(already in Table 4), **VisDial** (Sec. 3.6) and **MSVD QA and iVQA**(Appendix K). And MMICL shows **significant improvement on MM-VET** compared to the baseline model.
>
> - **MM-VET** indeed serves as an excellent benchmark; thank you for bringing it to our attention. Below are the detailed results.
>     - We have **included the free-form answering evaluation of MMICL on MM-VET** in **Appendix R** in the revised paper.
>
>     |  | rec | ocr | know | gen | spat | math | total |
>     | --- | --- | --- | --- | --- | --- | --- | --- |
>     | InstructBlip-t5 | 17.1 | 8.4 | 5.5 | 2.6 | 8.0 | 3.8 | 14.1 |
>     | MMICL-t5 | **29.9** | **14** | **17.4** | **11.2** | **18.1** | **3.8** | **24.8** |
>     - It is noteworthy that compared to the Flant5 backbone's instruct blip model, MMICL, with the identical backbone, **exhibits significant enhancement** on MM-VET. This testifies that our proposed method **prevents the model from overfitting a specific data structure** and substantially augments the model's **free-form answering capability**.
>     - The performance of MMICL on MM-VET is relatively low due to MMICL employing **the Flant5 backbone model**. As T5 is mainly finetuned on NLP benchmarks containing many multi-choice QA and classification datasets, it tends to **output shorter content** during freeform question answering, which is disadvantageous when evaluated with GPT4. This feature is also discovered in Instructionblip[1].
>     - The **Vicuna backbone** model generally outperforms the flant5 backbone model for open-domain QA questions.
>     - Later, we will add the evaluation result on the MM-VET using the MMICL with the vicuna backbone.
> - We have evaluated the **free-form answering ability of MMICL across various tasks.**
>     - Table 4 (**Sec. 3.4**) displays the effectiveness of MMICL on **WebSRC and Vizwiz**, the datasets targeted for open-domain question answering. Its performance on **VisDial**, a focus area for visual dialog, is presented in Table 6 (**Sec. 3.6**). Furthermore, the results of MMICL on the video datasets **MSVD QA and iVQA** are presented in Table 22 (**Appendix K**),  the datasets focus on the open-domain video question answering.
>
>     Those experiments all together showcased MMICL’s superior  performance within free-form answering scenarios.
>
> - However, the **key innovation insight** of our paper is focused on **augmenting the ICL ability of the VLM** rather than improving the model's ability to handle freeform answers.
>
>     As shown in Sec. 3.3-Sec. 3.6, MMICL shows remarkable performance in **handling complex multi-image reasoning and multi-modal in-context learning.** The SOTA performance on the Raven and Bonground-HOI dataset shows the MMICL has impressive multi-image reasoning abilities. The 13-point improvement on the Winoground shows the MMICL understands complex text-to-image reference. Moreover, MMICL demonstrates impressive multi-modal ICL performance across various tasks.
>
> [1] Dai, W., Li, J., Li, D., Tiong, A.M., Zhao, J., Wang, W., Li, B.A., Fung, P., & Hoi, S.C. (2023). InstructBLIP: Towards General-purpose Vision-Language Models with Instruction Tuning. *ArXiv, abs/2305.06500*.

---

> > ### Author Response · Authors · 2023-11-19
> >
> > ## **Q3**:
> >
> > > The paper does not fully discuss **why MMICL's architecture is superior to the Flamingo or other paradigms[2]** or other existing methods, which leaves the comparison incomplete. Many claims and designs in Section 2.1/2.2 seems just the author's considerations and are lacking sufficient reasons. Although conducting more comparative experiments may be challenging, the author should at least provide more explanations in these aspects to make the arguments in this paper more substantial. Instead of presenting a final result by combining all designs together, the author should provide **further explanations to justify their choices**.
> > >
> >
> > ## **A3:**
> >
> > 1. **Design choices for model structure:**
> >
> >     Our model structure, as depicted in Figure 2, diverges from the Flamingo-style model, which **inserts new gated cross-attention layers into the LLM to inject visual features**. We argue that this approach **does not ensure equal treatment of image and text inputs** within the language model. Instead, we leverage the Q-former to encode images into embeddings understandable by the language model, constructing interleaved image-text inputs for the backbone model. **This design is a natural extension of the original attention mechanism in LLMs**.
> >
> >     The results presented in **Table 3 and Table 6** demonstrate that our model structure, when compared to the **Flamingo-style models (Kosmos, Otter)**, better understands the relationships between images and text, successfully addressing multi-image reasoning and image hallucination problems.
> >
> >     Additionally, our model's text-to-image reference performance, shown in the following table, illustrates a significant improvement, even compared to **GPT-4V**.
> >
> >     - We have updated the experiment results and explanations in **Table 2 and Sec 2.1** in the revised paper.
> >
> >
> >         | Winoground  | Text | Image | **Group** |
> >         |---------|------|-------|-------|
> >         | Otter   | 27.25| 22.75 | 11.75 |
> >         | BLIP2   | 44.00| 26.00 | 23.50 |
> >         | GPT-4V[2]| *69.25*| *46.25* | *39.25* |
> >         | MMICL   |  45.00  | 44.99 | **43.00** |
> >
> >     - Furthermore, our proposed context scheme and multi-modal in-context tuning can also be applied to the Flamingo-style model. We are confident that it would perform well on such models, as the **limitations we aim to address** are prevalent across various VLMs, making it a promising area for our future research.
> >
> > 2. **Design choices for context-scheme；**
> >
> >     To address the limitations of VLMs in handling complex multi-modal prompts, our proposed context-scheme incorporates three key designs:
> >
> >     1. **Image Declaration:** Enhances the VLM's understanding of text-to-image reference, outperforming the Single image-text training method **(Blip2) (Table 2)** by **13 points on Winoground.**
> >     2. **Multiple Interconnected Images Format**: Aids the VLM in better understanding the relationship between multiple images, resulting in a **12-point improvement on Raven** compared to the Single image-text training model **(Kosmos) (Table 3)** and **15-point, 13-point improvement on MSVD-QA and iVQA** compared to Single image-text training model(**Flamingo-80B**) **(Table 22).**
> >     3. **Unified Multi-modal ICL Data Format**: Assists the VLM in better learning from multi-modal in-context demonstrations, showcasing impressive performance across various tasks compared to the methods involving **Single image-text training (Flamingo and Kosmos)(Table 6).**
> > 3. **Comparison with Concurrent Work (Deepspeed-Visual Chat):**
> >
> >     Considering that Deepspeed-Visual Chat was released on **September 25**, **subsequent to our release**, we did not include a comparison experiment with deepspeed-visual chat in the current paper.
> >
> >     We plan to augment our revised paper with additional experiments that include a comparison with Deepspeed-Visual Chat.
> >
> >
> > We trust that these detailed explanations clarify the rationale behind our design choices and further justify our approach.
> >
> > [2] Wu, Y., Zhang, P., Xiong, W., Oğuz, B., Gee, J.C., & Nie, Y. (2023). The Role of Chain-of-Thought in Complex Vision-Language Reasoning Task.

---

> ### Author Response · Authors · 2023-11-19
>
> ### **Q4:**
>
> > Additionally, the paper should discuss any limitations of the proposed methods, such as potential overfitting to a specific data structure or benchmark, to provide a balanced and transparent evaluation of the approach.
> >
>
> ### **A4:**
>
> Thank you for your detailed review, and we appreciate the opportunity to further discuss the limitations of our current method:
>
> 1. **Potential Overfitting:**
>
>     We **mitigate the risk of overfitting** specific data structures by expanding the instruction templates with ChatGPT, **ensuring diversity** in the MIC dataset. **Table 25 in Appendix P** demonstrates the generalization ability of MMICL to an **unseen domain in Minecraft**. MMICL is able to generalize in the Minecraft domain even if the images are extremely different compared to the images used by training. This **out-of-domain understanding** ability demonstrates that MMICL does not overfit a specific data structure.
>
> 2. **Context length Constraints:**
>
>     The **context lengt**h of the backbone language model **limits the number of images** that can be included in the input. It makes our MMICL cannot accept unlimited images. Therefore, during MIC tuning, we incorporate up to 8 images per instance. However, MMICL exhibits **robust generalization beyond this limit.** For instance, on evaluations of the Bonground-HOI dataset, MMICL achieves remarkable performance even with a context of **14 images.** Similarly, in the context of video datasets, MMICL maintains great performance with input up to **12/16 frames**. We attribute this impressive generalization to the relative position embedding employed by Flant5 in its architecture.
>
> 3. **Model Exploration on Decoder-Only Architectures:**
>
>     We acknowledge that our exploration has focused on the T5 series models, and the effectiveness of our proposed approach to decoder-only architectures has not been fully explored. This aspect of our approach will be thoroughly investigated and analyzed.
>
>
> We have **included the discussion of the limitation of our work** in **Appendix T** in the revised paper.

---

> > ### Author Response · Authors · 2023-11-21
> >
> > Dear Reviewer 7yvn,
> >
> > Thank you again for your valuable feedback and comments! We would greatly appreciate it if you could let us know whether you are satisfied with our response. We will be happy to address any remaining concerns.
> >
> > Sincerely,
> >
> > Paper4878 Authors

---

> ### Comment · Reviewer_7yvn · 2023-11-21
> **Response to Authors Rebuttal**
>
> Thank you for your feedback!
>
> I believe these experimental results can indeed help us better understand the method of MMICL. However, after considering the opinions of other reviewers, especially #eWon, I agree with their point that the paper lacks an adequate description of the dataset itself. Specifically, we don't know where the insights for constructing this dataset came from. Was there any insight that led to the plan of constructing the dataset with an interleaved finetuning approach being better than single image finetuning?
>
> Moreover, the introduction of this dataset is indeed lacking rigor, and there is no way to obtain even a high-level understanding of the dataset construction process from the main text. I believe this is the most crucial factor hindering my understanding of the logical fluency of this paper.
>
> Until the aforementioned issues are clarified, I consider the contribution of this paper to be limited to a model that performs well on benchmarks, without demonstrating any significant emerging abilities. Therefore, I'm sorry to say that I currently don't think it is suitable to be accepted as an ICLR paper.

---

> > ### Author Response · Authors · 2023-11-21
> > **Response to Reviewer 7yvn**
> >
> > Thank you for your thorough response.
> > Regarding the questions you raised, we are willing to provide the following clarifications:
> >
> > **1. Insights for constructing our dataset.**
> >
> > Our initial insight is quite straightforward. We believe that the ability of VLMs to understand complex multimodal contexts (e.g., multi-image reasoning, text-to-image reference) could emerge from the ability to learn from multimodal demonstrations in context (ICL). Since ICL requires the model to refer to the examples in context to solve the current question, which is quite similar to the complex multimodal context scenarios.
> >
> > Currently, it is difficult to naturally find supervised multimodal instruction training data/tasks that necessitate the ability to comprehend complex multimodal contexts. Ideally, constructing datasets that require ICL ability is feasible. This is because we only need to reorganize the original SFT dataset by grouping multiple examples as an input. This requires the model to predict the answer to another question, given all the multimodal demonstrations. However, in reality, the model structure and data organization methods in the current training pipeline for VLMs hampers the acquisition of ICL abilities. For instance, they only support a single image(InstructBlip, llava, MiniGPT4) as input and their training tasks do not necessitate the model to understand multiple images(Flamingo, Kosmos).
> >
> > To address the previously mentioned challenges, we have developed the MIC dataset. It consists of three scheme designs as detailed in Section 2.2. Through extensive experiments and meticulously designed ablation studies, we have validated our initial insight. Our model significantly outperforms its counterparts by a substantial margin.
> >
> > **2. Description of MIC dataset (Sec 2.3 & Appendix D)**
> >
> > The comprehensive **illustration** of **data construction** can be found in  **Fig 4, Fig 9, and Fig 10**.
> >
> > For an in-depth understanding, we also provide the **detailed dataset** curation **pipeline** in **Section 2.3.**
> >
> > The data construction process across three different data sources can be found in **Appendix D**:
> >
> > 1.  Multi-model ICL format data gleaned from image-text pair datasets,
> > 2. Interleaved-image-text data derived from the VCR dataset,
> > 3. Multi-image data extracted from video datasets.
> >
> > Please kindly refer to Response '**A1**' of the Reviewer **eWon**, along with the details in Section 2.3, Appendix D, and Fig 4, 9, and 10 in the **revised** version of the paper.
> >
> > **3. Emerging abilities of MMICL**
> >
> > Our model exhibits **emerging abilities** across multiple domains and aspects.
> >
> > Regarding multi-image comprehension, our method equips the model with the ability to understand multiple images and references between images and texts, which previous models **lacked**.
> >
> > Furthermore, our model demonstrates **new abilities** in single-image understanding such as out-of-domain generalization, free-form answering, and a reduction in hallucination. These capabilities indicate that the model can expand its learning beyond the knowledge presented during the training phase.
> >
> > It strongly demonstrates our significant emerging abilities.

---

> > > ### Author Response · Authors · 2023-11-23
> > >
> > > Dear Reviewer 7yvn,
> > >
> > > We sincerely appreciate your careful review and valuable feedback, which has substantially improved our paper.
> > >
> > > Your feedback is highly valuable to me, and I want to make sure my responses effectively address the issues you have had.
> > > If you have any further questions or require additional clarification, please feel free to reach out.
> > >
> > > We also hope that you can kindly increase your score if our response has helped address your concerns.
> > >
> > > Best,
> > >
> > > Authors.

---

> ### Comment · Reviewer_7yvn · 2023-12-05
> **Final Comment**
>
> After reading all the reviews and responses, I believe this paper does demonstrate the value and significance of introducing in-context learning to MMICL model. However, whether the initial or the final version paper, the presentation was confusing and made it difficult for me to grasp the content intuitively, e.g. why it comes from your data? I also have to point out, as Reviewer #eWon mentioned, that the article is hard to read and the results are confusing.
>
> Taking into account all the reviewers' suggestions, I intend to give an increase of my original score. However, other reviewers have also expressed many concerns, so I intend to maintain a score of 5.

---

### Author Response · Authors · 2023-11-20
**General Response to all Reviewers**

Dear Reviewers,

We thank all reviewers for their insightful comments and acknowledgment of our contributions.

**We greatly appreciate your recognition of the strengths of our work as follows:**

- **Introduction of MMICL:**

    - We present the MMICL, **recognized by `7yvn` as a novel method**, offering a **novel perspective(`iVLP`)** to improve VLMs' understanding of complex multi-modal prompts. It excels in **handling multiple images and their relationships (`d1dY`).**
    - The key innovation is our proposed "**novel context schema"(`QUmJ`)**, acknowledged as '**a pioneering context schema**'(`iVLP`), designed to enhance VLMs in "handling multimodal inputs and facilitating in-context learning"(`iVLP`). Under the scheme, we successfully "**unify more tasks" (`d1dY`)**.
- **Performance Improvement:**
    - MMICL demonstrates strong comprehension of complex multi-modal prompts **(`d1dY`)** and improves understanding of text-to-image references and relationships between images **(`7yvn`)**.
    - Our method enhances the Multi-modal Understanding ability of VLM, achieving **State-of-the-Art Performance**, as **acknowledged unanimously by all reviewers**.
    - The "**promising results**" of MMICL(`7yvn`) showcase a "**significant reduction**"(`7yvn`) in **language bias** often seen in VLMs, which is "**crucial** for accurate multi-modal reasoning"(`7yvn`).
- **Comprehensive Experiments:**
    - Our work is acknowledged for its comprehensiveness by **most reviewers**(`d1dY`, `iVLP`, `d1dY`), addressing an "**interesting problem**" (`QUmJ`), and showcasing advantages on several well-known datasets(`QUmJ`).
    - **Extensive experiments**(`QUMJ`) average high scores compared to baselines, demonstrating the **state-of-the-art** results on a wide range of general vision-language **tasks and benchmarks** (**`eWon`,`7yvn`,`iVLP`**)**.**
    - The experiments reveal MMICL's potential for **in-context learning(`d1dY`)**. MMICL exhibits in-context learning ability on verious tasks under the few-shot setting(`iVLP`,`eWon`).
- **Dataset Contribution:**
    - The proposed MIC dataset, covering a wide range of tasks (`iVLP`), is highlighted by **`eWon`** as a **novel contribution** to the **multimodal community**.

**We've revised our manuscript per the reviewers' suggestions** (**highlighted in red** in the uploaded revision pdf). Detailed responses to each reviewer's concerns are carefully addressed point-by-point.

Below summarize the **major updates** we've made:

- **Presentation:**

    `QUmJ`(**Abstract**) A explicit meaning of MMICL (vision-language Model with **M**ulti-**M**odal **I**n-**C**ontext **L**earning).

    `QUmJ`(**Table 1 and 5**)  Include the model sizes in Tables.

    `eWon`(**Table 4**)  Revise the terminology to “w/o ICL examples” in Table 4:

    `iVLP`(**Table 4**) Revise the abnormal Flickr30k experiment results due to the failure of prediction pre-processing.

    `eWon`(**Table 9**)  Update all the licenses for the datasets marked as "Unknown".

    `eWon` Revise the typos in the paper

- **Experiment:**

    `7yvn`(**Table 2**)  Add the result of GPT-4V on the Winoground. Fix the experiment results due to the  loss of precision.

    `7yvn`(**Table 3**)  Add the results of InstructBlip and Otter on Raven IQ test.

    `iVLP` `7yvn` `eWon` `didY` (**Table 7, Table 26, and Appendix N**)  Add an ablation study in various settings to verify the effectiveness of our context scheme.

    `7yvn`(**Appendix R and Table 27**)  Add the free-form answering evaluation of MMICL.

    `7yvn`(**Appendix S**) Add the comparison of other multi-modal instruction tuning models.

- **Explaination:**

    `7yvn`(**Sec. 2.1**)  Add further explanations to justify our design choices.

    `eWon`(**Figure 4, Figure 9, and Figure 10**) Present **visual examples** of an **automatic data construction pipeline** for multi-model data with interconnected images, in-context examples or video.

    `eWon`(**Sec. 2.3**)  Revise the Sec. 2.3 and provide more details on how we use the **dataset construction pipeline to convert the open-source datasets into the MIC dataset according to our proposed context scheme.**

    `eWon`(**Appendix D**) A **comprehensive illustration** of the **data construction pipeline** for multi-model data with interconnected images, in-context examples, or video.

    `eWon`(**Sec.3.5 and Appendix Q**)  Provide details of the ScienceQA-IMG that we manually split. Share the information of human annotators involved in the separation process, as well as the rationales we provide to annotators.

    `7yvn`(**Appendix T**) Add the limitation discussion of our paper.


We believe our work could make a novel contribution to the multi-modal community and offer a novel perspective on bolstering VLMs to handle multi-modal inputs.

We would like to be involved in further discussions if any question is raised.

Best,

Authors.

---

### Meta-Review · Area_Chair_c5aQ · 2023-12-16

**Metareview:**

**Scientific Claims and Findings**:
The paper introduces MMICL (Multi-modal In-Context Learning), an approach designed to enhance Vision-Language Models (VLMs) in understanding complex multi-modal prompts, which include multiple images and text. MMICL addresses the limitations of VLMs in handling intricate text-to-image references and relationships among multiple images. It proposes a new context scheme involving an image declaration section and image proxy tokens to improve in-context learning. The paper also presents the construction of the Text-Image Interleaved Dataset (MIC dataset) for training VLMs on complex multi-modal prompts. MMICL achieves state-of-the-art zero-shot performance on various vision-language benchmarks, demonstrating significant improvements in understanding text-to-image references, relationships between images, and a reduction in language bias.

**Strengths of the Paper**:

- Innovative Context Scheme: MMICL introduces a novel context schema, including image declaration and proxy tokens, to enhance in-context learning for VLMs, addressing challenges in handling complex multi-modal prompts.
- Comprehensive Dataset: The Text-Image Interleaved Dataset (MIC dataset) covers a wide range of vision-language tasks, providing a diverse training set to improve the model's understanding of different types of prompts.
- Good Performance: MMICL achieves new benchmarks in zero-shot performance on vision-language tasks, especially on MME and MMBench benchmarks. The model demonstrates superiority over existing VLMs on cognition and perception tasks.
- Reduction in Language Bias: MMICL shows a promising reduction in language bias, ensuring VLMs don't overlook visual content, a critical aspect of accurate multi-modal reasoning.

**Weaknesses of the Paper**:

- Limited Few-Shot Improvement: The few-shot performance improvement of MMICL appears somewhat limited, especially in comparison to certain baselines. The paper should provide more insights into the factors influencing this limitation.
- Lack of Ablation Study: The paper lacks an ablation study on the synthetic dataset, training paradigms, and model structure, making it unclear which components contribute significantly to the model's performance. A more in-depth analysis of these aspects would strengthen the paper.
- Complex Design and Prompt Engineering: Some reviewers mention that the model's design is slightly complex, involving extensive prompt engineering. The paper could benefit from clearer justifications and explanations for these design choices to enhance understanding and reproducibility.
- Benchmark Diversity: The focus on benchmarks like MME and MMBench, which involve fixed-format questions, might limit the representation of the model's capabilities. The paper could include more diverse benchmarks, including those requiring freeform answers, to provide a more comprehensive evaluation.

**Justification For Why Not Higher Score:**

- Justification for Design Choices: The paper lacks detailed explanations justifying specific design choices, especially in Section 2.1/2.2.
- Discussion on Few-Shot Performance: The paper should include a more detailed discussion on the limitations or factors influencing the limited improvement in few-shot scenarios, as observed in the experiments.
- Benchmark Diversity: Including a wider range of benchmarks, especially those requiring freeform answers, would offer a more holistic assessment of MMICL's capabilities and generalizability.

**Justification For Why Not Lower Score:**

- Comprehensive Experiments and State-of-the-Art Results: The paper conducts comprehensive experiments on a wide range of vision-language tasks, demonstrating the effectiveness of the proposed approach.

- Relevance and Timeliness: The problem addressed in the paper—enhancing VLMs' understanding of complex multi-modal prompts—is highly relevant and timely. The proposed approach tackles a specific limitation in existing VLMs and provides a promising solution, contributing to the ongoing advancements in the field of vision-language models. The paper's relevance enhances its overall merit and potential impact in the research community.

---

### Decision · Program_Chairs · 2024-01-16

Accept (poster)